# On the Expected Complexity of Maxout Networks

**Hanna Tseran**
Max Planck Institute for Mathematics in the Sciences
04103 Leipzig, Germany
`hanna.tseran@mis.mpg.de`

**Guido Montúfar**
Department of Mathematics and Department of Statistics, UCLA
Los Angeles, CA 90095, USA;
Max Planck Institute for Mathematics in the Sciences
04103 Leipzig, Germany
`montufar@math.ucla.edu`

## Abstract

Learning with neural networks relies on the complexity of the representable functions, but more importantly, the particular assignment of typical parameters to functions of different complexity. Taking the number of activation regions as a complexity measure, recent works have shown that the practical complexity of deep ReLU networks is often far from the theoretical maximum. In this work, we show that this phenomenon also occurs in networks with maxout (multi-argument) activation functions and when considering the decision boundaries in classification tasks. We also show that the parameter space has a multitude of full-dimensional regions with widely different complexity, and obtain nontrivial lower bounds on the expected complexity. Finally, we investigate different parameter initialization procedures and show that they can increase the speed of convergence in training.

## 1 Introduction

We are interested in the functions parametrized by artificial feedforward neural networks with maxout units. Maxout units compute parametric affine functions followed by a fixed multi-argument activation function of the form $(s_1, \ldots, s_K) \mapsto \max\{s_1, \ldots, s_K\}$ and can be regarded as a natural generalization of ReLUs, which have a single-argument activation function $s \mapsto \max\{0, s\}$. For any choice of parameters, these networks subdivide their input space into activation regions where different pre-activation features attain the maximum and the computed function is (affine) linear. We are concerned with the expected number of activation regions and their volume given probability distributions of parameters, as well as corresponding properties for the decision boundaries in classification tasks. We show that different architectures can attain very different numbers of regions with positive probability, but for parameter distributions for which the conditional densities of bias values and the expected gradients of activation values are bounded, the expected number of regions is at most polynomial in the rank $K$ and the total number of units.

**Activation regions of neural networks** For neural networks with piecewise linear activation functions, the number of activation regions serves as a complexity measure and summary description which has proven useful in the investigation of approximation errors, Lipschitz constants, speed of convergence, implicit biases of parameter optimization, and robustness against adversarial attacks. In particular, Pascanu et al. (2013); Montúfar et al. (2014); Telgarsky (2015, 2016) obtained depth separation results showing that deep networks can represent functions with many more linear regions than any of the functions that can be represented by shallow networks with the same number of

units or parameters. This implies that certain tasks require enormous shallow networks but can be solved with small deep networks. The geometry of the boundaries between linear regions has been used to study function-preserving transformations of the network weights (Phuong and Lampert, 2019; Serra et al., 2020) and robustness (Croce et al., 2019; Lee et al., 2019). Steinwart (2019) demonstrated empirically that the distribution of regions at initialization can be related to the speed of convergence of gradient descent, and Williams et al. (2019); Jin and Montúfar (2020) related the density of breakpoints at initialization to the curvature of the solutions after training. The properties of linear regions in relation to training have been recently studied by Zhang and Wu (2020). The number of regions has also been utilized to study the eigenvalues of the neural tangent kernel and Lipschitz constants (Nguyen et al., 2020).

**Maximum number of regions** Especially the maximum number of linear regions has been studied intensively. In particular, Montúfar (2017); Serra et al. (2018) improved the upper bounds from Montúfar et al. (2014) by accounting for output dimension bottlenecks across layers. Hinz and Van de Geer (2019) introduced a histogram framework for a fine grained analysis of such dimensions in ReLU networks. Based on this, Xie et al. (2020); Hinz (2021) obtained still tighter upper bounds for ReLU networks. The maximum number of regions has been studied not only for fully connected networks, but also convolutional neural networks (Xiong et al., 2020), graph neural networks (GNNs) and message passing simplicial networks (MPSN) (Bodnar et al., 2021).

**Expected number of regions** Although the maximum possible number of regions gives useful complexity bounds and insights into different architectures, in practice one may be more interested in the expected behavior for typical choices of the parameters. The first results on the expected number of regions were obtained by Hanin and Rolnick (2019a,b) for the case of ReLU networks or single-argument piecewise linear activations. They show that if one has a distribution of parameters such that the conditional densities of bias values are bounded and the expected gradients of activation values are bounded, then the expected number of linear regions can be much smaller than the maximum theoretically possible number. Moreover, they obtain bounds for the expected number and volume of lower dimensional linear pieces of the represented functions. These results do not directly apply to the case of maxout units, but we will adapt the proofs to obtain corresponding results.

**Regions of maxout networks** Most previous works focus on ReLUs or single-argument activation functions. In this case, the linear regions of individual layers are described by hyperplane arrangements, which have been investigated since the 19th century (Steiner, 1826; Buck, 1943; Zaslavsky, 1975). Hence, the main challenge in these works is the description of compositions of several layers. In contrast, the linear regions of maxout layers are described by complex arrangements that are not so well understood yet. The study of maxout networks poses significant challenges already at the level of individual layers and in fact single units. For maxout networks, the maximum possible number of regions has been studied by Pascanu et al. (2013); Montúfar et al. (2014); Serra et al. (2018). Recently, Montúfar et al. (2021) obtained counting formulas and sharp (asymptotic) upper bounds for the number of regions of shallow (deep) maxout networks. However, their focus was on the maximum possible value, and not on the generic behavior, which we investigate here.

**Related notions** The activation regions of neural networks can be approached from several perspectives. In particular, the functions represented by networks with piecewise linear activations correspond to so-called tropical rational functions and can be studied from the perspective of tropical geometry (Zhang et al., 2018; Charisopoulos and Maragos, 2018). In the case of piecewise affine convex nonlinearities, these can be studied in terms of so-called max-affine splines (Balestriero et al., 2019). A related but complementary notion of network expressivity is trajectory length, proposed by Raghu et al. (2017), which measures transitions between activation patterns along one-dimensional paths on the input space, which also leads to depth separation results. Recent work (Hanin et al., 2021) shows that ReLU networks preserve expected length.

**Contributions** We obtain the following results for maxout networks.

- There are widely different numbers of linear regions that are attained with positive probability over the parameters (Theorem 7). There is a non-trivial lower bound on the number of linear regions that holds for almost every choice of the parameters (Theorem 8). These results advance the maximum complexity analysis of Montúfar et al. (2021) from the perspective of generic parameters.

- For common parameter distributions, the expected number of activation regions is polynomial in the number of units (Theorem 9). Moreover, the expected volume of activation regions of different

dimensions is polynomial in the number of units (Theorem 10). These results correspond to maxout versions of results from Hanin and Rolnick (2019b) and Hanin and Rolnick (2019a).

- For multi-class classifiers, we obtain an upper bound on the expected number of linear pieces (Theorem 11) and the expected volume (Theorem 12) of the decision boundary, along with a lower bound on the expected distance between input points and decision boundaries (Corollary 13).
- We provide an algorithm and implementation for counting the number of linear regions of maxout networks (Algorithm 1).
- We present parameter initialization procedures for maxout networks maximizing the number of regions or normalizing the mean activations across layers (similar to Glorot and Bengio 2010; He et al. 2015), and observe experimentally that these can lead to faster convergence of training.

## 2 Activation regions of maxout networks

We consider feedforward neural networks with $n_0$ inputs, $L$ hidden layers of widths $n_1, \ldots, n_L$, and no skip connections, which implement functions of the form $f = \psi \circ \phi_L \circ \cdots \circ \phi_1$. The $l$-th hidden layer implements a function $\phi_l \colon \mathbb{R}^{n_{l-1}} \to \mathbb{R}^{n_l}$ with output coordinates, i.e. units, given by trainable affine functions followed by a fixed real-valued activation function, and $\psi \colon \mathbb{R}^{n_L} \to \mathbb{R}^{n_{L+1}}$ is a linear output layer. We denote the total number of hidden units by $N = n_1 + \cdots + n_L$, and index them by $z \in [N] := \{1, \ldots, N\}$. The collection of all trainable parameters is denoted by $\theta$.

We consider networks with maxout units, introduced by Goodfellow et al. (2013). A rank-$K$ maxout unit with $n$ inputs implements a function $\mathbb{R}^n \to \mathbb{R}; x \mapsto \max_{k \in [K]}\{w_k \cdot x + b_k\}$, where $w_k \in \mathbb{R}^n$ and $b_k \in \mathbb{R}$, $k \in [K]$, are trainable weights and biases. The activation function $(s_1, \ldots, s_K) \mapsto \max\{s_1, \ldots, s_K\}$ can be regarded as a multi-argument generalization of the rectified linear unit (ReLU) activation function $s \mapsto \max\{0, s\}$. The $K$ arguments of the maximum are called the pre-activation features of the maxout unit. For unit $z$ in a maxout network, we denote $\zeta_{z,k}(x; \theta)$ its $k$-th pre-activation feature, considered as a function of the input to the network.

For any choice of the trainable parameters, the function represented by a maxout network is piecewise linear, meaning it splits the input space into countably many regions over each of which it is linear.

**Definition 1** (Linear regions). Let $f \colon \mathbb{R}^{n_0} \to \mathbb{R}$ be a piecewise linear function. A linear region of $f$ is a maximal connected subset of $\mathbb{R}^{n_0}$ on which $f$ has a constant gradient.

We will relate the linear regions of the represented functions to activation regions defined next.

**Definition 2** (Activation patterns). An activation pattern of a network with $N$ rank-$K$ maxout units is an assignment of a non-empty set $J_z \subseteq [K]$ to each unit $z \in [N]$. An activation pattern $J = (J_z)_{z \in [N]}$ with $\sum_{z \in [N]}(|J_z| - 1) = r$ is called an $r$-partial activation pattern. The set of all possible activation patterns is denoted $\mathcal{P}$, and the set of $r$-partial activation patterns is denoted $\mathcal{P}_r$. An activation sub-pattern is a pattern where we disregard all $J_z$ with $|J_z| = 1$. The set of all possible activation sub-patterns is denoted $\mathcal{S}$, and the set of $r$-partial activation sub-patterns is denoted $\mathcal{S}_r$.

**Definition 3** (Activation regions). Consider a network $\mathcal{N}$ with $N$ maxout units. For any parameter value $\theta$ and any activation pattern $J$, the corresponding activation region is

$$\mathcal{R}(J, \theta) := \big\{ x \in \mathbb{R}^{n_0} \mid \operatorname*{argmax}_{k \in [K]} \zeta_{z,k}(x; \theta) = J_z \ \text{ for each } z \in [N] \big\}.$$

For any $r \in \{0, \ldots, n_0\}$ we denote the union of $r$-partial activation regions by

$$\mathcal{X}_{\mathcal{N}, r}(\theta) := \bigcup_{J \in \mathcal{P}_r} \mathcal{R}(J; \theta).$$

By these definitions, we have a decomposition of the input space as a disjoint union of activation regions, $\mathbb{R}^{n_0} = \sqcup_{J \in \mathcal{P}} \mathcal{R}(J, \theta)$. See Figure 1. Next we observe that for almost every choice of $\theta$, $r$-partial activation regions are either empty or relatively open convex polyhedra of co-dimension $r$. In particular, for almost every choice of the parameters, if $r$ is larger than $n_0$, the $r$-partial activation regions are empty. Therefore, in our discussion we only need to consider $r$ up to $n_0$.

**Lemma 4** ($r$-partial activation regions are relatively open convex polyhedra). *Consider a maxout network $\mathcal{N}$. Let $r \in \{0, \ldots, n_0\}$ and $J \in \mathcal{P}_r$. Then for any $\theta$, $\mathcal{R}(J, \theta)$ is a relatively open convex polyhedron in $\mathbb{R}^{n_0}$. For almost every $\theta$, it is either empty or has co-dimension $r$.*

The proof of Lemma 4 is given in Appendix A. Next we show that for almost every choice of $\theta$, 0-partial activation regions and linear regions correspond to each other.

**Lemma 5** (Activation regions vs linear regions). *Consider a maxout network $\mathcal{N}$. The set of parameter values $\theta$ for which the represented function has the same gradient on two distinct activation regions is a null set. In particular, for almost every $\theta$, linear regions and activation regions correspond to each other.*

The proof of Lemma 5 is given in Appendix A. We note that for specific parameters, linear regions can be the union of several activation regions and can be non-convex. Such situation is more common in ReLU networks, whose units can more readily output zero, thereby hiding the activation pattern of the units in the previous layers.

To summarize the above observations, for almost every $\theta$, the 0-partial activation regions are $n_0$-dimensional open convex polyhedra which agree with the linear regions of the represented function, and for $r = 1, \ldots, n_0$ the $r$-partial activation regions are co-dimension-$r$ polyhedral pieces of the boundary between linear regions. Next we investigate the number non-empty $r$-partial activation regions and their volume within given subsets of the input space. We are concerned with their generic numbers, where we use "generic" in the standard sense, to refer to a positive Lebesgue measure event.

## 3 Numbers of regions attained with positive probability

We start with a simple upper bound.

**Lemma 6** (Simple upper bound on the number of $r$-partial activation patterns). *Let $r \in \mathbb{N}_0$. The number of $r$-partial activation patterns and sub-patterns in a network with a total of $N$ rank-$K$ maxout units are upper bounded by $|\mathcal{P}_r| \leq \binom{rK}{2r}\binom{N}{r}K^{N-r}$ and $|\mathcal{S}_r| \leq \binom{rK}{2r}\binom{N}{r}$ respectively.*

The upper bound has asymptotic order $O(N^r K^{N+r})$ in $K$ and $N$. The proof of Lemma 6 is given in Appendix A, where we also provide an exact but unhandy counting formula.

By definition, the number of $r$-partial activation patterns is a trivial upper bound on the number of non-empty $r$-partial activation regions for any choice of parameters. Depending on the network architecture, this bound may not be attainable for any choice of the parameters. Montúfar et al. (2021, Theorems 3.7 and 3.12) obtained bounds for the maximum number of linear regions. For a shallow network with $n_0$ inputs and a single layer of $n_1$ rank-$K$ maxout units it has order $\Theta((n_1 K)^{n_0})$ in $K$ and $n_1$, and for a deep network with $n_0$ inputs and $L$ layers of $n_1, \ldots, n_L$ rank-$K$ maxout units it has order $\Theta(\prod_{l=1}^{L}(n_l K)^{n_0})$ in $K$ and $n_1, \ldots, n_L$. Hence the maximum number of non-empty activation regions can be very large, especially for deep networks.

Intuitively, linear regions have a non-zero volume and cannot 'disappear' under small perturbations of parameters. This raises the question about which numbers of linear regions are attained with positive probability, i.e. over positive Lebesgue measure subsets of parameter values. Figure 1 shows that the number of linear regions of a maxout network is a very intricate function of the parameter values.

For a network with $n_0$ inputs and a single layer of $n_1$ ReLUs, the maximum number of linear regions is $\sum_{j=0}^{n_0}\binom{n_1}{j}$, and is attained for almost all parameter values. This is a consequence of the generic behavior of hyperplane arrangements (see Buck, 1943; Zaslavsky, 1975; Montúfar et al., 2014). In contrast, shallow maxout networks can attain different numbers of linear regions with positive probability. The intuitive reason is that the nonlinear locus of maxout units is described not only by linear equations $\langle w_i, x \rangle + b_i = \langle w_j, x \rangle + b_j$ but also linear inequalities $\langle w_i, x \rangle + b_i \geq \langle w_k, x \rangle + b_k$. See Figure 1 for an example. We obtain the following result.

**Theorem 7** (Numbers of linear regions).

- *Consider a rank-$K$ maxout unit with $n_0$ inputs. This corresponds to a network with an input layer of size $n_0$ and single maxout layer with a single maxout unit. For each $1 \leq k \leq K$, there is a set of parameter values for which the number of linear regions is $k$. For $\min\{K, n_0 + 1\} \leq k \leq K$, the corresponding set has positive measure, and else it is a null set.*

- *Consider a layer of $n_1$ rank-$K$ maxout units with $n_0$ inputs. This corresponds to a network with a single maxout layer, $L = 1$, and $n_L = n_1$. For each choice of $1 \leq k_1, \ldots, k_{n_1} \leq K$, there are parameters for which the number of linear regions is $\sum_{j=0}^{n_0}\sum_{S \in \binom{[n_1]}{j}}\prod_{i \in S}(k_i - 1)$. For*

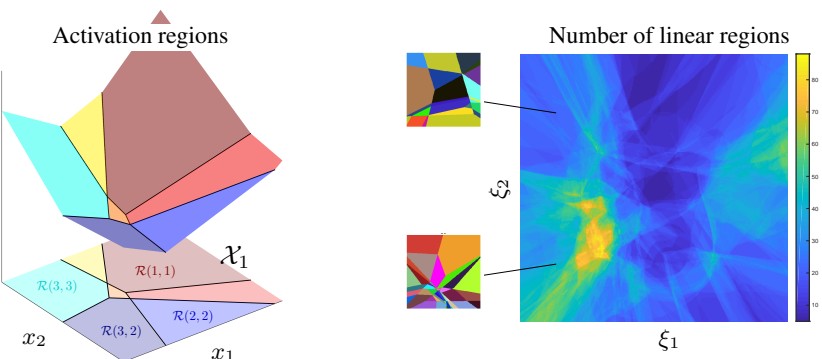

Figure 1: Left: Shown is a piecewise linear function $\mathbb{R}^2 \to \mathbb{R}$ represented by a network with a layer of two rank-3 maxout units for a choice of the parameters. The input space is subdivided into activation regions $\mathcal{R}(J; \theta)$ with linear regions separated by $\mathcal{X}_1(\theta)$. Right: Shown is the number of linear regions of a 3 layer maxout network over a portion of the input space as a function of a 2D affine subspace of parameter values $\theta(\xi_1, \xi_2)$. Shown are also two examples of the input-space subdivisions of functions represented by the network for different parameter values. More details about this figure are given in Appendix K. As the figure illustrates, the function taking parameters to number of regions is rather intricate. In this work we characterize values attained with positive probability and upper bound the expected value given a parameter distribution.

$\min\{K, n_0 + 1\} \leq k_1, \ldots, k_{n_1} \leq K$, *the corresponding set has positive measure. Here $S \in \binom{[n_1]}{j}$ means that $S$ is a subset of $[n_1] := \{1, \ldots, n_1\}$ of cardinality $|S| = j$.*

- *Consider a network with $n_0$ inputs and $L$ layers of $n_1, \ldots, n_L$ rank-$K$ maxout units, $K \geq 2$, $\frac{n_l}{n_0}$ even. Then, for each choice of $1 \leq k_{li} \leq K$, $i = 1, \ldots, n_0$, $l = 1, \ldots, L$, there are parameters for which the number of linear regions is $\prod_{l=1}^{L} \prod_{i=1}^{n_0} (\frac{n_l}{n_0}(k_{li} - 1) + 1)$. There is a positive measure subset of parameters for which the latter is the number of linear regions over $(0,1)^{n_0}$.*

The proof is provided in Appendix B. The result shows that maxout networks have a multitude of positive measure subsets of parameters over which they attain widely different numbers of linear regions. In the last statement of the theorem we consider inputs from a cube, but qualitatively similar statements can be formulated for the entire input space.

There are specific parameter values for which the network represents functions with very few linear regions (e.g., setting the weights and biases of the last layer to zero). However, the smallest numbers of regions are only attained over null sets of parameters:

**Theorem 8** (Generic lower bound on the number of linear regions). *Consider a rank-$K$ maxout network, $K \geq 2$, with $n_0$ inputs, $n_1$ units in the first layer, and any number of additional nonzero width layers. Then, for almost every choice of the parameters, the number of linear regions is at least $\sum_{j=0}^{n_0} \binom{n_1}{j}$ and the number of bounded linear regions is at least $\binom{n_1-1}{n_0}$.*

This lower bound has asymptotic order $\Omega(n_1^{n_0})$ in $K$ and $n_1, \ldots, n_L$. The proof is provided in Appendix B. To our knowledge, this is the first non-trivial probability-one lower bound for a maxout network. Note that this statement does not apply to ReLU networks unless they have a single layer of ReLUs. In the next section we investigate the expected number of activation regions for given probability distributions over the parameter space.

## 4  Expected number and volume of activation regions

For the expected number of activation regions we obtain the following upper bound, which corresponds to a maxout version of (Hanin and Rolnick, 2019b, Theorem 10).

**Theorem 9** (Upper bound on the expected number of partial activation regions). *Let $\mathcal{N}$ be a fully-connected feed-forward maxout network with $n_0$ inputs and a total of $N$ rank $K$ maxout units. Suppose we have a probability distribution over the parameters so that:*

1. *The distribution of all weights has a density with respect to the Lebesgue measure on $\mathbb{R}^{\#\mathrm{weights}}$.*

2. *Every collection of biases has a conditional density with respect to Lebesgue measure given the values of all other weights and biases.*

3. *There exists $C_{\mathrm{grad}} > 0$ so that for any $t \in \mathbb{N}$ and any pre-activation feature $\zeta_{z,k}$,*

$$\sup_{x \in \mathbb{R}^{n_0}} \mathbb{E}[\|\nabla\zeta_{z,k}(x)\|^t] \leq C_{\mathrm{grad}}^t.$$

4. *There exists $C_{\mathrm{bias}} > 0$ so that for any pre-activation features $\zeta_1, \ldots, \zeta_t$ from any neurons, the conditional density of their biases $\rho_{b_1,\ldots,b_t}$ given all the other weights and biases satisfies*

$$\sup_{b_1,\ldots,b_t \in \mathbb{R}} \rho_{b_1,\ldots,b_t}(b_1, \ldots, b_t) \leq C_{\mathrm{bias}}^t.$$

*Fix $r \in \{0, \ldots, n_0\}$ and let $T = 2^5 C_{\mathrm{grad}} C_{\mathrm{bias}}$. Then, there exists $\delta_0 \leq 1/(2C_{\mathrm{grad}}C_{\mathrm{bias}})$ such that for all cubes $C \subseteq \mathbb{R}^{n_0}$ with side length $\delta > \delta_0$ we have*

$$\frac{\mathbb{E}[\# \ r\text{-partial activation regions of } \mathcal{N} \text{ in } C]}{\mathrm{vol}(C)} \leq \begin{cases} \binom{rK}{2r}\binom{N}{r}K^{N-r}, & N \leq n_0 \\ \frac{(TKN)^{n_0}\binom{n_0 K}{2n_0}}{(2K)^r n_0!}, & N \geq n_0 \end{cases}.$$

*Here the expectation is taken with respect to the distribution of weights and biases in $\mathcal{N}$. Of particular interest is the case $r = 0$, which corresponds to the number of linear regions.*

The proof of Theorem 9 is given in Appendix E. The upper bound has asymptotic order $O(N^{n_0}K^{3n_0-r})$ in $K$ and $N$, which is polynomial. In contrast, Montúfar et al. (2021) shows that the maximum number of linear regions of a deep network of width $n$ is $\Theta((nK)^{\frac{n_0}{n}N})$, which for constant width is exponential in $N$; see Appendix B. We present an analogue of Theorem 9 for networks without biases in Appendix G.

When the rank is $K = 2$, the formula coincides with the result obtained previously by Hanin and Rolnick (2019b, Theorem 10) for ReLU networks, up to a factor $K^r$. For some settings, we expect that the result can be further improved. For instance, for iid Gaussian weights and biases, one can show that the expected number of regions of a rank $K$ maxout unit grows only like $\log K$, as we discuss in Appendix C.

We note that the constants $C_{\mathrm{bias}}$ and $C_{\mathrm{grad}}$ only need to be evaluated over the inputs in the region $C$. Intuitively, the bound on the conditional density of bias values corresponds to a bound on the density of non-linear locations over the input. The bound on the expected gradient norm of the pre-activation features is determined by the distribution of weights. We provide more details in Appendix F.

For the expected volume of the $r$-dimensional part of the non-linear locus we obtain the following upper bound, which corresponds to a maxout version of (Hanin and Rolnick, 2019a, Corollary 7).

**Theorem 10** (Upper bound on the expected volume of the non-linear locus)**.** *Consider a bounded measurable set $S \subset \mathbb{R}^{n_0}$ and the settings of Theorem 9 with constants $C_{grad}$ and $C_{bias}$ evaluated over $S$. Then, for any $r \in \{1, \ldots, n_0\}$,*

$$\frac{\mathbb{E}[\mathrm{vol}_{n_0-r}(\mathcal{X}_{\mathcal{N},r} \cap S)]}{\mathrm{vol}_{n_0}(S)} \leq (2C_{grad}C_{bias})^r \binom{rK}{2r}\binom{N}{r}.$$

The proof of Theorem 10 is given in Appendix D. When the rank is $K = 2$, the formula coincides with the result obtained previously by Hanin and Rolnick (2019a, Corollary 7) for ReLU networks. A table comparing the results for maxout and ReLU networks is given in Appendix E.

## 5    Expected number of pieces and volume of the decision boundary

In the case of classification problems, we are primarily interested in the decision boundary, rather than the overall function. We define an $M$-class classifier by appending an argmax gate to a network with $M$ outputs. The decision boundary is then a union of certain $r$-partial activation regions for the network with a maxout unit as the output layer. For simplicity, here we present the results for

the $n_0 - 1$-dimensional regions, which we call 'pieces', and present the results for arbitrary values of $r$ in Appendix H. The number of pieces of the decision boundary is at most equal to the number of activation regions in the original network times $\binom{M}{2}$. A related statement appeared in Alfarra et al. (2020). For specific choices of the network parameters, the decision boundary does intersects most activation regions and can have as many as $\Omega(M^2 \prod_{l=1}^{L}(n_l K)^{n_0})$ pieces (see Appendix H). However, in general this upper bound can be improved. For the expected number of pieces and volume of the decision boundary we obtain the following results. We write $\mathcal{X}_{\text{DB}}$ for the decision boundary, and $\mathcal{X}_{\text{DB},r}$ for the union of $r$-partial activation regions which include equations from the decision boundary (generically these are the co-dimension-$r$ pieces of the decision boundary).

**Theorem 11** (Upper bound on the expected number of linear pieces of the decision boundary)**.** *Let $\mathcal{N}$ be a fully-connected feedforward maxout network, with $n_0$ inputs, a total of $N$ rank-$K$ maxout units, and $M$ linear output units used for multi-class classification. Under the assumptions of Theorem 9, there exists $\delta_0 \leq 1/(2C_{\text{grad}}C_{\text{bias}})$ such that for all cubes $C \subseteq \mathbb{R}^{n_0}$ with side length $\delta > \delta_0$,*

$$\frac{\mathbb{E}\left[\substack{\text{\# linear pieces in the} \\ \text{decision boundary of } \mathcal{N} \text{ in } C}\right]}{\text{vol}(C)} \leq \begin{cases} \binom{M}{2}K^N, & N \leq n_0 \\ \frac{(2^4 C_{\text{grad}}C_{\text{bias}})^{n_0}(2KN)^{n_0-1}}{(n_0-1)!}\binom{M}{2}\binom{K(n_0-1)}{2(n_0-1)}, & N \geq n_0 \end{cases} .$$

*Here the expectation is taken with respect to the distribution of weights and biases in $\mathcal{N}$.*

For binary classification, $M = 2$, this bound has asymptotic order $O((K^3 N)^{n_0-1})$ in $K$ and $N$. For the expected volume we have the following.

**Theorem 12** (Upper bound on the volume of the $(n_0 - r)$-skeleton of the decision boundary)**.** *Consider a bounded measurable set $S \subset \mathbb{R}^{n_0}$. Consider the notation and assumptions of Theorem 9, whereby the constants $C_{\text{grad}}$ and $C_{\text{bias}}$ are over $S$. Then, for any $r \in \{1, \dots, n_0\}$ we have*

$$\frac{\mathbb{E}[\text{vol}_{n_0-r}(\mathcal{X}_{\text{DB},r} \cap S)]}{\text{vol}_{n_0}(S)} \leq (2C_{\text{grad}}C_{\text{bias}})^r \sum_{i=1}^{\min\{M-1,r\}} \binom{M}{i+1}\binom{K(r-i)}{2(r-i)}\binom{N}{r-i}.$$

Moreover, the expected distance to the decision boundary can be bounded as follows.

**Corollary 13** (Distance to the decision boundary)**.** *Suppose $\mathcal{N}$ is as in Theorem 9. For any compact set $S \subset \mathbb{R}^{n_0}$ let $x$ be a uniform point in $S$. There exists $c > 0$ independent of $S$ so that*

$$\mathbb{E}[\text{distance}(x, \mathcal{X}_{\text{DB}})] \geq \frac{c}{2C_{\text{grad}}C_{\text{bias}}M^{m+1}m},$$

*where $m := \min\{M - 1, n_0\}$.*

The proofs are presented in Appendix H, where we also extend Theorem 11 to address the expected number of co-dimension-$r$ pieces of the decision boundary. A corresponding result applies for the case of ReLU networks (see details in Appendix H).

## 6 Experiments

In the experiments we used fully-connected networks. We describe the network architecture in terms of the depth and total number of units, with units evenly distributed across the layers with larger lower layers if needed. For instance, a network of depth 3 with 110 units has 3 hidden layers of widths $37, 37, 36$. Details and additional experiments are presented in Appendix K. The computer implementation of the key functions is available on GitHub at `https://github.com/hanna-tseran/maxout_complexity`.

**Initialization procedures** We consider several initialization procedures detailed in Appendix J: 1) ReLU-He initializes the parameters as iid samples from the distribution proposed by He et al. (2015) for ReLUs. 2) Maxout-He follows a similar reasoning to normalize the expected norm of activation vectors across layers, but for the case of maxout networks. The weight distribution has standard deviation depending on $K$ and the assumed type of data distribution, as shown in Table 1. 3) "Sphere" ensures each unit has the maximum number of regions. 4) "Many regions" ensures each layer has the maximum number of regions.

Table 1: Standard deviation of the weight distribution for maxout-He initialization.

| Maxout rank | Standard deviation |
|---|---|
| 2 | $\sqrt{1/n_l}$ |
| 3 | $\sqrt{2\pi/((\sqrt{3}+2\pi)n_l)}$ |
| 4 | $\sqrt{\pi/((\sqrt{3}+\pi)n_l)}$ |
| 5 | $\sqrt{0.5555/n_l}$ |
| ReLU | $\sqrt{2/n_l}$ |

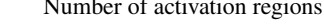

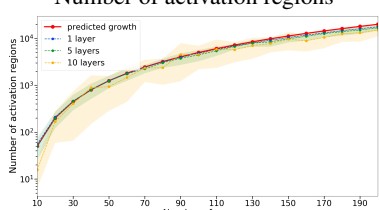
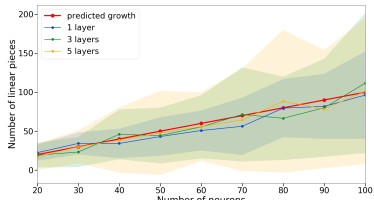

Figure 2: Shown are means and stds for 30 maxout-He normal initializations for networks with $K = 2$ and $n_0 = 2$. Left: Comparison of the theoretically predicted growth $O(N^{n_0}/n_0!)$ and the experimentally obtained number of regions for networks with different architectures. Right: Comparison of the theoretically predicted growth $O(N)$ and the experimentally obtained number of linear pieces of the decision boundary for networks with different architectures.

**Algorithm for counting activation regions**  Several approaches for counting linear regions of ReLU networks have been considered (e.g., Serra et al., 2018; Hanin and Rolnick, 2019b; Serra and Rama-lingam, 2020; Xiong et al., 2020). For maxout networks we count the activation regions and pieces of the decision boundary by iterative addition of linear inequality constraints and feasibility verification using linear programming. Pseudocode and complexity analysis are provided in Appendix I.

**Number of regions and decision boundary for different networks**  Figure 2 shows a close agreement, up to constants, of the theoretical upper bounds on the expected number of activation regions and on the expected number of linear pieces of the decision boundaries with the empirically obtained values for different networks. Further comparisons with constants and different values of $K$ are provided in Appendix K. Figure 3 shows that for common parameter distributions, the growth of the expected number of activation regions is more significantly determined by the total number of neurons than by the network's depth. In fact, we observe that for high rank units and certain types of distributions, deeper networks may have fewer activation regions. We attribute this to the fact that higher rank units tend to have smaller images (since they compute the max of more pre-activation features). Figure 4 shows how $n_0$ and $K$ affect the number of activation regions. For small input dimension, the number of regions per unit tends to be smaller than $K$. Indeed, for iid Gaussian parameters the number of regions per unit scales as $\log K$ (see Appendix C).

**Number of regions during training**  We consider the 10 class classification task with the MNIST dataset (LeCun et al., 2010) and optimization with Adam (Kingma and Ba, 2015) using different initialization strategies. Notice that for deep skinny fully connected networks the task is non-trivial. Figure 5 shows how the number of activation regions evolves during training. Shown are also the linear regions and decision boundaries over a 2D slice of the input space through 3 training data points. Figure 6 shows the training loss and accuracy curves for the different initializations. We observe that maxout networks with maxout-He, sphere, and many regions converge faster than with naive He initialization.

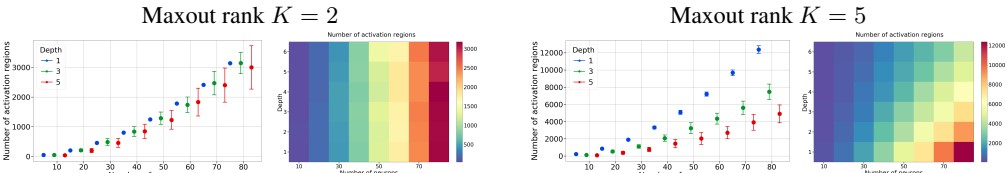

Figure 3: Effect of the depth and number of neurons on the number of activation regions at initialization for networks with $n_0 = 2$. Shown are means and stds for 30 maxout-He normal initializations.

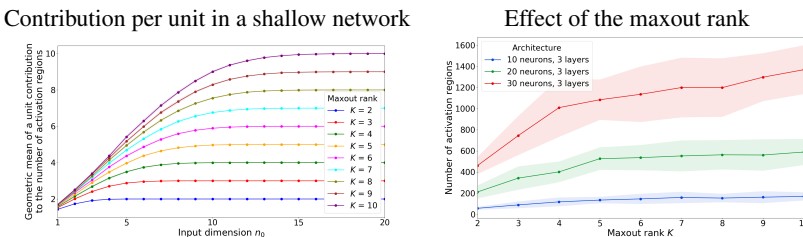

Figure 4: Left: Plotted is $\#\text{regions}^{1/N}$ for a shallow network with $N = 5$. The multiplicative contribution per unit increases with the input dimension until the trivial upper bound $K$ is reached. Right: Number of regions of 3 layer networks with $n_0 = 2$ depending on $K$. Shown are means and stds for 30 ReLU-He normal initializations.

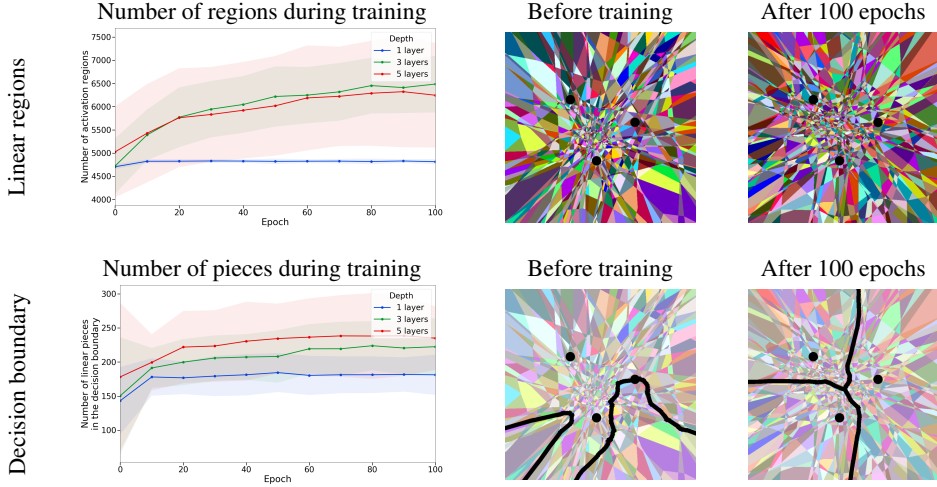

Figure 5: Evolution of the linear regions and the decision boundary during training on the MNIST dataset in a slice determined by three random points from different classes. The network had 100 maxout units of rank $K = 2$, and was initialized using maxout-He normal initialization. The right panel is for the 3 layer network. As expected, for the shallow rank-2 network, the number of regions is approximately constant. For deep networks we observe a moderate increase in the number of regions as training progresses, especially around the training data. However, the number of regions remains far from the theoretical maximum. This is consistent with previous observations for ReLU networks. There is also a slight increase in the number of linear pieces in the decision boundary, and at the end of training the decision boundary clearly separates the three reference points.

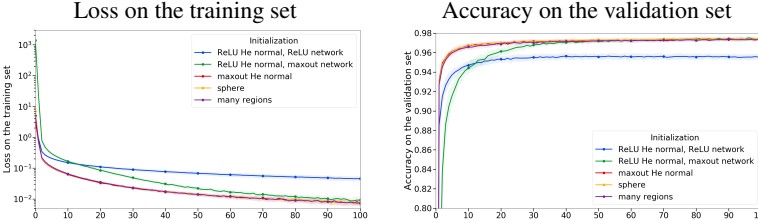

Figure 6: Comparison of training on MNIST with different initializations. All networks had 200 units, 10 layers, and maxout networks had rank $K = 5$. Shown are averages and std (barely noticeable) over 30 repetitions. The type of initialization has a significant impact on the training time of maxout networks, with maxout-He, sphere, and many regions giving better results for deep networks and larger maxout rank (more details on this in Appendix K).

## 7  Discussion

We advance a line of analysis recently proposed by Hanin and Rolnick (2019a,b), where the focus lies on the expected complexity of the functions represented by neural networks rather than worst case bounds. Whereas previous works focus on single-argument activations, our results apply to networks with multi-argument maxout units. We observe that maxout networks can assume widely different numbers of linear regions with positive probability and then computed an upper bound on the expected number of regions and volume given properties of the parameter distribution, covering the case of zero biases. Further, taking the standpoint of classification, we obtained corresponding results for the decision boundary of maxout (and ReLU) networks, along with bounds on the expected distance to the decision boundary.

Experiments show that the theoretical bounds capture the general behavior. We present algorithms for enumerating the regions of maxout networks and proposed parameter initialization strategies with two types of motivations, one to increase the number of regions, and second, to normalize the variance of the activations similar to Glorot and Bengio (2010) and He et al. (2015), but now for maxout. We observed experimentally that this can improve training in maxout networks.

**Limitations**  In our theory and experiments we have considered only fully connected networks. The analysis and implementation of other architectures for experiments with more diverse datasets are interesting extensions. By design, the results focus on parameter distributions which have a density.

**Future work**  In future work we would like to obtain a fine grained description of the distribution of activation regions over the input space depending on the parameter distribution and explore the relations to speed of convergence and implicit biases in gradient descent. Of significant interest would be an extension of the presented results to specific types of parameter distributions, including such which do not have a density or those one might obtain after training.

**Discussion of potential negative societal impacts**  This is foundational research and not tied to particular applications. To the best of our knowledge there are no direct paths to negative societal impacts.

**Funding transparency statement**  This project has received funding from the European Research Council (ERC) under the European Union's Horizon 2020 research and innovation programme (grant agreement n⁰ 757983).

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
