## Appendix

The appendix is organized as follows.

## A  Proofs related to activation patterns and activation regions

### A.1  Number of activation patterns

**Lemma 6** (Simple upper bound on the number of $r$-partial activation patterns)**.** *Let $r \in \mathbb{N}_0$. The number of $r$-partial activation patterns and sub-patterns in a network with a total of $N$ rank-$K$ maxout units are upper bounded by $|\mathcal{P}_r| \leq \binom{rK}{2r}\binom{N}{r}K^{N-r}$ and $|\mathcal{S}_r| \leq \binom{rK}{2r}\binom{N}{r}$ respectively.*

*Proof of Lemma 6.* To get an $r$-partial activation pattern one needs at most $r$ neurons. The number of ways to choose them is $\binom{N}{r}$. The number of ways to choose a pre-activation feature that attains a maximum in the rest of neurons is $K^{N-r}$. The $r$ chosen neurons have in total $rK$ pre-activation features. Out of them, we need to choose $r$ features that attain maximum, and $r$ additional features to construct the pre-activation pattern, so $2r$ features in total. We ignore the restriction that there needs to be at least one feature from each neuron, which gives us an upper-bound $r\binom{K}{2r}$. Notice that this way we also count $r$-partial patterns that require less than $r$ neurons. Combining everything, we get the desired result. For the sub-patters, we simply ignore the term $K^{n-r}$. $\qquad\square$

We will use the above upper bound in our calculations due to its simplicity. For completeness, we note that the exact number of partial activation patterns can be given as follows.

**Proposition 14** (Number of $r$-partial activation patterns)**.** *For a network with a total of $N$ rank-$K$ maxout units the number of distinct $r$-partial activation patterns is*

$$|\mathcal{P}_r| = \sum_{\substack{(N_0,\ldots,N_{K-1})\in\mathbb{N}_0^K:\\ \sum_{j=0}^{K-1}N_j=N, \sum_{j=0}^{K-1}jN_j=r}} \binom{N}{N_0,\ldots,N_{K-1}}\prod_{j=0}^{K-1}\binom{K}{1+j}^{N_j}.$$

*If $K = 2$ then the summation index takes only one value $(N_0, N_1) = (N - r, r)$ and the expression simplifies to $\binom{N}{N-r}2^{N-r}$.*

*Proof.* We have $N$ neurons. For a given activation pattern, for $j = 0,\ldots, K - 1$, denote $N_j$ the number of neurons with $(1 + j)$ pre-activation features attaining the maximum. Since every neuron has indecision in the range $0,\ldots, K - 1$, we have $\sum_{j=0}^{K-1} N_j = N$. The $r$-partial activation patterns are precisely those for which $\sum_j jN_j = r$. The number of distinct ways in which we can partition the set of $N$ neurons into $K$ sets of cardinalities $N_0,\ldots, N_{K-1}$ is precisely $\binom{N}{N_0,\ldots,N_{K-1}}$. For each $j$, the number of ways in which a given neuron can have $(1 + j)$ pre-activation features attaining the maximum is $\binom{K}{1+j}$. $\qquad\square$

## A.2 Generic correspondence between activation regions and linear regions

For a fixed activation pattern $J$, a *computation path* $\gamma$ is a path in the computation graph of the network $\mathcal{N}$ that goes from input to the output through one of the units in each layer, where $\gamma = (\gamma_0, \gamma_1, \ldots, \gamma_L)$, $\gamma_l \in [n_l] \times [K]$ specifies a unit and a corresponding pre-activation feature in layer $l$. For any input $x$ in the activation region $\mathcal{R}(J, \theta)$, the gradient with respect to $x$ can be expressed through the computation paths as

$$\nabla \mathcal{N}(x, \theta) = W_x^{(L+1)} W_x^{(L)} \cdots W_x^{(1)}, \qquad \frac{\partial}{\partial x_i} \mathcal{N}(x, \theta) = \sum_{\substack{\text{paths } \gamma \\ \text{starting at } i}} \prod_{l=1}^{L+1} w_\gamma^{(l)},$$

where in $W_x^{(l)} \in \mathbb{R}^{n_l \times n_{l-1}}$ is a piecewise constant matrix valued function of the input $x$ with rows corresponding to the pre-activation features that attain the maximum according to the pattern $J$, and $w_\gamma^{(l)} \in \mathbb{R}$ are corresponding weights on the edge of $\gamma$ between the layer $(l-1)$ and $l$, again depending on $J$. For a simple example of when one linear region is a union of several activation regions in a maxout network, consider a network with one of the weights in the single linear output unit set to zero. Such a situation can happen, for instance, at initialization, though with probability $0$. Then, switching between the maximums in the unit in the previous layer to which this weight connects will not be visible when we compute the gradient, and several activation regions created by the transitions between maximums in this unit will become a part of the same linear region.

**Lemma 5** (Activation regions vs linear regions). *Consider a maxout network $\mathcal{N}$. The set of parameter values $\theta$ for which the represented function has the same gradient on two distinct activation regions is a null set. In particular, for almost every $\theta$, linear regions and activation regions correspond to each other.*

*Proof of the Lemma 5.* Consider two different non-empty activation regions corresponding to activation patterns $J_1$ and $J_2$ for which $\nabla \mathcal{N}(x; \theta)$ has the same value. This means that $n_0$ equations of the form

$$\sum_{\text{paths } \gamma \in \Gamma_{1,i}} \prod_{l=1}^{L+1} w_\gamma^{(l)} = \sum_{\text{paths } \gamma \in \Gamma_{2,i}} \prod_{l=1}^{L+1} w_\gamma^{(l)}$$

are satisfied, where $\Gamma_{1,i}, \Gamma_{2,i}$ are collections of paths starting at $i$ corresponding to the activation patterns $J_1$ and $J_2$ respectively. For different values of $i$ the sets of paths differ only at the input layer.

Based on this equation, there exists $c_{\gamma,i} \in \{\pm 1\}$ and a non-empty collection of paths $\Gamma_i$ (the symmetric difference of $\Gamma_{1,i}$ and $\Gamma_{2,i}$) so that

$$\sum_{\text{paths } \gamma \in \Gamma_i} c_{\gamma,i} \prod_{l=1}^{L+1} w_\gamma^{(l)} = 0.$$

This is a polynomial equation in the weights of the network. Each monomial occurs either with coefficient $1$ or $-1$. In particular, this polynomial is not identically zero. The zero set of a polynomial is of measure zero on $\mathbb{R}^{\#\text{weights}}$ unless it is identically zero, see e.g. Caron and Traynor (2005). We have a system of $n_0$ such equations (one for each $i$). The intersection of the solution sets is again a set of measure zero. The total number of pairs of activation regions is finite, upper bounded by $\binom{K^N}{2}$. A countable union of measure zero sets is of measure zero, thus the set of weights for which two activation regions have the same gradient values has measure zero with respect to the Lebesgue measure on $\mathbb{R}^{\#\text{weights}}$. $\qquad\square$

## A.3 Partial activation regions

Now we introduce several objects that are needed to discuss $r$-partial activation regions.

**Definition 15.** Fix a value $\theta$ of the trainable parameters. For a neuron $z$ in $\mathcal{N}$ and a set $J_z \subseteq [K]$, the $J_z$-**activation region** of a unit $z$ is

$$\mathcal{H}(J_z; \theta) := \{x_0 \in \mathbb{R}^{n_0} \mid \operatorname*{argmax}_{k \in [K]} \zeta_{z,k}(x_{l(z)-1}; \theta) = J_z\}.$$

More generally, for a set of neurons $\mathcal{Z} = \{z\}$ and a corresponding list of sets $J_{\mathcal{Z}} = (J_z)_{z \in \mathcal{Z}}$, the corresponding $J_{\mathcal{Z}}$-activation region is

$$\mathcal{H}(J_{\mathcal{Z}}; \theta) := \bigcap_{z \in \mathcal{Z}} \mathcal{H}(J_z; \theta). \tag{1}$$

If we specify an activation pattern for every neuron, $J_{[N]}$, so that $\mathcal{Z} = [N]$, then we write

$$\mathcal{R}(J_{[N]}; \theta) = \mathcal{H}(J_{[N]}; \theta).$$

Recall that an activation pattern $J_{[N]}$ with with the property that $\sum_z (|J_z| - 1) = r$ is called an $r$-partial activation pattern. To distinguish such patterns, we denote them by $J^r \in \mathcal{P}_r$. The union of all corresponding activation regions is denoted

$$\mathcal{X}_{\mathcal{N}, r}(\theta) = \bigcup_{J^r \in \mathcal{P}_r} \mathcal{R}(J^r; \theta).$$

**Lemma 4** ($r$-partial activation regions are relatively open convex polyhedra)**.** *Consider a maxout network $\mathcal{N}$. Let $r \in \{0, \ldots, n_0\}$ and $J \in \mathcal{P}_r$. Then for any $\theta$, $\mathcal{R}(J, \theta)$ is a relatively open convex polyhedron in $\mathbb{R}^{n_0}$. For almost every $\theta$, it is either empty or has co-dimension $r$.*

*Proof of Lemma 4.* Fix an $r$-partial activation pattern $J^r \in \mathcal{P}_r$. Over the activation region $\mathcal{R}(J; \theta)$, the $k$-th pre-activation feature of each neuron $z$ is a linear function of the input to the network, namely

$$w_{z,k}^* \cdot x + b_{z,k}^* = w_{z,k}^{(l(z))}(w^{(l(z)-1)} \cdots (w^{(1)} \cdot x + b^{(1)}) \cdots + b^{(l(z)-1)}) + b_{z,k}^{(l(z))},$$

where $w_{z,k}^*$ and $b_{z,k}^*$, $k \in [K]$ denote the weights and biases of this linear function, which depend on the weights and biases and activation values of the units up to unit $z$. For each $z$ specify a fixed element $j_0 \in J_z$. The activation region can be written as

$$\bigcap_{z \in [N]} \Big\{ x \in \mathbb{R}^{n_0} \mid w_{z,j_0}^* \cdot x + b_{z,j_0}^* = w_{z,j}^* \cdot x + b_{z,j}^*, \quad \forall j \in J_z \setminus \{j_0\};$$

$$w_{z,j_0}^* \cdot x + b_{z,j_0}^* > w_{z,i}^* \cdot x + b_{z,i}^*, \quad \forall i \in [K] \setminus J_z \Big\}.$$

This means that an $r$-partial activation region is determined by a set of strict linear inequalities and $r$ linear equations. The equations are represented by vectors $v_{z,j} = (w_{z,j_0}^*, b_{z_{j_0}}^*) - (w_{z,j}^*, b_{z,j}^*)$ for all $j \in J_z \setminus \{j_0\}$ for all $z$ for which $|J_z| > 1$. For generic parameters these equations are linearly independent. Indeed, the vectors being linearly dependent means that there is a matrix $V^\top V$, where $V$ has rows $v_{z,j}$, with vanishing determinant. By similar arguments as in the proof of Lemma 5, the set of parameters solving a polynomial system has measure zero. Hence, for generic choices of parameters, the $r$ linear equations are independent and the polyhedron will have a co-dimension $r$ (or otherwise be empty). $\qquad\square$

The same result can be obtained for $r$-partial activation regions of ReLU networks since ReLU activation regions can be similarly written as a system of linear equations and inequalities.

We can make a statement about the shape of $r$-partial activation regions of maxout networks. Recall that a *convex polyhedron* is the closure of the solution set to finite system of linear inequalities. If it is bounded, it is called a convex polytope. The dimension of a polyhedron is the dimension of the smallest affine space containing it.

The next statement follows immediately from Lemma 4.

**Lemma 16** ($\mathcal{X}_{\mathcal{N}, r}$ consists of $(n_0 - r)$-dimensional pieces)**.** *With probability $1$ with respect to the distribution of the network parameters $\theta$, for any $x \in \mathcal{X}_{\mathcal{N}, r}$ there exists $\varepsilon > 0$ (depending on $x$ and $\theta$) s.t. $\mathcal{X}_{\mathcal{N}, r}$ intersected with the $\varepsilon$ ball $B_\varepsilon(x)$ is equal to the intersection of this ball with an $(n_0 - r)$-dimensional affine subspace of $\mathbb{R}^{n_0}$.*

**Corollary 17** ($r$-partial activation regions are relatively open convex polyhedra)**.** *Recall that an an $r$-partial activation sub-pattern $\hat{J} \in \mathcal{S}_r$ is a list $\hat{J} = (J_z)_{z \in Z}$ of sets $J_z \subseteq [K]$, $z \in Z \subseteq [N]$ with $|J_z| > 1$ and $\sum_{z \in Z}(|J_z| - 1) = r$. For almost all choices of the parameter (i.e., except for a null set with respect to the Lebesgue measure),*

$$\mathrm{vol}_{n_0 - r}\left(\mathcal{X}_{\mathcal{N}, r}(\theta)\right) = \sum_{\hat{J} \in \mathcal{S}_r} \mathrm{vol}_{n_0 - r}(\mathcal{H}(\hat{J}; \theta)).$$

*Proof of Corollary 17.* Given $\hat{J} \in \mathcal{S}_r$, we denote $Z \subseteq [N]$ the corresponding list of neurons. Using the notion of indecision loci from Definition 15, we can re-write $\mathcal{X}_{\mathcal{N},r}(\theta)$ as

$$\mathcal{X}_{\mathcal{N},r}(\theta) = \bigcup_{J \in \mathcal{P}_r} \mathcal{R}(J;\theta) = \bigcup_{J \in \mathcal{P}_r} \mathcal{H}(J;\theta) = \bigcup_{J \in \mathcal{P}_r} \bigcap_{z \in [N]} \mathcal{H}(J_z;\theta)$$

$$= \bigcup_{J \in \mathcal{P}_r} \left[ \bigcap_{z \in Z} \mathcal{H}(J_z;\theta) \cap \bigcap_{z \in [N] \setminus Z} \mathcal{H}(J_z;\theta) \right]$$

$$= \bigcup_{\hat{J} \in \mathcal{S}_r} \left[ \bigcap_{z \in Z} \mathcal{H}(J_z;\theta) \cap \bigcup_{J_z \in [K], z \in [N] \setminus Z} \bigcap_{z \in [N] \setminus Z} \mathcal{H}(J_z;\theta) \right]$$

$$= \bigcup_{\hat{J} \in \mathcal{S}_r} \left[ \bigcap_{z \in Z} \mathcal{H}(J_z;\theta) \cap \bigcap_{z \notin Z} \bigcup_{k \in [K]} \mathcal{H}(J_z = \{k\};\theta) \right].$$

Therefore,

$$\mathrm{vol}_{n_0 - r}\left( \mathcal{X}_{\mathcal{N},r}(\theta) \right) = \sum_{\hat{J} \in \mathcal{S}_r} \mathrm{vol}_{n_0 - r} \left( \bigcap_{z \in Z} \mathcal{H}(J_z;\theta) \cap \bigcap_{z \notin Z} \bigcup_{k \in [K]} \mathcal{H}(J_z = \{k\};\theta) \right).$$

Notice that $\left( \bigcap_{z \notin Z} \bigcup_{k \in [K]} \mathcal{H}(J_z = \{k\};\theta) \right)^c$ is a zero measure set in $\mathcal{X}_{\mathcal{N},r}(\theta)$, because over that set, by Lemma 16 the co-dimension of the corresponding activation regions is larger than $r$. Therefore, for any given $\hat{J} = (J_z)_{z \in Z} \in \mathcal{S}_r$,

$$\mathrm{vol}_{n_0 - r} \left( \bigcap_{z \in Z} \mathcal{H}(J_z;\theta) \cap \bigcap_{z \notin Z} \bigcup_{k \in [K]} \mathcal{H}(J_z = \{k\};\theta) \right) = \mathrm{vol}_{n_0 - r} \left( \bigcap_{z \in Z} \mathcal{H}(J_z;\theta) \right).$$

This completes the proof. $\qquad\square$

## B  Proofs related to the generic numbers of regions

### B.1  Number of regions and Newton polytopes

We start with the observation that the linear regions of a maxout unit correspond to the upper vertices of a polytope constructed from its parameters.

**Definition 18.** Consider a function of the form $f \colon \mathbb{R}^n \to \mathbb{R}$; $f(x) = \max\{w_j \cdot x + b_j\}$, where $w_j \in \mathbb{R}^n$ and $b_j \in \mathbb{R}$, $j = 1, \ldots, M$. The *lifted Newton polytope* of $f$ is defined as $P_f := \mathrm{conv}\{(w_j, b_j) \in \mathbb{R}^{n+1} \colon j = 1, \ldots, M\}$.

**Definition 19.** Let $P$ be a polytope in $\mathbb{R}^{n+1}$ and let $F$ be a face of $P$. An outer normal vector of $F$ is a vector $v \in \mathbb{R}^{n+1}$ with $\langle v, p - q \rangle > 0$ for all $p \in F$, $q \in P \setminus F$ and $\langle v, p - q \rangle = 0$ for all $p, q \in F$. The face $F$ is an *upper face* of $P$ if it has an outer normal vector $v$ whose last coordinate is positive, $v_{n+1} > 0$. It is a *strict upper face* if each of its outer normal vectors has a positive last coordinate.

The Newton polytope is a fundamental object in the study of polynomials. The naming in the context of piecewise linear functions stems from the fact that piecewise linear functions can be regarded as differences of so-called tropical polynomials. The connections between such polynomials and neural networks with piecewise linear activation functions have been discussed in several recent works (Zhang et al., 2018; Charisopoulos and Maragos, 2018; Alfarra et al., 2020). For details on tropical geometry, see (Maclagan and Sturmfels, 2015; Joswig, 2022). Although in the context of (tropical) polynomials the coefficients are integers, such a restriction is not needed in our discussion.

A convex analysis interpretation of the Newton polytope can be given as follows. Consider a piecewise linear convex function $f \colon \mathbb{R}^n \to \mathbb{R}$; $x \mapsto \max_j\{w_j \cdot x + b_j\}$. Then the upper faces of its lifted Newton polytope $P_f$ correspond to the graph $\{(x^*, -f^*(x^*)) \colon x^* \in \mathbb{R}^n \cap \mathrm{dom}(f^*)\}$ of the negated

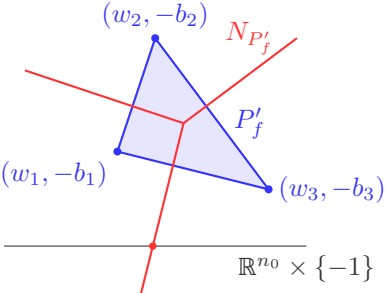

Figure 7: The linear regions of a function $f(x) = \max_j\{\langle w_j, x\rangle + b_j\}$ correspond to the lower vertices of the polytope $P'_f = \mathrm{conv}_j\{(w_j, -b_j)\} \subseteq \mathbb{R}^{n_0+1}$, or, equivalently, the upper vertices of the lifted Newton polytope $P_f = \mathrm{conv}_j\{(w_j, b_j)\} \subseteq \mathbb{R}^{n_0+1}$. The linear regions of $f$ can also be described as the intersection of the normal fan $N_{P'_f}$, consisting of outer normal cones of faces of $P'_f$, with the affine space $\mathbb{R}^{n_0} \times \{-1\}$.

convex conjugate $f^*\colon \mathbb{R}^n \to \mathbb{R}$; $x^* \mapsto \sup_{x\in\mathbb{R}^n}\langle x, x^*\rangle - f(x)$, which is a convex piecewise linear function. This implies that the upper vertices of $P_f$ are the points $(w_j, b_j) \in \mathbb{R}^{n+1}$ for which $f(x) = w_j \cdot x + b_j$ over a neighborhood of inputs. Hence the upper vertices of the Newton polytope correspond to the linear regions of $f$. This relationship holds more generally for boundaries between linear regions and other lower dimensional linear features of the graph of the function. We will use the following result, which is well known in tropical geometry (see Joswig, 2022).

**Proposition 20** (Regions correspond to upper faces)**.** *The $r$-partial activation regions of a function $f(x) = \max_j\{w_j \cdot x + b_j\}$ correspond to the $r$-dimensional upper faces of its lifted Newton polytope $P_f$. Moreover, the bounded activation regions correspond to the strict upper faces of $P_f$.*

The situation is illustrated in Figure 7.

### B.2 Bounds on the maximum number of linear regions

For reference, we briefly recall results providing upper bounds on the maximum number of linear regions of maxout networks. The maximum number of regions of maxout networks was studied by Pascanu et al. (2013); Montúfar et al. (2014), showing that deep networks can represent functions with many more linear regions than any of the functions that can be represented by a shallow network with the same number of units or parameters. Serra et al. (2018) obtained an upper bound for deep maxout networks based on multiplying upper bounds for individual layers. These bounds were recently improved by Montúfar et al. (2021), who obtained the following result, here stated in a simplified form.

**Theorem 21** (Maximum number of linear regions, Montúfar et al. 2021)**.**

- *For a network with $n_0$ inputs and a single layer of $n_1$ rank-$K$ maxout units, the maximum number of linear regions is $\sum_{j=0}^{n_0}\binom{n_1}{j}(K-1)^j$.*

- *For a network with $n_0$ inputs and $L$ layers of $n_1, \ldots, n_L$ rank-$K$ maxout units, if $n \le n_0$, $\frac{n_l}{n}$ even, and $e_l = \min\{n_0, \ldots, n_{l-1}\}$, the maximum number of linear regions is lower bounded by $\prod_{l=1}^{L}(\frac{n_l}{n}(K-1)+1)^n$ and upper bounded by $\prod_{l=1}^{L}\sum_{j=0}^{e_l}\binom{n_l}{j}(K-1)^j$.*

### B.3 Numbers of regions attained over positive measure subsets of parameters

A layer of maxout units can attain several different numbers of linear regions with positive probability over the parameters. This is illustrated in Figure 8. We obtain the following result, describing numbers of linear regions that can be attained by maxout units, layers, and deep maxout networks with positive probability over the parameters.

**Theorem 7** (Numbers of linear regions)**.**

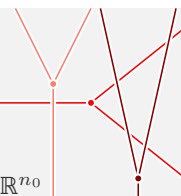

$\mathbb{R}^{n_0}$

Figure 8: A layer of maxout units of rank $K \geq 3$ attains several different numbers of linear regions with positive probability over the parameters. For a layer with two rank-3 maxout units, some neighborhoods of parameters give 6 linear regions and others 9, with nonlinear loci given by perturbations of the red-pink and red-darkred lines.

- *Consider a rank-$K$ maxout unit with $n_0$ inputs. For each $1 \leq k \leq K$, there is a set of parameter values for which the number of linear regions is $k$. For $\min\{K, n_0 + 1\} \leq k \leq K$, the corresponding set has positive measure, and else it is a null set. This corresponds to a network with an input layer of size $n_0$ and single maxout layer with a single maxout unit.*

- *For each choice of $1 \leq k_1, \ldots, k_{n_1} \leq K$, there are parameters for which the number of linear regions is $\sum_{j=0}^{n_0} \sum_{S \in \binom{[n_1]}{j}} \prod_{i \in S}(k_i - 1)$. For $\min\{K, n_0 + 1\} \leq k_1, \ldots, k_{n_1} \leq K$, the corresponding set has positive measure. Here $S \in \binom{[n_1]}{j}$ means that $S$ is a subset of $[n_1] := \{1, \ldots, n_1\}$ of cardinality $|S| = j$, and since the network has a single maxout layer, $L = 1$ and $n_L = n_1$.*

- *Consider a network with $n_0$ inputs and $L$ layers of $n_1, \ldots, n_L$ rank-$K$ maxout units, $K \geq 2$, $\frac{n_l}{n_0}$ even. Then, for each choice of $1 \leq k_{li} \leq K$, $i = 1, \ldots, n_0$, $l = 1, \ldots, L$, there are parameters for which the number of linear regions is $\prod_{l=1}^{L} \prod_{i=1}^{n_0} (\frac{n_l}{n_0}(k_{li} - 1) + 1)$. There is a positive measure subset of parameters for which the latter is the number of linear regions over $(0, 1)^{n_0}$.*

The strategy of the proof is as follows. We first show that there are parameters such that individual rank-$K$ maxout units behave as rank-$k$ maxout units, for any $1 \leq k \leq K$, and there are positive measure subsets of the parameters for which they behave as rank-$k$ maxout units, for any $n + 1 \leq k \leq K$. Further, there are positive measure subsets of the parameters of individual rank-$K$ maxout units for which, over the positive orthant $\mathbb{R}_{\geq 0}^n$, they behave as rank-$k$ maxout units, for any $1 \leq k \leq K$. Then we use a similar strategy as Montúfar et al. (2021) to construct parameters of a network with units of pre-specified ranks which attain a particular number of linear regions.

**Proposition 22.** *Consider a rank-$K$ maxout unit with $n$ inputs restricted to $\mathbb{R}_{\geq 0}^n$. For any $1 \leq k \leq K$, there is a positive measure subset of parameters for which the behaves as a rank-$k$ maxout unit. Moreover, this set can be made to contain parameters representing any desired function that can be computed by a rank-$k$ maxout unit.*

*Proof.* We need to show that for any choices of $(w_i, b_i)$, $i \in [k]$, there are generic choices of $(w_j, b_j)$, $j \in [K] \setminus [k]$, so that for each $J \subseteq [K]$ with $J \nsubseteq [k]$, the corresponding activation region $\mathcal{R}(J, \theta)$ does not intersect $\mathbb{R}_{\geq 0}^n$. Notice that, if $j \in J \setminus [k]$, then the corresponding activation region $\mathcal{R}(J, \theta)$ is contained in the arrangement consisting of hyperplanes $H_{ji} = \{x \colon (w_j - w_i) \cdot x + (b_j - b_i) = 0\}$, $i \in J \setminus \{j\}$. For each $j \in [K] \setminus [k]$, we choose $w_j = jc(-1, \ldots, -1) + \epsilon_j$, $b_j = -jc' + \epsilon_j'$ for some $c > 2\max\{\|w_i\|_\infty \colon i \in [k]\}$, $c' > 2\max\{b_i \colon i \in [k]\}$ and small $\epsilon_j \in \mathbb{R}^n$, $\epsilon_j' \in \mathbb{R}$. Then, for each $j \in [K] \setminus [k]$ and $i \in [k]$, $j < j$, the hyperplane $H_{ji}$ has a normal vector $(w_j - w_i) \in \mathbb{R}_{<0}$ and an intercept $b_j - b_i < 0$, and hence it does not intersect $\mathbb{R}_{\geq 0}^n$. $\square$

We are now ready to prove the theorem.

*Proof of Theorem 7. Single unit.* Consider a maxout unit $\max_{j \in [K]}\{w_j \cdot x + b_j\}$. To have this behave as a rank-$k$ maxout unit, $1 \leq k \leq K$, we simply set $(w_j, b_j) = (w_1, b_1 - 1)$, $j \in [K] \setminus [k]$. This is a non-generic choice of parameters. Consider now a rank-$k$ maxout unit with $n + 1 \leq k$ and generic parameters $(w_i, b_i)$, $i \in [k]$. We want to show that there are generic choices of $(w_j, b_j)$, $j \in [K] \setminus [k]$ so that $\max_{j \in [K]}\{w_j \cdot x + b_j\} = \max_{i \in [k]}\{w_i \cdot x + b_i\}$ for all $x \in \mathbb{R}^n$. In view of Proposition 20, this is equivalent to $(w_j, b_j)$, $j \in [K] \setminus [k]$ not being upper vertices of the lifted

Newton polytope $P = \text{conv}\{(w_j, b_j) \colon j \in [K]\}$. Since any generic $n + 1$ points in $\mathbb{R}^n$ are affinely independent, we have that the convex hull $\text{conv}\{w_i \in \mathbb{R}^n \colon i \in [k]\}$ has full dimension $n$. Hence, any $w_j = \frac{1}{k}\sum_{i \in [k]} w_i + \epsilon_j$ and $b_j = \min_{i \in [k]}\{b_i\} - 1 + \epsilon_j'$ with sufficiently small $\epsilon_j \in \mathbb{R}^n$, $\epsilon_j' \in \mathbb{R}$, $j \in [K] \setminus [k]$ are strictly below $\text{conv}\{(w_i, b_i) \colon i \in [k]\}$ and are not upper vertices of $P$.

*Single layer.* We use the previous item to obtain $n_1$ maxout units of ranks $k_1, \ldots, k_{n_1}$, either in the non-generic or in the generic cases. Then we apply the construction of parameters and the region counting argument from Montúfar et al. (2021, Proposition 3.4) to this layer, to obtain a function with $\sum_{j=0}^{n_0} \sum_{S \subseteq \binom{[n_1]}{j}} \prod_{i \in S}(k_i - 1)$ linear regions. For each of the units $i = 1, \ldots, n_1$, one may choose a generic vector $v_i \in \mathbb{R}^n$ and define the weights and biases of the pre-activation features as $w_{ij} = \frac{j}{k_i} v_i$ and $b_{ij} = -g(\frac{j}{k_i} + \epsilon_i)$, $j = 1, \ldots, k_i$, where $g \colon \mathbb{R} \to \mathbb{R}$ is any strictly convex function and $\epsilon_i$ is chosen generically. Then the non-linear locus of each unit consists of $k_i - 1$ parallel hyperplanes with a generic shift $\epsilon_i$, and the normal vectors $v_i$ of different units are in general position. The number of regions defined by such an arrangement of hyperplanes in $\mathbb{R}^n$ can be computed using Zaslavsky's theorem, giving the indicated result. It remains to show that, for $n_0 + 1 \le k_1, \ldots, k_{n_1}$, there are positive measure perturbations of these parameters that do change the number of regions. By the lower semi-continuity discussed in Section 3, the number of regions does not decrease for sufficiently small generic perturbations of the parameters. To show that it does not increase, we note that, by Theorem 21 this number of regions is the maximum that can be attained by a layer of $n_1$ maxout units of ranks $k_1, \ldots, k_n$.

*Deep network.* For the first statement, we use the first item to obtain maxout units of any desired ranks $1 \le k_{li} \le K$, $l = 1, \ldots, L$, $i = 1, \ldots, n_l$, and then apply the construction of parameters from Montúfar et al. (2021, Proposition 3.11) to this network, to obtain the indicated number of regions.

For the second statement, we use Proposition 22 to have the units behave as maxout units of any desired ranks over $[0, 1]^{n_0}$. For the $l$-th layer, we divide the $n_l$ units into $n_0$ blocks $x_{ij}^{(l)}$, $i = 1, \ldots, n_0$, $j = 1, \ldots, \frac{n_l}{n_0}$. For $i = 1, \ldots, n_0$, the $i$-th block consists of $\frac{n_l}{n_0}$ maxout units of rank $k_{li}$. We can choose the weights and biases so that over $[0, 1]^{n_0}$, the nonlinear locus of the $i$-th block $(x_{i,1}^{(1)}, \ldots, x_{i,\frac{n_1}{n_0}}^{(1)})$ consists of $\frac{n_l}{n_0}(k_{li} - 1)$ parallel hyperplanes with normal $e_i$, and the alternating sum $\sum_{j=1}^{n_l/n_0}(-1)^j x_{ij}^{(l)}$ is a zig-zag function along the direction $e_i$ which maps $(0, 1)^{n_0}$ to $(0, 1)$, and maps any point in $\mathbb{R}^{n_0} \setminus [0, 1]^{n_0}$ to a point in $\mathbb{R} \setminus [0, 1]$. In this way, the $l$-th layer, followed by a linear layer $\mathbb{R}^{n_l} \to \mathbb{R}^{n_0}$, maps $(0, 1)^{n_0}$ onto $(0, 1)^{n_0}$ in a $\prod_{i=1}^{n_0}(\frac{n_l}{n_0}(k_{li} - 1) + 1)$ to one manner. Sufficiently small perturbations of the parameters do not affect this general behavior. The composition of $L$ such layers gives the desired number of regions over $(0, 1)^{n_0}$. $\qquad\square$

### B.4 Minimum number of activation regions

One can easily construct parameters so that the represented function is identically zero. However, these are very special parameters. Moreover, it can be shown that the number of linear regions of a maxout network is a lower semi-continuous function of the parameters, in the sense that sufficiently small generic perturbations of the parameters do not decrease the number of linear regions (Montúfar et al., 2021, Proposition 3.2). Hence, the question arises: What is the smallest number of linear regions that will occur with positive probability over the parameter space (i.e. for all parameters except for a null set). For example, in the case of shallow ReLU networks, it is known that the number of regions for generic parameters is equal to the maximum. For maxout networks we saw in Theorem 7 that several numbers of linear regions can happen with positive probability. We prove the following lower bound on the number of regions for maxout networks with generic parameters.

**Theorem 8** (Generic lower bound on the number of linear regions). *Consider a rank-$K$ maxout network, $K \ge 2$, with $n_0$ inputs, $n_1$ units in the first layer, and any number of additional nonzero width layers. Then, for almost every choice of the parameters, the number of linear regions is at least $\sum_{j=0}^{n_0} \binom{n_1}{j}$ and the number of bounded linear regions is at least $\binom{n_1 - 1}{n_0}$.*

First we observe that for generic parameters, the number of linear regions of the function represented by a network is bounded below by the number of linear regions of the network restricted to the first layer. This is not trivial, since the deeper layers could in principle map the values from the first layer

to a constant value, resulting in a function with a single linear region. However, for maxout networks this only happens for a null set of parameters.

**Proposition 23.** *The number of activation regions of a maxout network is at least as large as the number of regions of the first layer. Moreover, for generic parameters the number of linear regions is equal to the number of activation regions.*

*Proof.* The number of regions never reduces as we pass through the network. The region is either kept as it is or split into parts by a neuron. The fact that for generic parameters activation regions correspond to linear regions is Lemma 5. □

In order to lower bound the number of regions of a single layer, we use the correspondence between linear regions and the upper vertices of the corresponding lifted Newton polytope, Proposition 20. We first observe that the Newton polytope of a shallow maxout units is the Minkowski sum of the Newton polytopes of the individual units. Recall that the Minkowski sum of two sets $A$ and $B$ is the set $A + B = \{a + b : a \in A, b \in B\}$.

**Proposition 24.** *Consider a layer of maxout units, $f \colon \mathbb{R}^n \to \mathbb{R}^m$; $f_i(x) = \max\{w_{ir} \cdot x + b_{ir} : r = 1, \dots, k\}$. Let $f(x) = \sum_{i=1}^m f_i(x)$. Then the lifted Newton polytope of $f$ is the Minkowski sum of the lifted Newton polytopes of $f_1, \dots, f_m$, $P_f = \sum_{i=1}^m P_{f_i}$.*

*Proof.* This follows from direct calculation. Details can be found in the works of Zhang et al. (2018) and Montúfar et al. (2021). □

Next, a family of polytopes $P_i = \text{conv}\{(w_{i,r}, b_{i,r}) \in \mathbb{R}^{n_0+1} \colon r = 1, \dots, K\}$ with generic $(w_{i,r}, b_{i,r})$, $r = 1, \dots, K$, $i = 1, \dots, n_1$, is in general orientation. For such a family, the Minkowski sum $P = P_1 + \cdots + P_{n_1}$ has at least as many vertices as a Minkowski sum of $n_1$ line segments in general orientation:

**Proposition 25** (Adiprasito 2017, Corollary 8.2). *The number of vertices of a Minkowski sum of $m$ polytopes in general orientation is lower bounded by the number of vertices of a sum of $m$ line segments in general orientations.*

From this, we derive a lower bound on the number of upper vertices of a Minkowski sum of polytopes in general orientations.

**Proposition 26.** *The number of upper vertices of a Minkowski sum of $n_1$ polytopes in $\mathbb{R}^{n_0+1}$ in general orientation is at least $\sum_{j=0}^{n_0} \binom{n_1}{j}$, and the number of strict upper vertices is at least $\binom{n_1-1}{n_0}$.*

*Proof.* Consider the sum $P = P_1 + \cdots + P_{n_1}$ of polytopes $P_i = \{(w_{i,r}, b_{i,r}) \colon r = 1, \dots, k\}$, $i = 1, \dots, n_1$. The set of upper vertices consists of 1) strict upper vertices and 2) vertices which are both upper and lower. The number of strict upper vertices of a Minkowski sum of $n_1$ positive dimensional polytopes in general orientations in $\mathbb{R}^{n_0+1}$ is at least $\binom{n_1-1}{n_0}$ (Montúfar et al., 2021, Corollary 3.8).

Now note that the vertices which are upper and lower are precisely the vertices of the Minkowski sum $Q = Q_1 + \cdots + Q_{n_1}$ of the polytopes $Q_i = \text{conv}\{w_{i,r} \in \mathbb{R}^{n_0} \colon r = 1, \dots, k\}$, $i = 1, \dots, n_1$. By Proposition 25 the total number of vertices of a Minkowski sum is at least equal to the number of vertices of a Minkowski sum of line segments. The latter is the same as the number of regions of a central hyperplane arrangement in $n_0$ dimensions, which is $\binom{n_1-1}{n_0-1} + \sum_{j=0}^{n_0-1} \binom{n_1}{j}$.

Hence for any generic Minkowski sum of $n_1$ positive-dimensional polytopes in $n_0 + 1$ dimensions, the number of upper vertices is at least

$$\binom{n_1 - 1}{n_0} + \binom{n_1 - 1}{n_0 - 1} + \sum_{j=0}^{n_0-1} \binom{n_1}{j} = \binom{n_1}{n_0} + \sum_{j=0}^{n_0-1} \binom{n_1}{j} = \sum_{j=0}^{n_0} \binom{n_1}{j}.$$

This concludes the proof. □

Now we have all tools we need to prove the theorem.

*Proof of Theorem 8.* By Proposition 23, the number of regions is lower bounded by the number of regions of the first layer. We now derive a lower bound for the number of regions of a single layer with $n_0$ inputs and $n_1$ maxout units. In view of Propositions 20 and 24, we need to lower bound the number of upper vertices of a generic Minkowksi sum. The bounded regions correspond to the strict upper vertices. The result follows from Proposition 26. □

**Remark 27.** The statement of Theorem 8 does not apply to ReLU networks unless they have a single layer of ReLUs. Indeed, for a network with 2 layers of ReLUs there exists a positive measure subset of parameters for which the represented functions have only 1 linear region. To see this, consider a ReLU network with pre-activation features of the units in the second layer always being non-positive. A subset of parameters required to achieve this is defined as a solution to a set of inequalities (for instance, when the input weights and biases of the second layer are non-positive) and has a positive measure. For such pre-activation features, the ReLUs in the second layer always output 0 and there is a single linear region for the network.

## C Expected number of activation regions of a single maxout unit

We discuss a single maxout unit with $n$ inputs. In this case, the $r$-partial activation patterns correspond to the $r$-dimensional upper faces of a polytope given as the convex hull of the points $(w_k, b_k) \in \mathbb{R}^{n+1}$, $k = 1, \ldots, K$. The statistics of faces of random polytopes have been studied in the literature (Hug et al., 2004; Hug and Reitzner, 2005; Bárány and Vu, 2007). We will use the following result.

**Theorem 28** (Hug et al. 2004, Theorem 1.1)**.** *If $v_1, \ldots, v_K$ are sampled iid according to the standard normal distribution in $\mathbb{R}^d$, then, the number of $s$-faces of the convex hull $P_K = \mathrm{conv}\{v_1, \ldots, v_K\}$, denoted $f_s(P_K)$, has expected value*

$$\mathbb{E} f_s(P_K) \sim \bar{c}(s, d)(\log K)^{\frac{d-1}{2}}, \tag{2}$$

*and the union $s$-faces of $P_K$, denoted $\mathrm{skel}_s(P_K)$, has expected volume*

$$\mathbb{E} \, \mathrm{vol}_s(\mathrm{skel}_s(P_K)) \sim c(s, d)(\log K)^{\frac{d-1}{2}}, \tag{3}$$

*where $\bar{c}(s, d)$ and $c(s, d)$ are constants depending only on $s$ and $d$.*

Based on this, we obtain the following upper bound for the expected number of linear regions of a maxout unit with iid Gaussian weights and biases.

**Proposition 29** (Expected number of regions of a large-rank Gaussian maxout unit)**.** *Consider a rank-$K$ maxout unit with $n_0$ inputs. If the weights and biases are sampled iid from a standard normal distribution, then for large $K$ the expected number of non-empty $r$-partial activation regions satisfies*

$$\mathbb{E}[\# \ r\text{-partial activation regions}] \leq \bar{c}(r, n_0)(\log K)^{\frac{n_0}{2}}.$$

*where $\bar{c}(r, n_0)$ is a constants depending solely on $r$ and $n_0$.*

*Proof of Proposition 29.* We use the correspondence between $r$-partial activation regions and the upper $r$-dimensional faces of the lifted Newton polytope (Proposition 20). The total number of $s$-dimensional faces of a polytope is an upper bound on the number of upper $s$-dimensional faces. Now we just apply Theorem 28. □

We can use the above result to upper bound the expectation value of the number of regions of a maxout network with iid Gaussian weights and biases. In particular, for a shallow maxout network we have the following.

**Proposition 30** (Expected number of linear regions of a large-rank Gaussian maxout layer)**.** *Consider network $\mathcal{N}$ with $n_0$ inputs and a single layer of $n_1$ rank-$K$ maxout units. Suppose the weights and biases are sampled iid from a standard normal distribution. Then, for sufficiently large $K$, the expected number of linear regions is bounded as*

$$\mathbb{E}[\# \ \text{linear regions}] \leq \sum_{j=0}^{n_0} \binom{n_1}{j} (\bar{c}(n_0)(\log K)^{\frac{n_0}{2}} - 1)^j,$$

*where $\bar{c}(n_0)$ is a constant depending solely on $n_0$. This upper bound behaves as $O(n_1^{n_0}(\log K)^{\frac{1}{2}n_0^2})$ in $n_1$ and $K$.*

*Proof.* By Montúfar et al. (2021, Theorem 3.6), the maximum number of regions of a layer with $n_0$ inputs and $n_1$ maxout units of ranks $k_1, \ldots, k_{n_1}$ is

$$\max[\# \text{ linear regions}] = \sum_{j=0}^{n_0} \sum_{S \in \binom{[n_1]}{j}} \prod_{i \in S} (k_i - 1).$$

Consider now our network with $n_1$ maxout units of rank $K$. For a given probability distribution over the parameter space, denote $\Pr(k_1, \ldots, k_{n_1})$ the probability of the event that the $i$-th unit has $k_i$ linear regions, $i = 1, \ldots, n_1$. If the parameters of the different units are independent, we have

$$\mathbb{E}[\# \text{ linear regions}] \leq \sum_{1 \leq k_1, \ldots, k_{n_1} \leq K} \Pr(k_1, \ldots, k_{n_1}) \sum_{j=0}^{n_0} \sum_{S \in \binom{[n_1]}{j}} \prod_{i \in S} (k_i - 1)$$

$$= \sum_{j=0}^{n_0} \sum_{S \in \binom{[n_1]}{j}} \prod_{i \in S} (\mathbb{E}[k_i] - 1).$$

If the weights and biases of each unit are iid normal, Proposition 29 allows us to upper bound the latter expression by

$$\leq \sum_{j=0}^{n_0} \binom{n_1}{j} (\bar{c}(n_0)(\log K)^{\frac{n_0}{2}} - 1)^j.$$

This concludes the proof. $\qquad\square$

## D   Proofs related to the expected volume

The following is a maxout version of a result obtained by Hanin and Rolnick (2019a, Theorem 6) for the case of networks with single-argument piecewise linear activation functions.

**Lemma 31** (Upper bound on the expected volume of $\mathcal{X}_{\mathcal{N},r}$). *Consider a rank-$K$ maxout network $\mathcal{N}$ with input dimension $n_0$, output dimension $1$, and random weights and biases satisfying:*

1. *The distribution of all weights has a density with respect to the Lebesgue measure.*

2. *Every collection of biases has a conditional density with respect to Lebesgue measure given the values of all weights and other biases.*

*Then, for any bounded measurable set $S \subset \mathbb{R}^{n_0}$ and any $r \in \{1, \ldots, n_0\}$, the expectation value of the $(n_0 - r)$-dimensional volume of $\mathcal{X}_{\mathcal{N},r}$ inside $S$ is upper bounded as*

$$\mathbb{E}[\text{vol}_{n_0-r}(\mathcal{X}_{\mathcal{N},r} \cap S)]$$
$$\leq \sum_{J \in \mathcal{S}_r} \int_S \mathbb{E}\left[\rho_{\mathbf{b}^r}((\mathbf{w}^m - \mathbf{w}^r) \cdot \mathbf{x}_{-1}^m + \mathbf{b}^m) \ \|\mathbf{J}((\mathbf{w}^m - \mathbf{w}^r) \cdot \mathbf{x}_{-1}^m + \mathbf{b}^m)\|\right] \ dx,$$

*where, for any given $r$-partial activation sub-pattern $J = (J_z)_{z \in \mathcal{Z}} \in \mathcal{S}_r$, for any given $J_z$ we denote its smallest element by $j_0$, we let $\rho_{\mathbf{b}^r}$ denote the joint conditional density of the biases of pre-activation features $j \in J_z \setminus \{j_0\}$ of the neurons $z \in \mathcal{Z}$, given all other network parameters, we let $g \colon \mathbb{R}^{n_0} \to \mathbb{R}^r$; $x \mapsto (\mathbf{w}^m - \mathbf{w}^r) \cdot \mathbf{x}_{-1}^m + \mathbf{b}^m := ((w_{z,j_0} - w_{z,j}) \cdot x_{l(z)-1} + b_{z,j_0})_{z \in \mathcal{Z}, j \in J_z \setminus \{j_0\}} \in \mathbb{R}^r$, denote $\mathbf{J}g$ the $r \times n_0$ Jacobian of $g$, and $\|\mathbf{J}g(x)\| = \det\left((\mathbf{J}g(x))(\mathbf{J}g(x))^\top\right)^{\frac{1}{2}}$, and the inner expectation is with respect to all parameters aside these biases.*

*Proof of Lemma 31.* The proof follows the arguments of Hanin and Rolnick (2019a, Theorem 6). The main difference is that maxout units are generically active and the activation regions of maxout units may involve several pre-activation features and additional inequalities. To obtain the upper bound, we will discard certain inequalities, and separate one distinguished pre-activation feature $j_0$ for each neuron participating in a sub-pattern, which allows us to relate inputs in the corresponding activation regions to bias values and apply the co-area formula.

Recall that an $r$-partial activation sub-pattern $J \in \mathcal{S}_r$ is a list of patterns $J_z \subseteq [K]$ of cardinality at least 2 for a collection of participating neurons $z \in \mathcal{Z}$, with $\sum_{z \in \mathcal{Z}}(|J_z| - 1) = r$. Further, for any given $J_z$ we denote $j_0$ its smallest element. When discussing a particular sub-pattern, we will write $m = |\mathcal{Z}|$ for the number of participating neurons. Finally, recall that $\mathcal{H}(J, \theta) = \bigcap_{z \in \mathcal{Z}} \mathcal{H}(J_z, \theta)$.

By Corollary 17, with probability 1 with respect to $\theta$,

$$\mathrm{vol}_{n_0-r}(\mathcal{X}_{\mathcal{N}, r}(\theta)) = \sum_{J \in \mathcal{S}_r} \mathrm{vol}_{n_0-r}(\mathcal{H}(J, \theta)).$$

Fix $J \in \mathcal{S}_r$. In the following we prove that

$$\mathbb{E}[\mathrm{vol}_{n_0-r}(\mathcal{H}(J, \theta) \cap S)] \leq \int_S \mathbb{E}\left[\rho_{\mathbf{b}^r}((\mathbf{w}^m - \mathbf{w}^r) \cdot \mathbf{x}_{-1}^m + \mathbf{b}^m) \; \|\mathbf{J}((\mathbf{w}^m - \mathbf{w}^r) \cdot \mathbf{x}_{-1}^m + \mathbf{b}^m)\|\right] \; dx.$$

We first note that

$$\mathrm{vol}_{n_0-r}(\mathcal{H}(J, \theta) \cap S) = \int_{\mathcal{H}(J, \theta) \cap S} d\,\mathrm{vol}_{n_0-r}(x). \tag{4}$$

For each $z \in \mathcal{Z}$ and $J_z$ we can pick an element $j_0 \in J_z$ and express $\mathcal{H}(J_z, \theta)$ in terms of $(|J_z| - 1)$ equations and $(K - |J_z|)$ inequalities (not necessarily linear),

$$\mathcal{H}(J_z, \theta) = \{x \in \mathbb{R}^{n_0} \mid w_{z,j_0} \cdot x_{l(z)-1} + b_{z,j_0} = w_{z,j} \cdot x_{l(z)-1} + b_{z,j}, \quad \forall j \in J_z \setminus \{j_0\}; \tag{5}$$
$$w_{z,j_0} \cdot x_{l(z)-1} + b_{z,j_0} > w_{z,i} \cdot x_{l(z)-1} + b_{z,i}, \quad \forall i \in [K] \setminus J_z\}.$$

Here, $x_{l(z)-1}$ are the activation values of the units in the layer preceding unit $z$, depending on the input $x$. Since $\sum_z(|J_z| - 1) = r$, the set $\mathcal{H}(J, \theta)$ is defined by $r$ equations (in addition to inequalities). We will denote with $\mathbf{b}^r \in \mathbb{R}^r$ the vector of biases $b_{z,j}$ that are involved in these $r$ equations, with subscripts $(z, j)$ with $j \in J_z \setminus \{j_0\}$ and $z \in \mathcal{Z}$.

We take the expectation of (4) with respect to the conditional distribution of $\mathbf{b}^r$ given the values of all the other network parameters. We have assumed that this has a density. Denoting the conditional density of $\mathbf{b}^r$ by $\rho_{\mathbf{b}^r}$, this is given by

$$\int_{\mathbb{R}^r} \int_{\mathcal{H}(J, \theta) \cap S} d\,\mathrm{vol}_{n_0-r}(x)\rho_{\mathbf{b}^r}(\mathbf{b}^r)d\mathbf{b}^r. \tag{6}$$

The equations in (5) imply that $b_{z,j} = (w_{z,j_0} - w_{z,j}) \cdot x_{l(z)-1} + b_{z,j_0}$ for any $x \in \mathcal{H}(J, \theta)$. We write all these equations concisely as $\mathbf{b}^r = (\mathbf{w}^m - \mathbf{w}^r) \cdot \mathbf{x}_{-1}^m + \mathbf{b}^m$. Then (6) becomes

$$\int_{\mathbb{R}^r} \int_{\mathcal{H}(J, \theta) \cap S} \rho_{\mathbf{b}^r}((\mathbf{w}^m - \mathbf{w}^r) \cdot \mathbf{x}_{-1}^m + \mathbf{b}^m) \; d\,\mathrm{vol}_{n_0-r}(x)d\mathbf{b}^r. \tag{7}$$

We will upper bound the volume of $\mathcal{H}(J, \theta)$ by the volume of the corresponding set without the inequalities,

$$\mathcal{H}'(J, \theta) := \bigcap_{z \in \mathcal{Z}} \{x \in \mathbb{R}^{n_0} \mid w_{z,j_0} \cdot x_{l(z)-1} + b_{z,j_0} = w_{z,j} \cdot x_{l(z)-1} + b_{z,j}, \quad \forall j \in J_z \setminus \{j_0\}\}.$$

Since $\mathcal{H}(J, \theta) \subseteq \mathcal{H}'(J, \theta)$, we can upper bound (7) by

$$\int_{\mathbb{R}^r} \int_{\mathcal{H}'(J, \theta) \cap S} \rho_{\mathbf{b}^r}((\mathbf{w}^m - \mathbf{w}^r) \cdot \mathbf{x}_{-1}^m + \mathbf{b}^m) \; d\,\mathrm{vol}_{n_0-r}(x)d\mathbf{b}^r. \tag{8}$$

Now we will use the co-area formula to express (8) as an integral over $S$ alone. Recall that the co-area formula says that if $\psi \in L^1(\mathbb{R}^n)$ and $g : \mathbb{R}^n \to \mathbb{R}^r$ with $r \leq n$ is Lipschitz, then

$$\int_{\mathbb{R}^r} \int_{g^{-1}(t)} \psi(x)d\,\mathrm{vol}_{n-r}(x)dt = \int_{\mathbb{R}^n} \psi(x)\|\mathbf{J}g(x)\|d\,\mathrm{vol}_n(x),$$

where $\mathbf{J}g$ is the $r \times n$ Jacobian of $g$ and $\|\mathbf{J}g(x)\| = \det((\mathbf{J}g(x))(\mathbf{J}g(x))^\top)^{\frac{1}{2}}$.

In our case $r = r$, $n = n_0$, which satisfy $r \le n_0$. Further, $\mathbf{b}^r \in \mathbb{R}^r$ plays the role of $t \in \mathbb{R}^r$, and $\mathbb{R}^{n_0} \to \mathbb{R}^r$; $x \mapsto \rho_{\mathbf{b}^r}((\mathbf{w}^m - \mathbf{w}^r) \cdot \mathbf{x}^m_{-1} + \mathbf{b}^m)$ plays the role of $\psi$. Since $(\mathbf{w}^m - \mathbf{w}^r) \cdot \mathbf{x}^m_{-1} + \mathbf{b}^m$ is continuous and $S$ is bounded, assuming $\rho_{\mathbf{b}^r}$ is continuous, this is in $L^1(S)$. Finally, we set $g \colon S \to \mathbb{R}^r$; $x \mapsto ((\mathbf{w}^m - \mathbf{w}^r) \cdot \mathbf{x}^m_{-1} + \mathbf{b}^m)$, which is Lipschitz.

Hence (8) can be expressed as

$$\int_S \rho_{\mathbf{b}^r}((\mathbf{w}^m - \mathbf{w}^r) \cdot \mathbf{x}^m_{-1} + \mathbf{b}^m) \ \|\mathbf{J}((\mathbf{w}^m - \mathbf{w}^r) \cdot \mathbf{x}^m_{-1} + \mathbf{b}^m)\| \ dx.$$

Taking expectation with respect to all other weights and biases, and interchanging the integral over $S$ with the expectation (according to Fubini's theorem, since the integral is non-negative),

$$\int_S \mathbb{E}\left[\rho_{\mathbf{b}^r}((\mathbf{w}^m - \mathbf{w}^r) \cdot \mathbf{x}^m_{-1} + \mathbf{b}^m) \ \|\mathbf{J}((\mathbf{w}^m - \mathbf{w}^r) \cdot \mathbf{x}^m_{-1} + \mathbf{b}^m)\|\right] \ dx.$$

Summing over all $r$-partial activation sub-patterns $J \in \mathcal{S}_r$ gives the desired result. $\qquad\square$

Based on the preceding Lemma 31, now we derive a more explicit upper bound expressed in terms of properties of the probability distribution of the network parameters.

**Theorem 10** (Upper bound on the expected volume of the non-linear locus). *Consider a bounded measurable set $S \subset \mathbb{R}^{n_0}$ and the settings of Theorem 9 with constants $C_{grad}$ and $C_{bias}$ evaluated over $S$. Then, for any $r \in \{1, \dots, n_0\}$,*

$$\frac{\mathbb{E}[\mathrm{vol}_{n_0-r}(\mathcal{X}_{\mathcal{N},r} \cap S)]}{\mathrm{vol}_{n_0}(S)} \le (2C_{\mathrm{grad}}C_{\mathrm{bias}})^r \binom{rK}{2r}\binom{N}{r}.$$

*Proof of Theorem 10.* By Lemma 31, $\mathbb{E}\left[\mathrm{vol}_{n_0-r}(\mathcal{X}_{\mathcal{N},r} \cap S)\right]$ is upper bounded by

$$\sum_{J \in \mathcal{S}_r} \int_S \mathbb{E}\left[\rho_{\mathbf{b}^r}((\mathbf{w}^m - \mathbf{w}^r) \cdot \mathbf{x}^m_{-1} + \mathbf{b}^m) \ \|\mathbf{J}((\mathbf{w}^m - \mathbf{w}^r) \cdot \mathbf{x}^m_{-1} + \mathbf{b}^m)\|\right] \ dx.$$

Since we have assumed that for any collection of $t$ biases the conditional density given all weights and the other biases can be upper-bounded with $C_{\mathrm{bias}}^t$, we have $\rho_{\mathbf{b}^r}((\mathbf{w}^m - \mathbf{w}^r) \cdot \mathbf{x}^m_{-1} + \mathbf{b}^m) \le C_{\mathrm{bias}}^r$.

As for the term $\mathbb{E}[\|\mathbf{J}((\mathbf{w}^m - \mathbf{w}^r) \cdot \mathbf{x}^m_{-1} + \mathbf{b}^m)\|]$, note that

$$\|\mathbf{J}((\mathbf{w}^m - \mathbf{w}^r) \cdot \mathbf{x}^m_{-1} + \mathbf{b}^m)\|$$
$$= \det\left(\mathbf{J}((\mathbf{w}^m - \mathbf{w}^r) \cdot \mathbf{x}^m_{-1} + \mathbf{b}^m)^T \mathbf{J}((\mathbf{w}^m - \mathbf{w}^r) \cdot \mathbf{x}^m_{-1} + \mathbf{b}^m)\right)^{1/2}$$
$$= \det\left(\mathrm{Gram}\left(\nabla((w_{z_1,j_0} - w_{z_1,j_1}) \cdot x_{l(z_1)-1} + b_{z_1,j_0}), \dots, \right.\right.$$
$$\left.\left. \nabla((w_{z_m,j_0} - w_{z_m,j_{r_m}}) \cdot x_{l(z_m)-1} + b_{z_m,j_0})\right)\right)^{1/2}, \tag{9}$$

where for any $v_1, \dots, v_r \in \mathbb{R}^n$, $(\mathrm{Gram}(v_1, \dots, v_r))_{i,j} = \langle v_i, v_j \rangle$ is the associated Gram matrix.

It is known that the Gram determinant can also be expressed in terms of the exterior product of vectors, meaning that (9) can be written as

$$\|\nabla((w_{z_1,j_0} - w_{z_1,j_1}) \cdot x_{l(z_1)-1} + b_{z_1,j_0}) \wedge \dots \wedge \nabla((w_{z_m,j_0} - w_{z_m,j_{r_m}}) \cdot x_{l(z_m)-1} + b_{z_m,j_0})\|,$$

which is the the $r$-dimensional volume of the parallelepiped in $\mathbb{R}^{n_0}$ spanned by $r$ elements. Therefore, for $J \in \mathcal{S}_r$ with participating neurons $Z$, we can upper bound this expression by (see Gover and Krikorian, 2010)

$$\prod_{z \in Z} \prod_{j \in J_z \setminus \{j_0\}} \|\nabla((w_{z,j_0} - w_{z_i,j}) \cdot x_{l(z)-1} + b_{z,j_0})\|$$
$$\le \prod_{z \in Z} \prod_{j \in J_z \setminus \{j_0\}} 2\max\left\{\|\nabla(w_{z,j_0} \cdot x_{l(z)-1})\|, \|\nabla(w_{z,j} \cdot x_{l(z)-1})\|\right\}$$
$$\le 2^r \max_{z \in Z, j \in J_z} \left\{\|\nabla(w_{z,j} \cdot x_{l(z)-1})\|\right\}^r.$$

In the second line we use the triangle inequality. Considering the assumption that we have made on the gradients, for the expectation we obtain the upper bound $(2C_{\text{grad}})^r$.

By Lemma 6, we can upper-bound the number of entries of the sum $\sum_{J \in \mathcal{S}_r}$ with $\binom{rK}{2r}\binom{N}{r}$. Combining everything, we get the final upper bound

$$(2C_{\text{grad}}C_{\text{bias}})^r \binom{rK}{2r}\binom{N}{r} \text{vol}_{n_0}(S).$$

This concludes the proof. $\qquad\square$

## E   Proofs related to the expected number of regions

**Theorem 9** (Upper bound on the expected number of partial activation regions)**.** *Let $\mathcal{N}$ be a fully-connected feed-forward maxout network, with $n_0$ inputs, a total of $N$ rank $K$ maxout units. Suppose we have a probability distribution over the parameters so that:*

1. *The distribution of all weights has a density with respect to Lebesgue measure on $\mathbb{R}^{\#\text{weights}}$.*

2. *Every collection of biases has a conditional density with respect to Lebesgue measure given the values of all other weights and biases.*

3. *There exists $C_{\text{grad}} > 0$ so that for any $t \in \mathbb{N}$ and any pre-activation feature $\zeta_{z,k}$,*

$$\sup_{x \in \mathbb{R}^{n_0}} \mathbb{E}[\|\nabla \zeta_{z,k}(x)\|^t] \le C_{\text{grad}}^t.$$

4. *There exists $C_{\text{bias}} > 0$ so that for any pre-activation features $\zeta_1, \ldots, \zeta_t$ from any neurons, the conditional density of their biases $\rho_{b_1,\ldots,b_t}$ given all the other weights and biases satisfies*

$$\sup_{b_1,\ldots,b_t \in \mathbb{R}} \rho_{b_1,\ldots,b_t}(b_1, \ldots, b_t) \le C_{\text{bias}}^t.$$

*Fix $r \in \{0, \ldots, n_0\}$ and let $T = 2^5 C_{\text{grad}} C_{\text{bias}}$. Then, there exists $\delta_0 \le 1/(2C_{\text{grad}}C_{\text{bias}})$ such that for all cubes $C \subseteq \mathbb{R}^{n_0}$ with side length $\delta > \delta_0$, we have*

$$\frac{\mathbb{E}[\# \ r\text{-partial activation regions of } \mathcal{N} \text{ in } C]}{\text{vol}(C)} \le \begin{cases} \binom{rK}{2r}\binom{N}{r}K^{N-r}, & N \le n_0 \\ \dfrac{(TKN)^{n_0}\binom{n_0 K}{2n_0}}{(2K)^r n_0!}, & N \ge n_0 \end{cases}.$$

*Here the expectation is taken with respect to the distribution of weights and biases in $\mathcal{N}$. Of particular interest is the case $r = 0$, which corresponds to the number of linear regions.*

*Proof of Theorem 9.* The proof follows closely the arguments of Hanin and Rolnick (2019b, Proof of Theorem 10), whereby we use appropriate supporting results for maxout networks and need to accommodate the combinatorics depending on $K$. Fix a network $\mathcal{N}$ with rank-$K$ maxout units, input dimension $n_0$ and output dimension 1. Let $0 \le r \le n_0$. For $N \le n_0$, the statement follows direction from the simple upper bound on the number of distinct $r$-partial activation patterns given in Lemma 6.

Consider now the case $N \ge n_0$. Fix a closed cube $C \subseteq \mathbb{R}^{n_0}$ of sidelength $\delta > 0$. For any $t \in \{0, \ldots, n_0\}$ let

$$C_t := t\text{-skeleton of } C$$

denote the union of $t$-dimensional faces of $C$. For example, $C_0$ is the set of $2^{n_0}$ vertices of $C$, $C_{n_0-1}$ is the set of $2n_0$ facets of $C$, and $C_{n_0}$ is $C$. In general, $C_t$ consists of $\binom{n_0}{t}2^{n_0-t}$ faces of dimension $t$, each with $t$-volume $\delta^t$. Hence,

$$\text{vol}_t(C_t) = \binom{n_0}{t}2^{n_0-t}\delta^t. \qquad (10)$$

For any choice of $\theta$ let

$$\mathcal{V}_t(\theta) := \mathcal{X}_{\mathcal{N},t}(\theta) \cap C_t.$$

By Lemma 33 below, for any $t$ and almost every choice of $\theta$, the set $\mathcal{V}_t(\theta)$ is a finite set of points. For each $t \in \{0, \ldots, n_0\}$, we also define

$$C_{t,\varepsilon} := \{x \in \mathbb{R}^{n_0} \mid \text{dist}(x, C_t) \leq \varepsilon\},$$

the $\varepsilon$-thickening of $C_t$. For almost every $\theta$, Lemma 34 ensures the existence of an $\varepsilon > 0$ such that for all $v \in \mathcal{V}_t(\theta)$, the radius-$\varepsilon$ balls $B_\varepsilon(v)$ are contained in $C_{t,\varepsilon}$ and are disjoint. Hence, writing $\omega_{n_0-t}$ for the $(n_0 - t)$-volume of the $(n_0 - t)$-dimensional ball with unit radius,

$$\text{vol}_{n_0-t}(X_{\mathcal{N},t} \cap C_{t,\varepsilon}) \geq \sum_{v \in \mathcal{V}_t} \varepsilon^{n_0-t}\omega_{n_0-t} = \#\mathcal{V}_t \cdot \varepsilon^{n_0-t}\omega_{n_0-t}.$$

Therefore, for all but a measure 0 set of $\theta \in \mathbb{R}^{\#\text{params}}$, there exists $\varepsilon > 0$ so that

$$\frac{\text{vol}_{n_0-t}(X_{\mathcal{N},t} \cap C_{t,\varepsilon})}{\varepsilon^{n_0-t}\omega_{n_0-t}} \geq \#\mathcal{V}_t.$$

Thus taking the limit $\varepsilon \to 0$ and taking expectation with respect to the parameter $\theta$, and applying Fatou's lemma to upper bound the result by the expression with exchanged limit and expectation,

$$\mathbb{E}\left[\#\mathcal{V}_t\right] \leq \mathbb{E}\left[\lim_{\varepsilon \to 0} \frac{\text{vol}_{n_0-t}(\mathcal{X}_{\mathcal{N},t} \cap C_{t,\varepsilon})}{\varepsilon^{n_0-t}\omega_{n_0-t}}\right] \leq \lim_{\varepsilon \to 0} \mathbb{E}\left[\frac{\text{vol}_{n_0-t}(\mathcal{X}_{\mathcal{N},t} \cap C_{t,\varepsilon})}{\varepsilon^{n_0-t}\omega_{n_0-t}}\right].$$

Then,

$$\mathbb{E}\left[\#\mathcal{V}_t\right] \leq \lim_{\varepsilon \to 0} \mathbb{E}\left[\frac{\text{vol}_{n_0-t}(\mathcal{X}_{\mathcal{N},t} \cap C_{t,\varepsilon})}{\text{vol}_{n_0}(C_{t,\varepsilon})} \cdot \frac{\text{vol}_{n_0}(C_{t,\varepsilon})}{\varepsilon^{n_0-t}\omega_{n_0-t}}\right]$$

$$= \lim_{\varepsilon \to 0} \mathbb{E}\left[\frac{\text{vol}_{n_0-t}(\mathcal{X}_{\mathcal{N},t} \cap C_{t,\varepsilon})}{\text{vol}_{n_0}(C_{t,\varepsilon})}\right] \cdot \lim_{\varepsilon \to 0} \frac{\text{vol}_{n_0}(C_{t,\varepsilon})}{\varepsilon^{n_0-t}\omega_{n_0-t}}$$

$$\leq (2C_{\text{grad}}C_{\text{bias}})^t \binom{tK}{2t}\binom{N}{t} \text{vol}_t(C_t).$$

To obtain the last line, the first term is upper bounded using Theorem 10, and the second term is evaluated using

$$\lim_{\varepsilon \to 0} \frac{\text{vol}_{n_0}(C_{t,\varepsilon})}{\varepsilon^{n_0-t}\omega_{n_0-t}} = \text{vol}_t(C_t).$$

Combining this with Lemma 33 and the formula (10) for $\text{vol}_t(C_t)$, we find

$$\mathbb{E}\left[\#\{r\text{-partial activation regions with } \mathcal{R}(J^r; \theta) \cap C \neq \emptyset\}\right]$$

$$\leq \sum_{t=r}^{n_0} \binom{t}{r} K^{t-r}(2C_{\text{grad}}C_{\text{bias}})^t \binom{tK}{2t}\binom{N}{t}\binom{n_0}{t}2^{n_0-t}\delta^t$$

$$\overset{\delta \geq 1/(2C_{\text{grad}}C_{\text{bias}})}{\leq} (2\delta C_{\text{grad}}C_{\text{bias}})^{n_0}\binom{n_0 K}{2n_0}(2K)^{n_0-r}\sum_{t=r}^{n_0}\binom{t}{r}\binom{N}{t}\binom{n_0}{t}. \tag{11}$$

The last line uses the assumption that $\delta \geq 1/(2C_{\text{grad}}C_{\text{bias}})$ and Lemma 32, which states that $\binom{tK}{2t} \leq \binom{nK}{2n}$ for $t \leq n$.

In the following we simplify (11). Note that $\binom{t}{r} \leq \sum_{r=0}^{t}\binom{t}{r} = 2^t \leq 2^{n_0}$. Hence (11) can be upper bounded by

$$(4\delta C_{\text{grad}}C_{\text{bias}})^{n_0}\binom{n_0 K}{2n_0}(2K)^{n_0-r}\sum_{t=r}^{n_0}\binom{N}{t}\binom{n_0}{t}$$

$$= (4\delta C_{\text{grad}}C_{\text{bias}})^{n_0}\binom{n_0 K}{2n_0}(2K)^{n_0-r}\sum_{t=r}^{n_0}\binom{n_0}{t}^2\frac{\binom{N}{t}}{\binom{n_0}{t}}.$$

Using $n_0 \leq N$, observe that

$$\frac{\binom{N}{t}}{\binom{n_0}{t}} = \frac{N! \cdot (n_0 - t)!}{(N-t)! \cdot n_0!} \leq N^t \cdot \frac{(n_0 - t)!}{n_0!} = \frac{N^{n_0}}{N^{n_0-t}} \cdot \frac{(n_0 - t)!}{n_0!} = \frac{N^{n_0}}{n_0!} \cdot \frac{(n_0 - t)!}{N^{n_0-t}}$$

$$\leq \frac{N^{n_0}}{n_0!} \cdot \frac{(n_0 - t)^{n_0-t}}{N^{n_0-t}} \leq \frac{N^{n_0}}{n_0!}.$$

Also, using Vandermonde's identity, observe that

$$\sum_{t=0}^{n_0} \binom{n_0}{t}^2 = \binom{2n_0}{n_0} \leq 4^{n_0}.$$

Combing everything, (11) is upper bounded by

$$(16\delta C_{\text{grad}} C_{\text{bias}})^{n_0} \binom{n_0 K}{2n_0} (2K)^{n_0-r} \frac{N^{n_0}}{n_0!} = (32 K C_{\text{grad}} C_{\text{bias}})^{n_0} \binom{n_0 K}{2n_0} \frac{N^{n_0}}{(2K)^r n_0!} \text{vol}(C).$$

Setting $T = 2^5 C_{\text{grad}} C_{\text{bias}}$, we get

$$\frac{(TKN)^{n_0} \binom{n_0 K}{2n_0}}{(2K)^r n_0!} \text{vol}(C).$$

This concludes the proof. $\qquad\square$

We state and prove lemmas used in the proof of Theorem 9.

**Lemma 32.** *For any $t \leq n$, $\binom{tK}{2t} \leq \binom{nK}{2n}$.*

*Proof.* To see this, note that $\binom{tK}{2t} \leq \binom{nK}{2n}$ is equivalent to the following:

$$\frac{(Kr)!}{(2r)!(Kr - 2r)!} \leq \frac{(Kn)!}{(2n)!(Kn - 2n)!}$$

$$\frac{(2n)!}{(2r)!} \frac{(Kn - 2n)!}{(Kr - 2r)!} \leq \frac{(Kn)!}{(Kr)!}$$

$$\prod_{i=1}^{2n-2r} (2r + i) \prod_{j=1}^{(K-2)n-(K-2)r} (Kr - 2r + j) \leq \prod_{k=1}^{Kn-Kr} (Kr + k).$$

Since $\prod_{i=1}^{2n-2r}(2r + i) \leq \prod_{k=1}^{2n-2r}(Kr + k)$ and $\prod_{j=1}^{(K-2)n-(K-2)r}(Kr - 2r + j) \leq \prod_{k=2n-2r+1}^{Kn-Kr}(Kr + k)$ the inequality holds. $\qquad\square$

**Lemma 33.** *For almost every $\theta$, for each $t \in \{0, \ldots, n_0\}$, the set $\mathcal{V}_t(\theta) = \mathcal{X}_{\mathcal{N},t}(\theta) \cap C_t$ consists of finitely many points and*

$$\#\{r\text{-partial activation regions } \mathcal{R}(J^r, \theta) \text{ with } \mathcal{R}(J^r, \theta) \cap C \neq \emptyset\} \leq \sum_{t=r}^{n_0} \binom{t}{r} K^{t-r} \#\mathcal{V}_t(\theta), \quad (12)$$

*where $\#\mathcal{V}_t(\theta)$ is the number of points in $\mathcal{V}_t(\theta)$.*

*Proof.* The proof is similar to the proof of (Hanin and Rolnick, 2019b, Lemma 12). The difference lies in the types of equations that appear in the partial activation regions of maxout networks. The dimension of $\mathcal{V}_t(\theta)$ is 0 with probability 1, because the set $C_t$ has dimension $t$ and, by Lemma 16, with probability 1 the set $\mathcal{X}_{\mathcal{N},t}$ coincides locally with a subspace of codimension $t$. The intersection of two generic affine spaces of complementary dimension has dimension 0.

Now we prove (12). If $J^r$ is an $r$-partial activation pattern and $\mathcal{R}(J^r, \theta) \cap C \neq \emptyset$, then the closure $\text{cl} \, \mathcal{R}(J^r, \theta) \cap C$ is a non-empty polytope. The intersection is bounded because $C$ is bounded, and, by Lemma 4, the closure of $\mathcal{R}(J^r, \theta)$ is a polyhedron. As a non-empty polytope, this set has at least one

vertex. Generically, if a vertex is in an $(n_0 - t)$-face of $\mathrm{cl}\,\mathcal{R}(J^r, \theta)$, then it is in a $t$-face of $C$. Hence, with probability 1,

$$\mathcal{R}(J^r, \theta) \cap C \neq \emptyset \quad \Rightarrow \quad \exists\, t \in \{r, \ldots, n_0\} \quad \text{s.t.} \quad \mathrm{cl}\,\mathcal{R}(J^r, \theta) \cap \mathcal{V}_t \neq \emptyset.$$

Thus, with probability 1,

$$\#\{r\text{-partial activation regions with } \mathcal{R}(J^r, \theta) \cap C \neq \emptyset\} \leq \sum_{t=r}^{n_0} T_t \# \mathcal{V}_t,$$

where $T_t$ is the maximum over all $v \in \mathcal{V}_t$ of the number of $r$-partial activation regions whose closure contains $v$.

To complete the proof, it remains to check that, with probability 1,

$$T_t \leq \binom{t}{r} K^{t-r}.$$

By the definition of $\mathcal{X}_{\mathcal{N},t}$, each $v \in \mathcal{V}_t$ is an element of exactly one $t$-partial activation region defined by $t$ equations. To upper bound the number of $r$-partial activation regions that contain $v$, we upper bound the number of ways in which one can get an $r$-partial region from this $t$-partial region. We have $\binom{t}{r}$ options to pick $r$ equations that will remain satisfied. In each case, there are at most $t - r$ neurons for which we need to specify a pre-activation feature attaining the maximum, for a total of at most $K^{t-r}$ options. This concludes the proof. $\qquad\square$

**Lemma 34.** *Fix $t \in \{0, \ldots n_0\}$. For almost every choice of $\theta$, there exists $\varepsilon > 0$ (depending on $\theta$) so that the balls $\mathcal{B}_\varepsilon(v)$ of radius $\varepsilon$ centered at $v \in \mathcal{V}_t$ are disjoint and*

$$\mathrm{vol}_{n_0 - t}(\mathcal{X}_{\mathcal{N},t} \cap \mathcal{B}_\varepsilon(v)) = \varepsilon^{n_0 - t} \omega_{n_0 - t},$$

*where $\omega_t$ is the volume of a unit ball in $\mathbb{R}^t$.*

*Proof.* The proof is similar to the proof of Hanin and Rolnick (2019b, Lemma 13), whereby we use Lemma 33 and the results for maxout networks obtained in Section A. By Lemma 33, with probability 1 over $\theta$, each $\mathcal{V}_t$ is a finite set of points. Hence, we may choose $\varepsilon > 0$ sufficiently small so that the balls $B_\varepsilon(v)$ are disjoint. Moreover, by Lemma 16, in a sufficiently small neighborhood of $v \in \mathcal{V}_t$, the set $\mathcal{X}_{\mathcal{N},t}$ coincides with a $(n_0 - t)$-dimensional subspace. The $(n_0 - t)$-dimensional volume of this subspace in $B_\varepsilon(v)$ is the volume of $(n_0 - t)$-dimensional ball of radius $\varepsilon$, which equals $\varepsilon^{n_0 - t} \omega_{n_0 - t}$, completing the proof. $\qquad\square$

To conclude this section, we compare the results on the numbers of activation regions of maxout and ReLU networks in Table 2.

Table 2: Comparison of the activation region results for maxout and ReLU networks.

| | ReLU network | Maxout network |
|---|---|---|
| Generic lower bound on the number of linear regions for a deep network | 1, Remark 27 | $\sum_{j=0}^{n_0} \binom{n_1}{j}$, Theorem 8 |
| Trivial upper-bound on the number of $r$-partial activation regions | $\binom{N}{r} 2^{N-r}$, (Hanin and Rolnick, 2019b, Theorem 10) | $\binom{rK}{2r}\binom{N}{r} K^{N-r}$, Lemma 6, see also Proposition 14 |
| Upper-bound on the expected number of $r$-partial activation regions, $N \geq n_0$ | $\frac{(TN)^{n_0}}{2^r n_0!}$, $T = 2^5 C_{\mathrm{grad}} C_{\mathrm{bias}}$, (Hanin and Rolnick, 2019b, Theorem 10) | $\frac{(TKN)^{n_0}\binom{n_0 K}{2n_0}}{(2K)^r n_0!}$, $T = 2^5 C_{\mathrm{grad}} C_{\mathrm{bias}}$, Theorem 9 |
| Upper bound on the expected $(n_0 - r)$-dimensional volume of the non-linear locus | $(2 C_{\mathrm{grad}} C_{\mathrm{bias}})^r \binom{N}{r}$, (Hanin and Rolnick, 2019a, Corollary 7) | $(2 C_{\mathrm{grad}} C_{\mathrm{bias}})^r \binom{rK}{2r}\binom{N}{r}$, Theorem 10 |

# F   Upper bounding the constants

We briefly discuss the constants $C_{\text{bias}}$ and $C_{\text{bias}}$ in the hypothesis of Theorem 9. The constant $C_{\text{bias}}$ can be evaluated at initialization using the definition, since we know the distribution of biases. Recall that we defined $C_{\text{bias}}$ as an upper bound on

$$\left( \sup_{b_1,\ldots,b_t \in \mathbb{R}} \rho_{b_1,\ldots,b_t}(b_1,\ldots,b_t) \right)^{1/t},$$

where $\rho_{b_1,\ldots,b_t}$ is the conditional distribution of any collection of biases given all the other weights and biases in $\mathcal{N}$ and $t \in \mathbb{N}$. If the biases are sampled independently, independently of the weights, this equals $\sup_{b \in \mathbb{R}} \rho_b(b)$. Then, for instance, for a normal distribution with standard deviation $\sqrt{C/n_l}$, the constant $C_{\text{bias}}$ can be chosen as

$$\max_{l \in \{0,\ldots,L-1\}} \sqrt{\frac{n_l}{2\pi C}}.$$

The constant $C_{\text{grad}}$ was defined as an upper bound on

$$\left( \sup_{x \in \mathbb{R}^{n_0}} \mathbb{E}[\|\nabla \zeta_{z,k}(x)\|^t] \right)^{1/t}.$$

Therefore we need to upper-bound $\mathbb{E}\left[\|\nabla \zeta_{z,k}(x)\|^t\right]$. This expression stands for the $t$-th moment of the L2 norm of the gradient of a pre-activation feature $\zeta_{z,k}$ in a network, with respect to the input to the network.

One possible calculation is as follows. We consider $J_x = [\nabla_x \mathcal{N}_1(x;\theta),\ldots,\nabla_x \mathcal{N}_{n_L}(x;\theta)]^\top$ the Jacobian of the output vector with respect to the input, for a given parameter $\theta$ and input $x$. Note that the gradient $\nabla \zeta_{z,k}(x)$ for a pre-activation feature of a unit in the $l$-th layer of a network is a row in the Jacobian matrix of an $l$-layer network. Therefore, $\|\nabla \zeta_{z,k}(x)\|$ can be upper-bounded by the spectral norm $\|J_x\|$ of the Jacobian, and the moments of the Jacobian norm can be used as an upper-bound on the $t$-th moments of the gradient norm, $t \geq 1$.

**Proposition 35** (Upper bound on the moments of the Jacobian matrix norm). *Let $\mathcal{N}$ be a fully-connected feed-forward network with maxout units of rank $K$ and a linear last layer. Let the network have $L$ layers of widths $n_1,\ldots,n_L$ and $n_0$ inputs. Assume that the weights and biases of the units in the $l$-th layer are sampled iid from a Gaussian distribution with mean $0$ and variance $c/n_{l-1}$, $l = 1,\ldots,L$ and $c$ is some constant $c \in \mathbb{R}, c > 0$. Then*

$$\mathbb{E}[\|J_x\|^t] \leq c^{t/2} n_0^{-t/2} \mathbb{E}[\chi_{n_L}^t] \prod_{l=1}^{L-1} \mathbb{E}\left[\left(\frac{c}{n_l}\sum_{i=1}^{n_l} m_{n_{l-1},i}^{(K)}\right)^{t/2}\right],$$

*where $J_x$ is the Jacobian as defined above, $x \in \mathbb{R}^{n_0}$; $t \geq 1, t \in \mathbb{N}$; $m_{n_{l-1},i}^{(K)}$ is the largest order statistic in a sample of size $K$ of $\chi_{n_{l-1}}^2$ variables. Recall that the largest order statistic is a random variable defined as the maximum of a random sample, and that a sum of squares of $n$ independent Gaussian variables has a chi-squared distribution $\chi_n^2$.*

*Proof.* Our first goal will be to upper-bound $\|J_x\| = \sup_{\|u\|=1} \|J_x u\|$. The Jacobian $J_x$ of $\mathcal{N}(x)\colon \mathbb{R}^{n_0} \to \mathbb{R}^{n_L}$ can be written as a product of matrices $\overline{W}^{(l)}$, $l = 1,\ldots,L$ depending on the activation region of the input $x$. The matrix $\overline{W}^{(l)}$ consists of rows $\overline{W}_i^{(l)} = W_{i,k_i}^{(l)} \in \mathbb{R}^{n_{l-1}}$, where $k_i = \text{argmax}_{k \in [K]}\{W_{i,k}^{(l)} x^{(l-1)} + b_{i,k}^{(l)}\}$ for $i = 1,\ldots,n_l$, and $x^{(l-1)}$ is the $l$-th layer's input. For the last layer, which is linear, we have $\overline{W}^{(L)} = W^{(L)}$. Thus for any given $u \in \mathbb{R}^{n_0}$ we have

$$\|J_x u\| = \|W^{(L)} \overline{W}^{(L-1)} \cdots \overline{W}^{(1)} u\|.$$

Consider some $u^{(0)}$ with $\|u^{(0)}\| = 1$ and assume $\|\overline{W}^{(1)} u^{(0)}\| \neq 0$. Note that for fixed $u^{(0)}$, the probability of $\overline{W}^{(1)}$ being such that $\|\overline{W}^{(1)} u^{(0)}\| = 0$ is 0. Multiplying and dividing by $\|\overline{W}^{(1)} u^{(0)}\|$

we get

$$\|W^{(L)}\overline{W}^{(L-1)}\cdots\overline{W}^{(1)}u^{(0)}\|\frac{\|\overline{W}^{(1)}u^{(0)}\|}{\|\overline{W}^{(1)}u^{(0)}\|}$$

$$=\left\|W^{(L)}\overline{W}^{(L-1)}\cdots\overline{W}^{(2)}\frac{\overline{W}^{(1)}u^{(0)}}{\|\overline{W}^{(1)}u^{(0)}\|}\right\|\|\overline{W}^{(1)}u^{(0)}\|$$

$$=\left\|W^{(L)}\overline{W}^{(L-1)}\cdots\overline{W}^{(2)}u^{(1)}\right\|\|\overline{W}^{(1)}u^{(0)}\|,$$

where $u^{(1)} = \frac{\overline{W}^{(1)}u^{(0)}}{\|\overline{W}^{(1)}u^{(0)}\|}$. Notice, $\|u^{(1)}\| = 1$. Repeating this procedure layer-by-layer, we get

$$\|W^{(L)}u^{(L-1)}\|\|\overline{W}^{(L-1)}u^{(L-2)}\|\cdots\|\overline{W}^{(2)}u^{(1)}\|\|\overline{W}^{(1)}u^{(0)}\|.$$

Now consider one of the factors, $\|\overline{W}^{(l)}u^{(l-1)}\|$. We have

$$\|\overline{W}^{(l)}u^{(l-1)}\|^2 = \sum_{i=1}^{n_l}\langle\overline{W}_i^{(l)}, u^{(l-1)}\rangle^2 \overset{\overset{\text{Cauchy–Schwarz}}{\|u^{(l-1)}\|=1}}{\leq} \sum_{i=1}^{n_l}\|\overline{W}_i^{(l)}\|^2 \leq \sum_{i=1}^{n_l}\max_{k\in[K]}\left\{\|W_{i,k}^{(l)}\|^2\right\}.$$

Notice that this upper bound only depends on $W^{(l)}$ and is independent of all other weight matrices and of the input vector.

According to our assumptions, $W_{i,k}^{(l)} \overset{d}{=} \sqrt{\frac{c}{n_{l-1}}}v$, where $v$ is a standard Gaussian random vector in $\mathbb{R}^{n_{l-1}}$. Therefore, $\|W_{i,k}^{(l)}\|^2 \overset{d}{=} \frac{c}{n_{l-1}}\chi_{n_{l-1}}^2$ has the distribution of a chi-squared random variable scaled by $c/n_{l-1}$. Moreover, since the vectors $W_{i,1}^{(l)},\ldots,W_{i,K}^{(l)}$ consist of the same number of separate iid entries, the variables $\|W_{i,1}^{(l)}\|^2,\ldots,\|W_{i,K}^{(l)}\|^2$ are iid. In turn, $\max_{k\in[K]}\left\{\|W_{i,k}^{(l)}\|^2\right\} \overset{d}{=} \frac{c}{n_{l-1}}m_{n_{l-1},i}^{(K)}$, where $m_{n_{l-1},i}^{(K)}$ is the largest order statistic in a sample of size $K$ of $\chi_{n_{l-1}}^2$ variables.

Notice that $\|W^{(L)}u^{(L-1)}\|^2 \overset{d}{=} \frac{c}{n_{L-1}}\chi_{n_L}^2$. To see this, recall that if $u$ is a fixed vector and $w$ is a Gaussian random vector with mean $\mu$ and covariance matrix $\Sigma$, then the product $u^\top w$ is Gaussian with mean $u^\top\mu$ and variance $u^\top\Sigma u$. Hence, since $W_i^{(L)}$ is a Gaussian vector with mean zero and covariance matrix $\Sigma = \frac{c}{n_{L-1}}I$, $W_i^{(L)}u^{(L-1)}$ is Gaussian with mean zero and variance $\frac{c}{n_{L-1}}\|u^{(L-1)}\|^2 = \frac{c}{n_{L-1}}$.

Combining everything, we get

$$\|J_x\| = \sup_{\|u\|=1}\|J_x u\| \leq \left(\frac{c}{n_{L-1}}\chi_{n_L}^2\right)^{1/2}\left(\frac{c}{n_{L-2}}\sum_{i=1}^{n_{L-1}}m_{n_{L-2}}^{(K)}\right)^{1/2}\cdots\left(\frac{c}{n_0}\sum_{i=1}^{n_1}m_{n_0}^{(K)}\right)^{1/2}$$

$$=c^{L/2}\chi_{n_L}\left(\prod_{l=0}^{L-1}n_l^{-1/2}\right)\prod_{l=1}^{L-1}\left(\sum_{i=1}^{n_l}m_{n_{l-1},i}^{(K)}\right)^{1/2}.$$

Now using the monotonicity of the expectation, the moments of the right hand side upper-bound those of the left hand side. Moreover, using the independence of the individual factors, the expectation factorizes. For the $t$-th moment we get

$$\mathbb{E}[\|J_x\|^t] \leq \mathbb{E}\left[c^{tL/2}\chi_{n_L}\left(\prod_{l=0}^{L-1}n_l^{-1/2}\right)\prod_{l=1}^{L-1}\left(\sum_{i=1}^{n_l}m_{n_{l-1},i}^{(K)}\right)^{t/2}\right]$$

$$= c^{t/2}n_0^{-t/2}\mathbb{E}[\chi_{n_L}^t]\prod_{l=1}^{L-1}\mathbb{E}\left[\left(\frac{c}{n_l}\sum_{i=1}^{n_l}m_{n_{l-1},i}^{(K)}\right)^{t/2}\right].$$

$\square$

**Corollary 36** (Upper bound on $C_{\text{grad}}$). *Under the same assumptions as in Proposition 35, assuming that $c$ is set according to He initialization, meaning $c = 2$, or maxout-He initialization (see Table 1 for specific values of $c$ for various $K$), the following expression can be used as the value for $C_{grad}$:*

$$\left(\frac{c}{n_0}\right)^{1/2} \left(n_L(n_L + t)^{\frac{t}{2}-1}\right)^{1/t} \prod_{l=1}^{L-1} \left(\mathbb{E}\left[\left(\frac{c}{n_l}\sum_{i=1}^{n_l} m_{n_{l-1},i}^{(K)}\right)^{t/2}\right]\right)^{1/t},$$

*where $m_{n_{l-1},i}^{(K)}$ is the largest order statistic in a sample of size $K$ of $\chi_{n_{l-1}}^2$ variables.*

*Proof.* The constant $C_{\text{grad}}$ was defined as an upper bound on

$$\left(\sup_{x\in\mathbb{R}^{n_0}} \mathbb{E}[\|\nabla\zeta_{z,k}(x)\|^t]\right)^{1/t}.$$

Therefore, using the upper-bound on the moments of the Jacobian norm from Proposition 35, an upper-bound on the following expression can be used as a value for $C_{\text{grad}}$:

$$c^{1/2}n_0^{-1/2}\left(\mathbb{E}[\chi_{n_L}^t]\right)^{1/t}\prod_{l=1}^{L-1}\left(\mathbb{E}\left[\left(\frac{c}{n_l}\sum_{i=1}^{n_l} m_{n_{l-1},i}^{(K)}\right)^{t/2}\right]\right)^{1/t}.$$

The moments of the chi distribution are

$$\mathbb{E}\left[\chi_{n_L}^t\right] = 2^{t/2}\frac{\Gamma((n_L + t)/2)}{\Gamma(n_L/2)}.$$

Using an upper-bound on a Gamma function ratio (see Jameson, 2013, Equation 12), this can be upper-bounded with

$$n_L(n_L + t)^{\frac{t}{2}-1}.$$

The factor involving $m_{n_{l-1}}^{(K)}$ can be upper-bounded by considering the explicit expression for the moments of the largest order statistic of chi-squared variables. The closed form for these moments is available (see Nadarajah, 2008), but they have complicated form and we will keep the factor involving $m_{n_{l-1}}^{(K)}$ as it is. Then the total upper bound is

$$\left(\frac{c}{n_0}\right)^{1/2}\left(n_L(n_L + t)^{\frac{t}{2}-1}\right)^{1/t}\prod_{l=1}^{L-1}\left(\mathbb{E}\left[\left(\frac{c}{n_l}\sum_{i=1}^{n_l} m_{n_{l-1},i}^{(K)}\right)^{t/2}\right]\right)^{1/t}.$$

$\square$

Estimating the moments of the gradient of maxout networks is a challenging topic, as can be seen from the above discussion, and is worthy of a separate investigation. It might be possible to obtain tighter upper-bounds on it and on $C_{\text{grad}}$, a question that we leave for future work.

## G Expected number of regions for networks without bias

Zero biases of ReLU networks were discussed in Hanin and Rolnick (2019b) and studied in detail in Steinwart (2019). There is no distribution on the biases in the zero bias case, meaning that conditions on the biases from Theorem 9 are not satisfied. We closely follow the proofs in Hanin and Rolnick (2019b) and show that the arguments similar to those regarding the zero bias case in the ReLU networks also apply to the maxout networks. According to Lemma 37, activation regions of zero-bias maxout networks are convex cones, see Figure 9 for the illustration. In Corollary 39 we come to a conclusion that the number of activation regions in expectation in a network with zero biases grows as $O(n_0(KN)^{n_0-1}\binom{K(n_0-1)}{2(n_0-1)})$.

**Lemma 37.** *Let $\mathcal{N}$ be a maxout network with biases set to zero. Then,*

| Zero bias | Small bias | Non-zero bias |

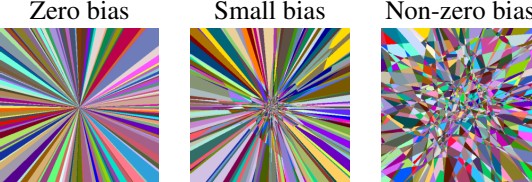

Figure 9: Linear regions of a 3 layer network with 100 units and the maxout rank $K = 2$. The network was initialized with the maxout-He distribution. Activation regions of a maxout network with zero biases are convex cones. Small biases are initialized as the biases sampled from the maxout-He distribution multiplied by $0.1$. Majority of linear regions of a network with small biases are cones and the ones that are not are small and concentrated around zero.

(a) $\mathcal{N}$ is nonnegative homogeneous : $\mathcal{N}(cx) = c\mathcal{N}(x)$ for each $c \geq 0$.

(b) For every activation region $\mathcal{R}$ of $\mathcal{N}$, and every point $x$ in $\mathcal{R}$, all points $cx$ are also in $\mathcal{R}$ for $c > 0$ and $\mathcal{R}$ is a convex polyhedral cone.

*Proof of Lemma 37.* Each neuron of the network computes a function of the form $z(x_1, \ldots, x_n) = \max_{k \in [K]} \left\{ \sum_{i=1}^{n} w_{i,k} \cdot x_i \right\}$. Note that for any $c \geq 0$:

$$z(cx_1, \ldots, cx_n) = \max_{k \in [K]} \left\{ c \sum_{i=1}^{n} w_{i,k} \cdot x_i \right\} = c \cdot \max_{k \in [K]} \left\{ \sum_{i=1}^{n} w_{i,k} \cdot x_i \right\} = c \cdot z(x_1, \ldots, x_n).$$

Therefore, each neuron is equivariant under multiplication by a nonnegative constant $c$, and thus the overall network as well, proving (a). If $c > 0$, the activation patterns for $x$ and $cx$ are also identical, since for any inequality in the activation region definition we have

$$\sum_{i=1}^{n} w_{i,j} \cdot cx_i > \sum_{i=1}^{n} w_{i,j'} \cdot cx_i \iff \sum_{i=1}^{n} w_{i,j} \cdot x_i > \sum_{i=1}^{n} w_{i,j'} \cdot x_i, \quad j, j' \in [K].$$

This implies that $x$ and $cx$ lie in the same activation region, and that $\mathcal{R}$ is a convex polyhedral cone, see e.g. Chandru and Hooker (2011). This proves (b). $\qquad\square$

**Proposition 38** (Networks without biases do not have more regions). *Suppose that $\mathcal{N}$ is a maxout network with biases and conditions from Theorem 9 are satisfied. Let $\mathcal{N}_0$ be the same network with all biases set to $0$. Then, the total number of activation regions (in all the input space) for $\mathcal{N}_0$ is no more than that for $\mathcal{N}$.*

*Proof of Proposition 38.* We define an injective mapping from activation regions of $\mathcal{N}_0$ to regions of $\mathcal{N}$. For each region $\mathcal{R}$ of $\mathcal{N}_0$, pick a point $x_\mathcal{R} \in \mathcal{R}$. By Lemma 37, $cx_\mathcal{R} \in \mathcal{R}$ for each $c > 0$. Let $\mathcal{N}_{1/c}$ be the network obtained from $\mathcal{N}$ by dividing all biases by $c$, and observe that $\mathcal{N}(cx_\mathcal{R}) = c\mathcal{N}_{1/c}(x_\mathcal{R})$, with the same activation pattern between the two networks.

By picking $c$ sufficiently large, $\mathcal{N}_{1/c}$ becomes arbitrarily close to $\mathcal{N}_0$. Therefore, for some sufficiently large $c$, $\mathcal{N}_0(cx_\mathcal{R})$ and $\mathcal{N}(cx_\mathcal{R})$ have the same pattern of activations. Regions of $\mathcal{N}$ in which $cx_\mathcal{R}$ lies are distinct for all distinct $\mathcal{R}$. Thus, the number of regions of $\mathcal{N}$ is at least as large as the number of regions of $\mathcal{N}_0$. $\qquad\square$

We obtain following corollary of Theorem 9 for the zero-bias case.

**Corollary 39** (Expected number of activation regions of zero-bias networks). *Suppose that $\mathcal{N}_0$ is a fully-connected feed-forward maxout network with zero biases, $n_0$ inputs, a total of $N$ rank $K$ maxout units. Also, suppose that all conditions from Theorem 9, except for the conditions on the biases, are satisfied. Then there exists a constant $T'$ depending on $C_{grad}$ so that*

$$\mathbb{E}[\#\text{activation regions of } \mathcal{N}_0] \leq \begin{cases} K^N, & N \leq n_0 \\ 2n_0 \dfrac{(T'KN)^{n_0-1}\binom{K(n_0-1)}{2(n_0-1)}}{(n_0-1)!}, & N \geq n_0 \end{cases}.$$

*The expectation is taken with respect to the distribution of weights in $\mathcal{N}_0$.*

*Proof of Corollary 39.* Based on Proposition 38 we can use the same upper bound as for the networks with biases, thus for the case $N \leq n_0$, the expectation is upper bounded with $K^N$.

Now consider the case $N \geq n_0$. We will add biases to $\mathcal{N}_0$ in such a way that the bias conditions of Theorem 9 are satisfied with some $C'_{\text{bias}}$. Denote the resulting network with $\mathcal{N}$. Then, by Proposition 38, $\mathcal{N}$ has a region corresponding to each region of $\mathcal{N}_0$. All the corresponding regions in $\mathcal{N}$ are unbounded because according to Proposition 38 for any $x_\mathcal{R}$ from a region of $\mathcal{N}_0$ there exists a constant $c > 0$ so that $cx_\mathcal{R}$ belongs to a region in $\mathcal{N}$. Since all regions in $\mathcal{N}_0$ are unbounded, all corresponding regions in $\mathcal{N}$ are unbounded under such a mapping.

Therefore, to obtain the result, it is enough to upper-bound the number of unbounded activation regions of $\mathcal{N}$. Similarly to the proof of Theorem 9, consider a hypercube with a side length $\delta > \delta_0$, large enough to interest all the unbounded regions. Then the total number of unbounded activation regions of $\mathcal{N}$ is upper bounded by the sum of the numbers of activation regions intersecting each of the hypercube $2n_0$ facets, each of dimension $(n_0 - 1)$. By Theorem 9, the expected number of activation regions of $\mathcal{N}$ in $\mathbb{R}^{n_0-1}$ is upper bounded with $(\delta 2^5 C_{\text{grad}} C'_{\text{bias}} KN)^{n_0-1} \binom{K(n_0-1)}{2(n_0-1)}/(n_0-1)!$. Denoting $\delta 2^5 C_{\text{grad}} C'_{\text{bias}}$ with $T'$ and combining everything we get the desired result. $\qquad\square$

# H  Proofs related to the decision boundary

## H.1  Simple upper bound on the number of pieces of the decision boundary

A network used for multi-class classification into $M \in \mathbb{N}, M \geq 2$ classes can be seen as a network with a rank $M$ maxout unit on top. Therefore, to discuss the decision boundary, we consider $r$-partial activation regions, $r \geq 1$, with at least one equation in the last unit. With $J^r_{\text{DB}}$, we denote the $r$-partial activation patterns corresponding to such regions and with $\mathcal{X}_{\text{DB},r} := \bigcup_{J^r_{\text{DB}} \in \mathcal{P}_{\text{DB},r}} \mathcal{R}(J^r_{\text{DB}}; \theta)$ their union. All decision boundary is then written as $\mathcal{X}_{DB}$.

**Lemma 40** (Simple upper bound on the number of $r$-partial activation patterns of the decision boundary)**.** *Let $r \in \mathbb{N}_+$. The number of $r$-partial activation patterns in the decision boundary of a network with a total of $N$ rank-$K$ maxout units is upper bounded by $|\mathcal{P}_{DB,r}| \leq \sum_{i=1}^{\min\{M-1,r\}} \binom{M}{i+1}\binom{K(r-i)}{2(r-i)}\binom{N}{r-i}K^{N-r+i}$. The number of $r$-partial activation sub-patterns is upper bounded by $|\mathcal{S}_{DB,r}| \leq \sum_{i=1}^{\min\{M-1,r\}} \binom{M}{i+1}\binom{K(r-i)}{2(r-i)}\binom{N}{r-i}$.*

*Proof of Lemma 40.* Activation patterns for the decision boundary regions should have at least one equality in the upper unit. At the same time, the maximum possible number of equations in the last unit is $\min\{M-1, r\}$. To get all suitable activation patterns we need to sum over all these options.

Now consider a fixed number of equations $i \in \{1, \ldots, \min\{M-1, r\}\}$. The number of ways to choose them is $\binom{M}{i+1}$ and the number of options for the all other units in the network is given by Lemma 6 for $r - i$. Combining everything, we get the claimed statement. $\qquad\square$

## H.2  Lower bound on the maximum number of pieces of the decision boundary

The lower bound in the second item of Theorem 21 is based on a construction of parameters for which the network maps an $n$-cube in the input space to an $n$-cube in the output space in many-to-one fashion. This means that any feature implemented over the last layer will replicate multiple times over the input layer. We infer the following lower bound on the maximum number of pieces of the decision boundary of a maxout network.

**Proposition 41** (Lower bound on the maximum number of pieces of the decision boundary)**.** *Consider a network $\mathcal{N}$ with $n_0$ inputs and $L$ layers of $n_1, \ldots, n_L$ rank-$K$ maxout units followed by an $M$-class classifier. Suppose $n \leq n_0$, $\frac{n_l}{n}$ even, and $e_l = \min\{n_0, \ldots, n_{l-1}\}$. Denote by $N(M, n)$ the maximum number of boundary pieces implemented by an $M$-class classifier over an $n$-cube. Then the maximum number of linear pieces of the decision boundary of $\mathcal{N}$ is lower bounded by $N(M, n)\prod_{l=1}^{L}(\frac{n_l}{n}(K-1)+1)^n$. If $n \geq M$ or $n \geq 4$, $N(M, n) = \binom{M}{2}$.*

The asymptotic order of this bound is $\Omega(M^2 \prod_{l=1}^{L}(n_l K)^{n_0})$.

*Proof.* We use the construction of parameters from Montúfar et al. (2021, Proposition 3.11) refining a previous construction for ReLU networks (Montúfar et al., 2014) to have the network represent a many-to-one map. There are $\prod_{l=1}^{L}(\frac{n_l}{n}(K-1)+1)^n$ distinct linear regions whose image in the output space of the last layer contains an $n$-cube. The linear pieces of the decision boundary of an $M$-class classifier over an $n$-cube at the $L$-th layer will have a corresponding multiplicity over the input space. An $M$-class classifier is implemented as $\mathbb{R}^M \to [M]; y = (y_1, \dots, y_M) \mapsto \operatorname{argmax}_{r \in [M]} y_r$. This has $\binom{M}{2}$ boundaries, one between any two classes. If $n \geq M$, then the image of the preceding layers intersects all of these boundaries. More generally, the number of boundary pieces of an $M$-class classifier over $n$-dimensional space can be seen to correspond to the number of edges of a polytope with $M$ vertices in $n$-dimensional space. The trivial upper bound $N(M, n) \leq \binom{M}{2}$ is attained if $1 < \lfloor \frac{n}{2} \rfloor$. This follows form the celebrated Upper Bound Theorem for the maximum number of faces of convex polytopes (McMullen, 1970). $\qquad\square$

## H.3 Upper bound on the expected volume of the decision boundary

**Theorem 12** (Upper bound on the volume of the $(n_0 - r)$-skeleton of the decision boundary)**.** *Consider a bounded measurable set $S \subset \mathbb{R}^{n_0}$. Consider the notation and assumptions of Theorem 9, whereby the constants $C_{grad}$ and $C_{bias}$ are over $S$. Then, for any $r \in \{1, \dots, n_0\}$ we have*

$$\frac{\mathbb{E}[\mathrm{vol}_{n_0-r}(\mathcal{X}_{\mathrm{DB},r} \cap S)]}{\mathrm{vol}_{n_0}(S)} \leq (2C_{\mathrm{grad}}C_{\mathrm{bias}})^r \sum_{i=1}^{\min\{M-1,r\}} \binom{M}{i+1}\binom{K(r-i)}{2(r-i)}\binom{N}{r-i}.$$

*Proof of Theorem 12.* Using Lemma 31, but considering only $r$-partial activation patterns that belong to the decision boundary, volume of the $(n_0 - r)$-skeleton of the decision boundary can be upper-bounded with

$$\sum_{\hat{J}_{\mathrm{DB}}^r} \int_S \mathbb{E}\left[\rho_{\mathbf{b}^r}((\mathbf{w}^m - \mathbf{w}^r) \cdot \mathbf{x}_{-1}^m + \mathbf{b}^m) \; \|\mathbf{J}((\mathbf{w}^m - \mathbf{w}^r) \cdot \mathbf{x}_{-1}^m + \mathbf{b}^m)\|\right] \; dx.$$

Upper-bounding the integral as in Theorem 10, but using Lemma 40 to count the number of entries in the sum, we get the final upper-bound

$$(2C_{\mathrm{grad}}C_{\mathrm{bias}})^r \sum_{i=1}^{\min\{M-1,r\}} \binom{N}{r-i}\binom{K(r-i)}{2(r-i)}\binom{M}{i+1} \mathrm{vol}_{n_0}(S).$$

$\qquad\square$

## H.4 Upper bound on the expected number of pieces of the decision boundary

**Lemma 42** (Upper bound on the expected number of $r$-partial activation regions of the decision boundary)**.** *Let $\mathcal{N}$ be a fully-connected feed-forward maxout network, with $n_0$ inputs, a total of $N$ rank $K$ maxout units, and $M$ linear output units used for multi-classification. Fix $r \in \{1, \dots, n_0\}$. Then, under the assumptions of Theorem 9, there exists $\delta_0 \leq 1/(2C_{grad}C_{bias})$ such that for all cubes $\mathcal{C} \subseteq \mathbb{R}^{n_0}$ with side length $\delta > \delta_0$,*

$$\frac{\mathbb{E}\left[\begin{smallmatrix}\# \ r\text{-partial activation regions in} \\ \text{the decision boundary of } \mathcal{N} \text{ in } \mathcal{C}\end{smallmatrix}\right]}{\mathrm{vol}(\mathcal{C})} \leq \begin{cases} \sum_{i=1}^{\min\{M-1,r\}} \binom{M}{i+1}\binom{K(r-i)}{2(r-i)}\binom{N}{r-i}K^{N-r+i}, & N \leq n_0 \\[2ex] \frac{(2^4 C_{\mathrm{grad}}C_{\mathrm{bias}}N)^{n_0}(2K)^{n_0-1}}{n_0!} \\ \times \sum_{i=1}^{\min\{M-1,n_0\}} \binom{M}{i+1}\binom{K(n_0-i)}{2(n_0-i)}\frac{\prod_{j=1}^{i}(n_0-j+1)}{\prod_{j=1}^{i}(N-1+j)}, & N \geq n_0 \end{cases}.$$

*Here the expectation is taken with respect to the distribution of weights and biases in $\mathcal{N}$.*

*Proof of Lemma 42.* Result for the case $N \leq n_0$ arises from Lemma 40. Consider $N \geq n_0$. The proof closely follows the proof of Theorem 10, and we only highlight the differences. Based on Lemma 12,

$$\mathbb{E}\left[\# \mathcal{V}_t\right] \leq (2C_{\mathrm{grad}}C_{\mathrm{bias}})^t \sum_{i=1}^{\min\{M-1,t\}} \binom{N}{t-i}\binom{K(t-i)}{2(t-i)}\binom{M}{i+1} \mathrm{vol}_t(\mathcal{C}_t).$$

Therefore, the upper bound on the expected number of $r$-partial activation regions in the decision boundary is

$$\sum_{t=r}^{n_0} \binom{t}{r} K^{t-r} (2C_{\text{grad}}C_{\text{bias}})^t \sum_{i=1}^{\min\{M-1,t\}} \binom{N}{t-i} \binom{K(t-i)}{2(t-i)} \binom{M}{i+1} \binom{n_0}{t} 2^{n_0-t} \delta^t$$

$$\leq (4\delta C_{\text{grad}}C_{\text{bias}})^{n_0} (2K)^{n_0-r} \sum_{i=1}^{\min\{M-1,n_0\}} \binom{M}{i+1} \binom{K(n_0-i)}{2(n_0-i)} \sum_{t=r}^{n_0} \binom{N}{t-i} \binom{n_0}{t}$$

Re-writing $\binom{N}{t-i} \binom{n_0}{t}$ as $\binom{n_0}{t}^2 \frac{\binom{N}{t-i}}{\binom{n_0}{t}}$ we can upper-bound it with

$$4^{n_0} \frac{\prod_{j=1}^{i}(t-j+1)}{\prod_{j=1}^{i}(N-t+j)} \frac{N^{n_0}}{n_0!} \leq 4^{n_0} \frac{\prod_{j=1}^{i}(n_0-j+1)}{\prod_{j=1}^{i}(N-r+j)} \frac{N^{n_0}}{n_0!}.$$

The final upper bound is then

$$\frac{(2^5 C_{\text{grad}}C_{\text{bias}}KN)^{n_0}}{(2K)^r n_0!} \sum_{i=1}^{\min\{M-1,n_0\}} \binom{M}{i+1} \binom{K(n_0-i)}{2(n_0-i)} \frac{\prod_{j=1}^{i}(n_0-j+1)}{\prod_{j=1}^{i}(N-r+j)} \text{vol}(C).$$

Dividing this expression by $\text{vol}(C)$ we get the desired result. $\qquad\square$

The next theorem follows immediately from Lemma 42 if $r$ is set to 1.

**Theorem 11** (Upper bound on the expected number of linear pieces of the decision boundary). *Let $\mathcal{N}$ be a fully-connected feedforward maxout network, with $n_0$ inputs, a total of $N$ rank-$K$ maxout units, and $M$ linear output units used for multi-class classification. Under the assumptions of Theorem 9, there exists $\delta_0 \leq 1/(2C_{\text{grad}}C_{\text{bias}})$ such that for all cubes $C \subseteq \mathbb{R}^{n_0}$ with side length $\delta > \delta_0$,*

$$\frac{\mathbb{E}\left[\substack{\# \text{ linear pieces in the} \\ \text{decision boundary of } \mathcal{N} \text{ in } C}\right]}{\text{vol}(C)} \leq \begin{cases} \binom{M}{2} K^N, & N \leq n_0 \\ \frac{(2^4 C_{\text{grad}}C_{\text{bias}})^{n_0} (2KN)^{n_0-1}}{(n_0-1)!} \binom{M}{2} \binom{K(n_0-1)}{2(n_0-1)}, & N \geq n_0 \end{cases}.$$

*Here the expectation is taken with respect to the distribution of weights and biases in $\mathcal{N}$.*

### H.5   Lower bound on the expected distance to the decision boundary

Now, using an approach similar to Hanin and Rolnick (2019a, Corollary 5), who provided a lower bound on the expected distance to the boundary of linear regions, we discuss a lower bound on the distance to the decision boundary. We will use the following result from that work.

**Lemma 43** (Hanin and Rolnick 2019a, Lemma 12). *Fix a positive integer $n \geq 1$, and let $Q \subseteq \mathbb{R}^n$ be a compact continuous piecewise linear submanifold with finitely many pieces. Define $Q_0 = \emptyset$ and let $Q_t$ be the union of the interiors of all $k$-dimensional pieces of $Q \setminus (Q_0 \cup \cdots \cup Q_{t-1})$. Denote by $T_\varepsilon(X)$ the $\varepsilon$-tubular neighborhood of any $X \subset \mathbb{R}^n$. We have $\text{vol}_n(T_\varepsilon(Q)) \leq \sum_{t=0}^{n} \omega_{n-t} \varepsilon^{n-t} \text{vol}_k(Q_t)$, where $\omega_d :=$ volume of ball of radius 1 in $\mathbb{R}^d$.*

We will prove the following.

**Corollary 13** (Distance to the decision boundary). *Suppose $\mathcal{N}$ is as in Theorem 9. For any compact set $S \subset \mathbb{R}^{n_0}$ let $x$ be a uniform point in $S$. There exists $c > 0$ independent of $S$ so that*

$$\mathbb{E}[\text{distance}(x, \mathcal{X}_{\text{DB}})] \geq \frac{c}{2C_{\text{grad}}C_{\text{bias}}M^{m+1}m},$$

*where $m := \min\{M-1, n_0\}$.*

*Proof of Corollary 13.* Let $x \in K$ be uniformly chosen. Then, for any $\varepsilon > 0$, using Markov's inequality and Lemma 43, we have

$$\mathbb{E}[\text{distance}(x, \mathcal{X}_{DB})] \geq \varepsilon P(\text{distance}(x, \mathcal{X}_{DB}) > \varepsilon) = \varepsilon(1 - P(\text{distance}(x, \mathcal{X}_{DB}) \leq \varepsilon))$$

$$= \varepsilon \left(1 - \mathbb{E}\left[\text{vol}_{n_0}(T_\varepsilon(\mathcal{X}_{DB}))\right]\right) \geq \varepsilon \left(1 - \sum_{t=1}^{n_0} \omega_{n_0-t} \varepsilon^{n_0-t} \mathbb{E}\left[\text{vol}_{n_0-t}(\mathcal{X}_{DB})\right]\right)$$

The upper bound from Theorem 12 can be upper bounded further with

$$\mathbb{E}[\text{vol}_{n_0-t}(\mathcal{X}_{\text{DB},t} \cap S)] \leq (2C_{\text{grad}}C_{\text{bias}})^t \sum_{i=1}^{\min\{M-1,t\}} \binom{M}{i+1} \binom{K(t-i)}{2(t-i)} \binom{N}{t-i} \text{vol}_{n_0}(S)$$

$$\leq (2C_{\text{grad}}C_{\text{bias}})^t (4K^2 N)^{t-1} M^{m^*+1} m^* \text{vol}_{n_0}(S),$$

where $m^* := \min\{M-1, t\}$. Then expectation of the distance can be lower bounded with

$$\varepsilon \left(1 - \sum_{t=1}^{n_0} (2C_{\text{grad}}C_{\text{bias}}\varepsilon)^t (4\varepsilon K^2 N)^{t-1} M^{m^*+1} m^*\right) \geq \varepsilon \left(1 - 2C_{\text{grad}}C_{\text{bias}} M^{m+1} m\varepsilon\right),$$

where $m := \min\{M-1, n_0\}$. Taking $\varepsilon$ to be a small constant $c$ times $1/(2C_{\text{grad}}C_{\text{bias}}M^{m+1}m)$ completes the proof. $\qquad\square$

**Remark 44** (Decision boundary of ReLU networks). All proofs consider the indecision locus of the last unit on top of the network and reuse results on the volume of the boundary and the number of activation regions. If one sets $K$ to 2, these results differ only in $2^{-r}$ from those for the ReLU networks. Therefore, the decision boundary analysis should also apply to the ReLU networks if one sets $K$ to 2 with a difference only in the constant.

# I Counting algorithms

## I.1 Approximate counting of the activation regions

First, we describe an approximate method for counting linear regions that is useful for quickly estimating the number of linear regions or plotting them.

We generate a grid of inputs in an $n_0$-dimensional cube, compute the gradients with respect to the input, which is simply a product of weights on the path that corresponds to a given input, and then sum the gradient values for each input dimension of one input. Then, we compute the number of unique sums and use it as the number of linear regions.

The method is not exact because it works by computing network gradients on a grid, so it is possible to miss a small region. Also, it does not distinguish between regions with the same gradient value, which is one more reason it might miss some linear regions and why it counts linear regions, not activation regions. However, from what we have seen, if the grid has many points, the difference between the exact and approximate method is not that big.

## I.2 Exact counting of the activation regions

The algorithm starts with a cube in which we want to count the activation regions defined with a set of linear inequalities in $\mathbb{R}^{n_0}$. We go through the network layer by layer, unit by unit, and for each unit, we determine if its pre-activation features attain a maximum on the regions obtained so far by checking the feasibility of the corresponding linear inequalities systems. For this, we use linear programming. More specifically, an interior-point method implementation from `scipy.optimize.linprog`. The use of linear programming is justified since, according to Lemma 4, the activation regions are convex.

The input to the simplex method becomes the combined system of inequalities for the region and the pre-activation feature. We set the objective to zero, meaning that any $x$ can satisfy it. One has to use non-strict inequalities in linear programming methods, implying the boundary of activation regions is also included. We also add a small $\varepsilon = 1e-6$ to avoid zero solutions in a zero bias case. The inequalities for a pre-activation feature of some neuron $z$ have the form

$$\{x \in \mathbb{R}^{n_0} \mid a_{z,j_0}(x;\theta) + b_{z,j_0} \geq a_{z,i}(x;\theta) + b_{z,i} + \varepsilon, \quad \forall i \in [K]\backslash[j_0]\}.$$

As a result, we get a new list of activation regions and pass it to the next unit.

To correctly estimate inequalities corresponding to a pre-activation feature on a specific region, one has to keep track of the function computed on this region, which has the form: $w_J^{(l)} \ldots (w_J^{(0)} \cdot x + b_J^{(0)}) + \cdots + b_J^{(l)}$, where $J$ is an activation pattern of the region.

The pseudocode for the algorithm is in Algorithm 1, and the pseudocode for a check for one pre-activation feature is in Algorithm 2.

---

**Algorithm 1** Exactly Count the Number of Activation Regions in a Maxout Network

---

1: **function** COUNTACTIVATIONREGIONS
2:     `activation_regions = [starting_cube]`
3:     **for** `layer` in $\{1, \ldots, L\}$ **do**
4:         **for** `unit` in `layer` **do**
5:             `new_activation_regions = []`
6:             **for** `region` in `activation_regions` **do**
7:                 **for** `feature` in `unit` **do**
8:                     ▷ See Algorithm 2
9:                     **if** NewRegionCheck(`unit.features`, `feature`, `region`) **then**
10:                        `new_activation_regions.append(new_region)`
11:            `activation_regions = new_activation_regions`
12:        **for** `region` in `activation_regions` **do**
13:            `region.function = region.next_layer_function`
14:            `region.next_layer_function = []`
15:    **return** `length(activation_regions)`

---

### I.3 Exact counting of linear pieces in the decision boundary

We define an algorithm for exactly counting linear pieces in the decision boundary based on the algorithm from Section I.2. Consider a classification problem with $M$ classes, and to describe the decision boundary, add a maxout unit of rank $M$ on top of the network. To count the number of linear pieces in the decision boundary, for each pair of classes, go through all the activation regions of the network. Construct a linear program for which the set of inequalities is given by a union of the region inequalities and inequalities which determine if the given classes attain maximum. Also, add the equality between these two classes. If the problem is feasible, there is a piece in the decision boundary. At the end of this process, one gets the total number of linear pieces in the decision boundary.

### I.4 Algorithm discussion

There are two useful modifications to the method. First, to count the number of regions in a ReLU network instead of systems of $(K-1)$ linear inequalities, one can use inequalities of the form $w \cdot x + b \geq 0$ and $w \cdot x + b \leq 0$.

Second, to compute the number of activation regions in a slice, one can define a parametrization of the input space. We consider as the slice of a cube $\mathcal{C}$ the 2-space through three points $x_1, x_2, x_3 \in \mathbb{R}^{n_0}$, meaning the slice has the form $V = \{x = v_0 + v_1 y_1 + v_2 y_2 \in \mathbb{R}^{n_0} : (y_1, y_2) \in \mathbb{R}^2 \cap \mathcal{C}\}$, where $v_0 = (x_1 + x_2 + x_3)/3 \in \mathbb{R}^{n_0}$, and $v_1, v_2 \in \mathbb{R}^{n_0}$ are an orthogonal basis of $\mathrm{span}\{x_2 - x_1, x_3 - x_1\}$, and $v_1, v_2$ are orthonormal. We can evaluate the network function over such a slice by augmenting the network by a linear layer $\phi \colon \mathbb{R}^2 \to \mathbb{R}^{n_0}$ with weights $v_1, v_2$ and biases $v_0$. We used images from 3 different classes as the points that define the slice.

We usually performed the computation in a 2D slice, which is reasonably fast because the number of regions is not large if the input dimension is not high, as suggested by Theorem 9. Additionally, note that the check for a given unit is embarrassingly parallel, meaning the computation can be accelerated. To demonstrate that the computation can be carried out in a reasonable time, we also analyse the algorithm's space-time complexity.

**Space-time complexity of the algorithm**

To start, we estimate complexities for some number of activation regions $R$. Firstly, consider the space complexity. Since we store all activation regions, the space requirement grows as $R$ multiplied by an activation region size. We store a region as a constant size function computed on it and as a system of linear inequalities. The maximum number of inequalities is attained when each of $N$ neurons adds a new system of inequalities to the region, while $K-1$ inequalities determine that one pre-activation feature attains a maximum. Therefore, the space complexity of the algorithm is $\mathcal{O}(RKN)$.

---

**Algorithm 2** Auxiliary Function That Checks if a Pre-Activation Feature Creates a New Region

---

1: **function** NEWREGIONCHECK(`unit_features`, `feature`, `region`)
2:     `objective` = zeros
3:     `inequalites` = `region.inequalities`
4:     `unit_features.weights` = `unit_features.weights` × `region.weights`
5:     `unit_features.biases` = `unit_features.weights` × `region.biases`
6:                             + `unit_features.biases`
7:     **for** `another_feature` in `unit_features` \ `feature` **do**
8:         `inequalities.append(another_feature.weights` - `feature.weights` × $x$
9:                             ≤ `feature.bias` - `another_feature.bias`)
10:    **if** LinearProgramming.Solve(`objective`, `inequalities`) **then**
11:        `next_layer_function` = `region.next_layer_function`
12:                             + [`feature.weights`, `feature.bias`]
13:        **return** Region(`inequalites`, `region.function`, `next_layer_function`)
14:    **return** None

---

Now consider the time complexity. Since we traverse the network unit by unit, and for each pre-activation feature of a unit and each available activation region, we solve a linear programming problem, the time complexity is $\mathcal{O}$ of $RKN$ times the time complexity of a linear programming method. We have used an interior point method that has a polynomial-time complexity of $\mathcal{O}(\frac{n^3}{\log n}L)$, see Anstreicher (1999), where $n$ is the dimension of the variables, which is the dimension of the network input $n_0$, and $L$ is the number of bits used to represent the method input. The input is the set of inequalities, and as we have just discussed, its size is $\mathcal{O}(KN)$. Combining everything, and using $\mathcal{O}(n^3 L)$ instead of $\mathcal{O}(\frac{n^3}{\log n}L)$ for simplicity, we get that the time complexity of the whole algorithm is $\mathcal{O}(RK^2N^2n_0^3)$.

To get complexities for the average case, assume $N \geq n_0$. Then, based on Theorem 9, $R$ grows as $\mathcal{O}((K^3N)^{n_0})$. Therefore, the space complexity is $\mathcal{O}(KN(K^3N)^{n_0})$ and the time complexity is $\mathcal{O}(K^2N^2n_0^3(K^3N)^{n_0})$. Both space and time complexities grow exponentially with the input dimension but polynomially with the number of neurons and a maxout unit's rank.

## J   Parameter initialization

### J.1   He initialization

We briefly recall the parameter initialization procedure for ReLU networks which is commonly referred to as "He initialization" (He et al., 2015). This follows the motivation of the work by Xavier and co-authors (Glorot and Bengio, 2010). To train deep networks, one would like to avoid vanishing or exploding gradients. The approach formulates a sufficient condition for the norms of the activations across layers to not blow up or vanish. For ReLU networks this leads to sampling the weights from a distribution with standard deviation $\sqrt{2/n_l}$.

### J.2   He-like initialization for maxout (Maxout-He)

We follow the derivation from Glorot and Bengio (2010) and He et al. (2015) but for the case of maxout units. We note that a He-like initialization for maxouts was considered by Sun et al. (2018) but only for $K = 2$. We focus on the forward pass and consider fully-connected layers. The idea is to investigate the variance of the responses in each layer. We use the following notations. For a given layer $l$ with $d$ units and $n_l$ inputs, a (pre-activation) response is $\mathbf{y}_l = W_l\mathbf{x}_l + \mathbf{b}_l$, where $\mathbf{x}_l \in \mathbb{R}^{n_l}$ is an input vector to the layer, $W_l \in \mathbb{R}^{d \times n_l}$ is a matrix, $\mathbf{b}_l \in \mathbb{R}^d$ is a vector of biases. We have $\mathbf{x}_l = \phi(\mathbf{y}_{l-1})$, where $\phi$ is the activation function.

We assume the elements in $W_l$ are independent and identically distributed (iid). We assume that the elements in $\mathbf{x}_l$ are also iid. We assume that $\mathbf{x}_l$ and $W_l$ are independent of each other. Denote $y_l$, $w_l$, and $x_l$ the random variables of each element in $\mathbf{y}_l$, $W_l$, and $\mathbf{x}_l$ respectively. In the following we assume that biases are zero. Then we have:

$$\text{Var}[y_l] = n_l\text{Var}[w_l \cdot x_l].$$

If we assume further that $w_l$ has zero mean, then the variance of the product of independent variables gives us:

$$\text{Var}[y_l] = n_l \text{Var}[w_l] \mathbb{E}[x_l^2]. \tag{13}$$

We need to estimate $\mathbb{E}[x_l^2]$. For ReLU, $\mathbb{E}[x_l^2] = \frac{1}{2}\text{Var}[y_{l-1}]$. For maxout we get a different result. Let $K$ be the rank of a maxout unit. Then $x_l = \phi(y_{l-1}) = \max_{k \in [K]}\{y_{l-1,k}\}$. The $y_{l-1,1}, \ldots, y_{l-1,K}$ are independent and have the same distribution. We denote $f(t)$ and $F(t)$ the pdf and cdf of this distribution. The cdf for $x_l = \max_{k \in [K]}\{y_{l-1,k}\}$ is, dropping the subscript $l - 1$ of $y_{l-1,k}$ for simplicity of notation,

$$\Pr\left(\max_{k \in [K]}\{y_k\} < t\right) = \Pr(y_1, \ldots, y_K < t) = \prod_{k=1}^{K} \Pr(y_k < t) = (F(t))^K.$$

In turn, the expectation of the square is

$$\mathbb{E}\left[\max_{k \in [K]}\{y_k\}^2\right] = \int_{\mathbb{R}} t^2 \frac{d}{dt}\left[(F(t))^K\right] dt = K \int_{\mathbb{R}} t^2 (F(t))^{K-1} f(t) dt.$$

Now we can apply this formula to discuss the cases of a uniform distribution on an interval and a normal distribution. If we assume that $w_{l-1}$ has a symmetric distribution around zero, then $y_{l-1}$ has zero mean and has a symmetric distribution around zero.

**Uniform Distribution** Assuming $y_{l-1}$ has a uniform distribution on the interval $[-a, a]$, we get $\text{Var}[y_{l-1}] = a^2/3$, and

$$K = 2 : \mathbb{E}[x_l^2] = \frac{a^2}{3} = \text{Var}[y_{l-1}],$$

$$K = 3 : \mathbb{E}[x_l^2] = \frac{2a^2}{5} = \frac{6}{5}\text{Var}[y_{l-1}],$$

$$K = 4 : \mathbb{E}[x_l^2] = \frac{7a^2}{15} = \frac{7}{5}\text{Var}[y_{l-1}],$$

$$K = 5 : \mathbb{E}[x_l^2] = \frac{11a^2}{21} = \frac{11}{7}\text{Var}[y_{l-1}].$$

More generally, $\mathbb{E}[x_l^2] = 4a^2(\frac{K}{K+2} - \frac{K}{K+1} + \frac{K}{4K})$.

**Normal Distribution** Assuming $y_{l-1}$ has a normal distribution $\mathcal{N}(0, \sigma^2)$, the closed form solution is available for up to $K = 4$. We have:

$$K = 2 : \mathbb{E}[x_l^2] = \text{Var}[y_{l-1}],$$

$$K = 3 : \mathbb{E}[x_l^2] = \frac{\sqrt{3} + 2\pi}{2\pi}\text{Var}[y_{l-1}],$$

$$K = 4 : \mathbb{E}[x_l^2] = \frac{\sqrt{3} + \pi}{\pi}\text{Var}[y_{l-1}],$$

$$K = 5 : \mathbb{E}[x_l^2] \approx 1.80002\text{Var}[y_{l-1}].$$

Inserting the expressions for $\mathbb{E}[x_l^2]$ into (13),

$$\text{Var}[y_l] = n_l \text{Var}[w_l] c \text{Var}[y_{l-1}],$$

where $c$ depends on the distribution and on $K$. Putting the results together for all layers,

$$\text{Var}[y_L] = \text{Var}[y_1] \prod_{l=2}^{L} c n_l \text{Var}[w_l].$$

A sufficient condition for this product not to increase or decrease exponentially in $L$ is that, for each layer, $c n_l \text{Var}[w_l] = 1$. This is achieved by setting the standard deviation (std) of $w_l$ as $\sqrt{1/cn_l}$. For

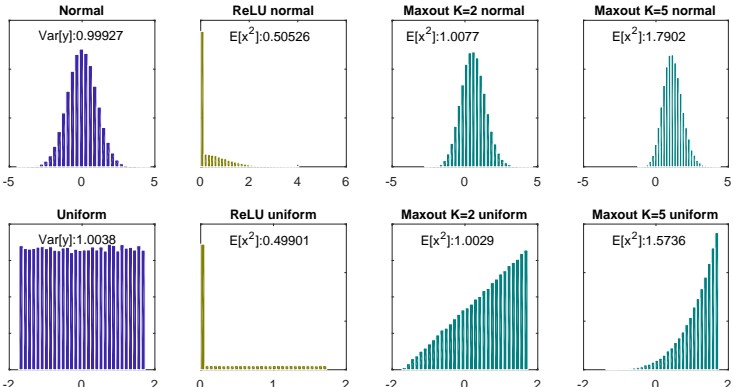

Figure 10: Shown are normal (top) and uniform (bottom) input distributions, as well as the corresponding response distributions for ReLU, maxout of rank $K = 2$, and maxout of rank $K = 5$. The expectation of the square response for maxouts of rank $K > 2$ depends not only on the variance but also on the particular shape of the input distribution.

$K = 2$ this is $\sqrt{1/n_l}$ for both uniform and normal distribution. For a uniform distribution, we obtain the condition $\mathrm{Var}[w_l] = \frac{1}{n_l(\frac{1}{4} - \frac{K}{(K+2)(K+1)})}$.

We notice that for ReLU, the particular shape of the distribution of the (pre-activation) response $y_{l-1}$ does not impact the expected square of the activation $x_l$. Indeed, as soon as $w_l$ is assumed to be symmetric around zero, one obtains $\mathbb{E}[x_l^2] = \frac{1}{2}\mathrm{Var}[y_{l-1}]$. In contrast, for maxout units of rank $K > 2$, the particular shape of the distribution of $y_{l-1}$ does affect the value of $\mathbb{E}[x_l^2]$. This is why we obtain different conditions on the standard deviation of the weight distributions depending on the assumed response distribution. The situation is illustrated in Figure 10. Among the possible distributions that one might assume for $y_{l-1}$, a normal distribution appears most natural. Therefore, we take the standard deviations obtained under this assumption as the ones defining the maxout-He initialization procedure. The values of the std of $w_l$ for $K$ up to 5 for normal distributions are shown in Table 1.

### J.3 Sphere initialization

If we initialize the pre-activation features of a maxout unit independently, then we expect the number of regions of the unit will be significantly smaller than $K$, as discussed in Appendix C. In view of Proposition 20, the number of regions of a maxout unit with weights $w_1, \ldots, w_K \in \mathbb{R}^n$ and biases $b_1, \ldots, b_K \in \mathbb{R}$ is equal to the number of upper vertices of the polytope $\mathrm{conv}\{(w_r, b_r) : r \in [K]\}$. Hence one way to have each rank-$K$ maxout unit have $K$ linear regions over its input at initialization is to initialize the pre-activation feature parameters as points in the upper half-sphere $\{(w, b) \in \mathbb{R}^{n+1} : \|(w, b)\| = 1, b > 0\}$. This can be done as follows. For each pre-activation feature $i = 1, \ldots, K$:

1. Sample $(w_i, b_i)$ from a Gaussian on $\mathbb{R}^{n+1}$.

2. Normalize $(w_i, b_i)/\|(w_i, b_i)\|$.

3. Replace $b_i$ with $|b_i|$.

If desired, subtract a constant $c$ from each of the biases $b_1, \ldots, b_K$. For instance one may choose $c$ so that the mean output of the maxout unit is approximately 0 for inputs from a Gaussian distribution. We have used $c = 1/\sqrt{Kn_l}$ in our implementation, and Gaussian had zero mean and unit covariance.

### J.4 Many regions initialization

We can initialize the parameters of a maxout layer so that the layer has the largest possible number of linear regions over its input space. A description of parameter choices maximizing the number of regions for a layer of maxout units has been given by Montúfar et al. (2021, Proposition 3.4).

The number of regions of a layer of maxout units corresponds to the number of upper vertices of a Minkowski sum of polytopes. A construction maximizing the number of vertices of Minkowski sums was presented earlier by Weibel (2012). The procedure is as follows. Let the layer have input dimension $n$. For each unit $j = 1, \ldots, m$:

1. Sample a vector $v_j \in \mathbb{R}^n$ from a distribution which has a density.
2. For each pre-activation feature $i = 1, \ldots, K$ set the weights and bias as $w_{j,i} = v_j \cos(\pi i / K)$ and $b_{j,i} = \sin(\pi i / K)$.

This construction ensures that each unit has $K$ linear regions separated by $K - 1$ parallel hyperplanes, and the hyperplanes of different units are in general position. Then the number of regions of the layer is the one indicated in the first item of Theorem 21.

If desired, one can add some noise to each of the above parameters (e.g. standard normal times a small constant) in order to have a parameter distribution which has a density. If desired, one can also normalize the initialization by subtracting an appropriate constant (e.g. to achieve a zero mean activation) and dividing by an appropriate standard deviation (e.g. to achieve that the activations have a unit mean norm). We were sampling $v_j$ from a Gaussian distribution with mean zero and std chosen according to maxout-He.

### J.5 Steinwart-like initialization for maxout

Steinwart (2019) investigated initialization in ReLU networks. He suggested that having the nonlinear locus of different units evenly spaced over the input space at initialization could lead to faster convergence of training, which he also supported with experiments on the datasets from the UCI repository. We can formulate a version of this general idea for the case of maxout networks as follows.

1. Assume we have some generic initialization procedure for individual units, which gives us weights $w_1, \ldots, w_K \in \mathbb{R}^n$ and biases $b_1, \ldots, b_K \in \mathbb{R}$. The initialization procedure could be for instance "Sphere". Upon initialization, our unit is computing a function $x \mapsto \max\{\langle w_1, x \rangle + b_1, \ldots, \langle w_K, x \rangle + b_K\}$ with non-linear locus that we denote $L$.
2. For each unit, sample a vector $c$ uniformly from the cube $[-1, 1]^n$. Alternatively, sample $c$ as a random convex combination of a random subset of the training data, so that $c = \sum_{i=1}^{m} p_i x_i$, where $(p_1, \ldots, p_m)$ is a random probability vector and $x_1, \ldots, x_m$ are $m$ randomly selected training input examples.
3. Now set the weights as $w_1, \ldots, w_K$ and the biases as $b_1 + \langle w_1, c \rangle, \ldots, b_K + \langle w_K, c \rangle$. Now our unit is computing a function $x \mapsto \max\{\langle w_k, x \rangle + b_k + \langle w_k, c \rangle\} = \max\{\langle w_k, x + c \rangle + b_k\}$. Hence the nonlinear becomes $L - c$.

## K  Experiments

In this section, we provide details on the implementation and additional experimental results. All the experiments were implemented in Python using PyTorch (Paszke et al., 2019), numpy (Harris et al., 2020), scipy (Jones et al., 2001) and mpi4py (Dalcin et al., 2011), with plots created using matplotlib (Hunter, 2007). In the experiments concerning the network training, we used the MNIST dataset (LeCun et al., 2010). PyTorch, numpy, scipy and mpi4py are made available under the BSD license, matplotlib under the PSF license and MNIST dataset under the Creative Commons Attribution-Share Alike 3.0 license. We conducted all experiments on a CPU cluster that uses Intel Xeon IceLake-SP processors (Platinum 8360Y) with 72 cores per node and 256 GB RAM. The most extensive experiments were usually running for 2-3 days on 32 nodes. The computer implementation of the key functions is available on GitHub at `https://github.com/hanna-tseran/maxout_complexity`.

For the MNIST experiments we use the Adam optimizer with mini-batches of size $128$ with learning rate $0.001$ and the standard Adam hyperparameters from PyTorch (betas are $0.9$ and $0.999$). Counting at initialization was performed in the window $[-50, 50]^2$, in the training experiments in the window $[-400, 400]^2$ defined on the slice, and images of the regions and the decision boundary were obtained in the window $[-50, 50]^2$ also defined on the slice. All results are averaged over 30 instances where applicable. The network architectures are specified in the individual experiments. The parameter initialization procedures are implemented following the descriptions in Appendix J. For the



Figure 11: A few functions represented by a maxout network for different parameter values in a 2D slice of parameter space. For each function we plot regions of the input space with different gradient values using different colors.

experiments counting the number of activation regions and pieces in the decision boundary we use home made implementations of the algorithms described in Appendix I. Further below we present the details and additional results of the individual experiments.

**Details on Figure 1** We consider a network with 2 input units, three layers of rank-3 maxout units of width 3, and a single linear output unit. We fix three parameter vectors $\theta_0, \theta_1, \theta_2$ drawn from a normal distribution over the parameter space and define a grid of parameter values $\theta(\xi_1, \xi_2) = \theta_0 + \xi_1 \theta_1 + \xi_2 \theta_2$ with $(\xi_1, \xi_2)$ taking 102400 uniformly spaced values in $[-1, 1]^2$. For each of these parameter values, we estimate the number of linear regions that the represented function has over the square $[-1, 1]^2$ in the input space. To this end, we evaluate the gradient of the function over 102400 uniformly spaced input points and take the number of distinct values an estimate for the number of linear regions. Then we plot the estimated number of linear regions as a function of $(\xi_1, \xi_2)$. A subset of 25 out of the evaluated functions is shown in Figure 11.

**Comparison to the upper bound** Figures 12 and 13 complement Figure 2. Figure 12 compares the number of activation regions and linear pieces in the decision boundary to the formulas both with and without the constants, while Figure 13 demonstrates the results for different values of $K$.

**Effects of the depth and the number of units on the number of linear regions** Results adding more information to Figure 3 are in Figure 14. It shows that ReLU networks and maxout networks with $K = 2$ have a similar number of activation regions that does not depend on the network depth but rather on the total number of units. This figure also shows that maxout networks with ranks $K > 2$ tend to have fewer regions as the depth increases, but the number of units remains constant and that the difference in the number of regions becomes more apparent for larger ranks.

**Effects of different initializations on training** Figure 16 is a more detailed version of Figure 6. It shows how convergence speed changes for different network depths and different maxout ranks given different initializations. The improvement from maxout-He, sphere, and many regions initializations compared to ReLU-He initialization becomes more noticeable with larger network depth and larger maxout rank. We have also checked how the Steinwart initialization affects the convergence speed, but found no significant difference in this particular experiment. We used the approach where $c$ is taken as a convex combination of all training data points (weights $p$ uniformly at random from the probability simplex). The results are shown in Figure 15.

**Effects of different initializations on the number of activation regions and pieces in the decision boundary during training** Figure 18 adds more information to Figure 5 and demonstrates how the number of activation regions and linear pieces in the decision boundary changes for different initializations during training on the MNIST dataset. We observe that the number of activation regions and pieces of the decision boundary increase for all tested initialization procedures as training progresses. Nonetheless, the number remains much lower than the theoretical maximum. Figure 17 illustrates how linear regions and the decision boundary evolve during training.

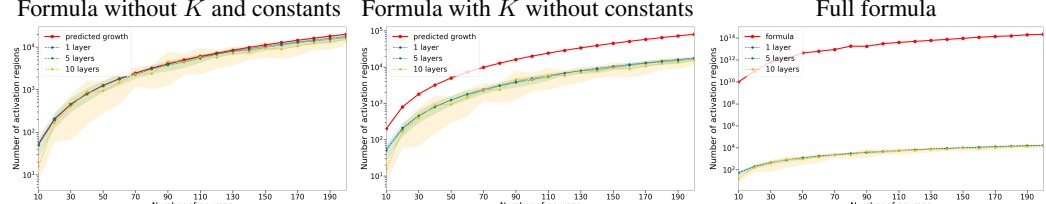

(a) Number of activation regions for a network with ReLU-He normal initialization.

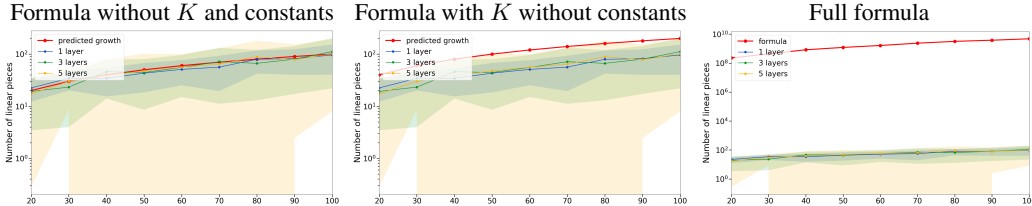

(b) Number of linear pieces in the decision boundary for a network with maxout-He normal initialization.

Figure 12: Comparison to the formulas with and without the constants for the number of activation regions and linear pieces in the decision boundary from Theorem 9 and Theorem 11 respectively. Networks had 100 units and maxout rank $K = 2$. The settings are similar to those in Figure 2.

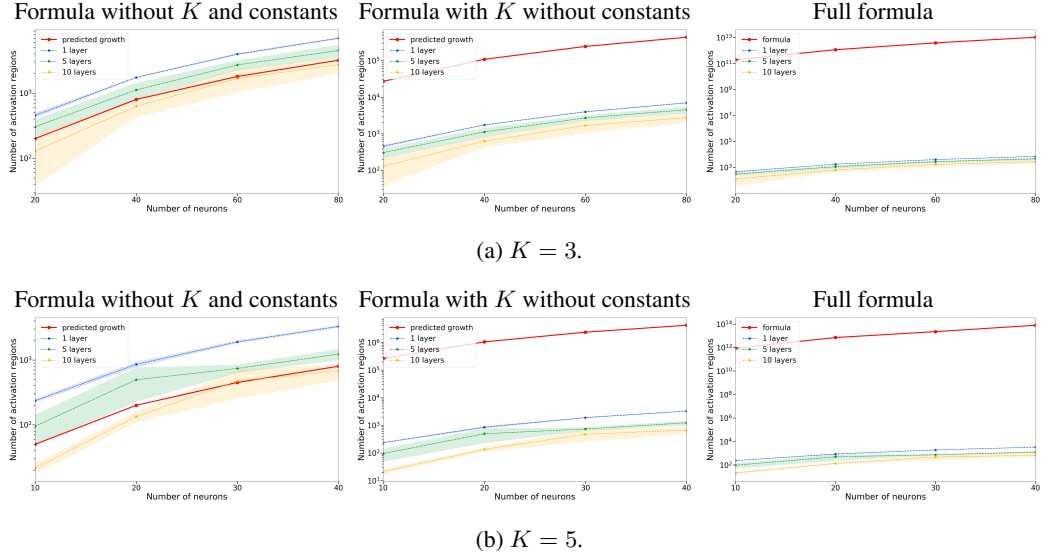

Figure 13: Comparison to the formula from Theorem 9 for maxout ranks $K = 3$ and $K = 5$. The networks were initialized with maxout-He normal initialization. We observe the increase in the number of activation regions as the maxout rank increases, and for networks with higher maxout rank deeper networks tend to have less regions than less deep networks with the same rank.

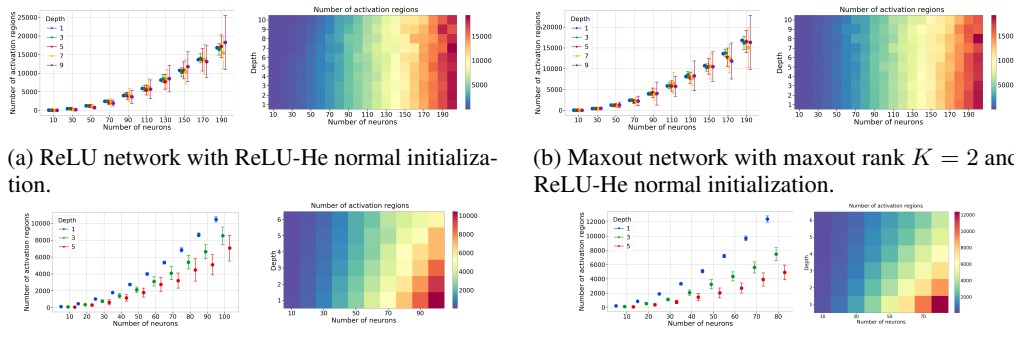

(a) ReLU network with ReLU-He normal initialization.

(b) Maxout network with maxout rank $K = 2$ and ReLU-He normal initialization.

(c) Maxout network with $K = 3$. Maxout-He normal initialization.

(d) Maxout network with $K = 5$. Maxout-He normal initialization.

Figure 14: Difference between the effects of depth and number of neurons on the number of activation regions. These plots are additional to Figure 3 and have similar settings. ReLU and maxout networks with $K = 2$ have a similar number of linear regions. For maxout rank $K > 2$ deeper networks tend to have less regions than less deep networks with the same rank. For $K = 3$ the gaps between different depths are smaller than for $K = 5$.

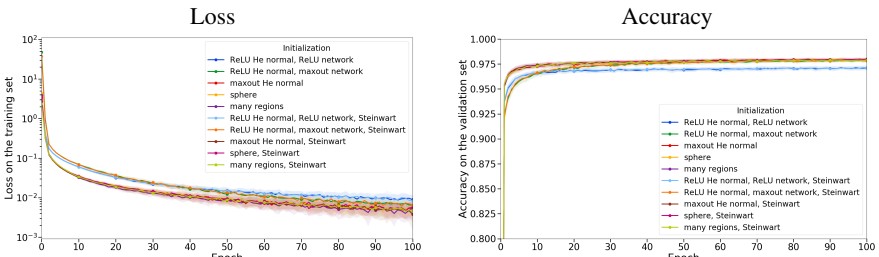

Figure 15: Effect of the Steinwart initialization approach on the convergence speed during training on the MNIST dataset for a network with 200 units and 5 layers. Maxout rank was $K = 5$. In this experiment, for various initialization procedures, the addition or omission of a random shift of the non-linear regions of different units led to similar training curves.

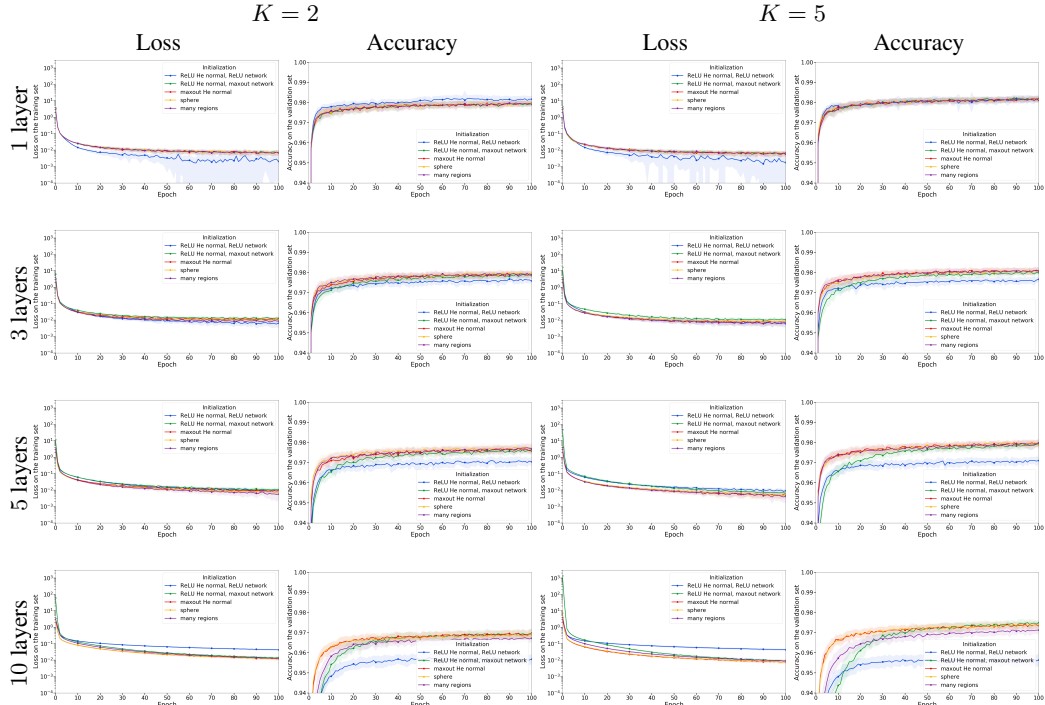

Figure 16: Effect of the initialization on the convergence speed during training on the MNIST dataset of networks with 200 units depending on the network depth and the maxout rank. Maxout-He, sphere, and many regions initializations behave similarly, and the improvement in the convergence speed becomes more noticeable for larger network depth and maxout rank.

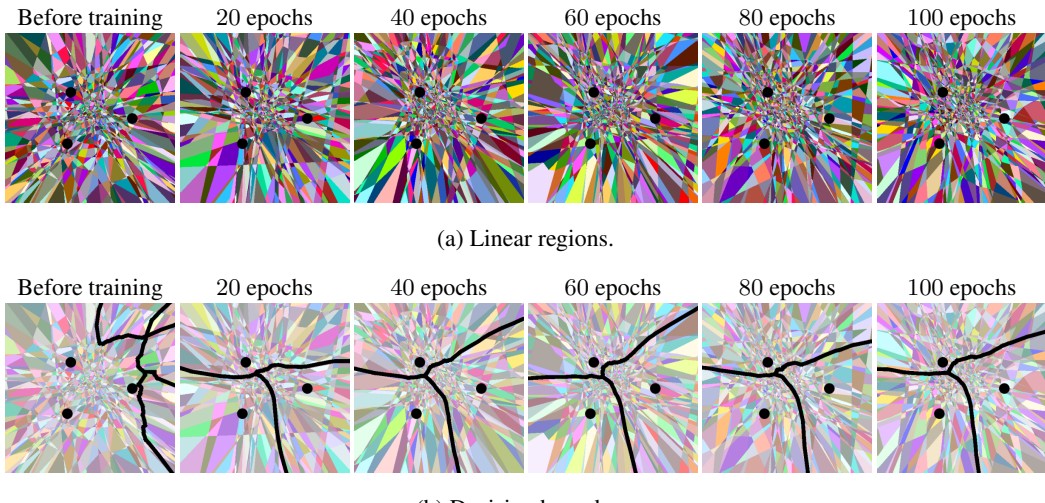

(b) Decision boundary.

Figure 17: Evolution of the linear regions and the decision boundary during training on MNIST in a 2D slice determined by three random input points from the dataset. The network had 3 layers, a total of 100 maxout units of rank $K = 2$, and was initialized with the maxout-He initialization.

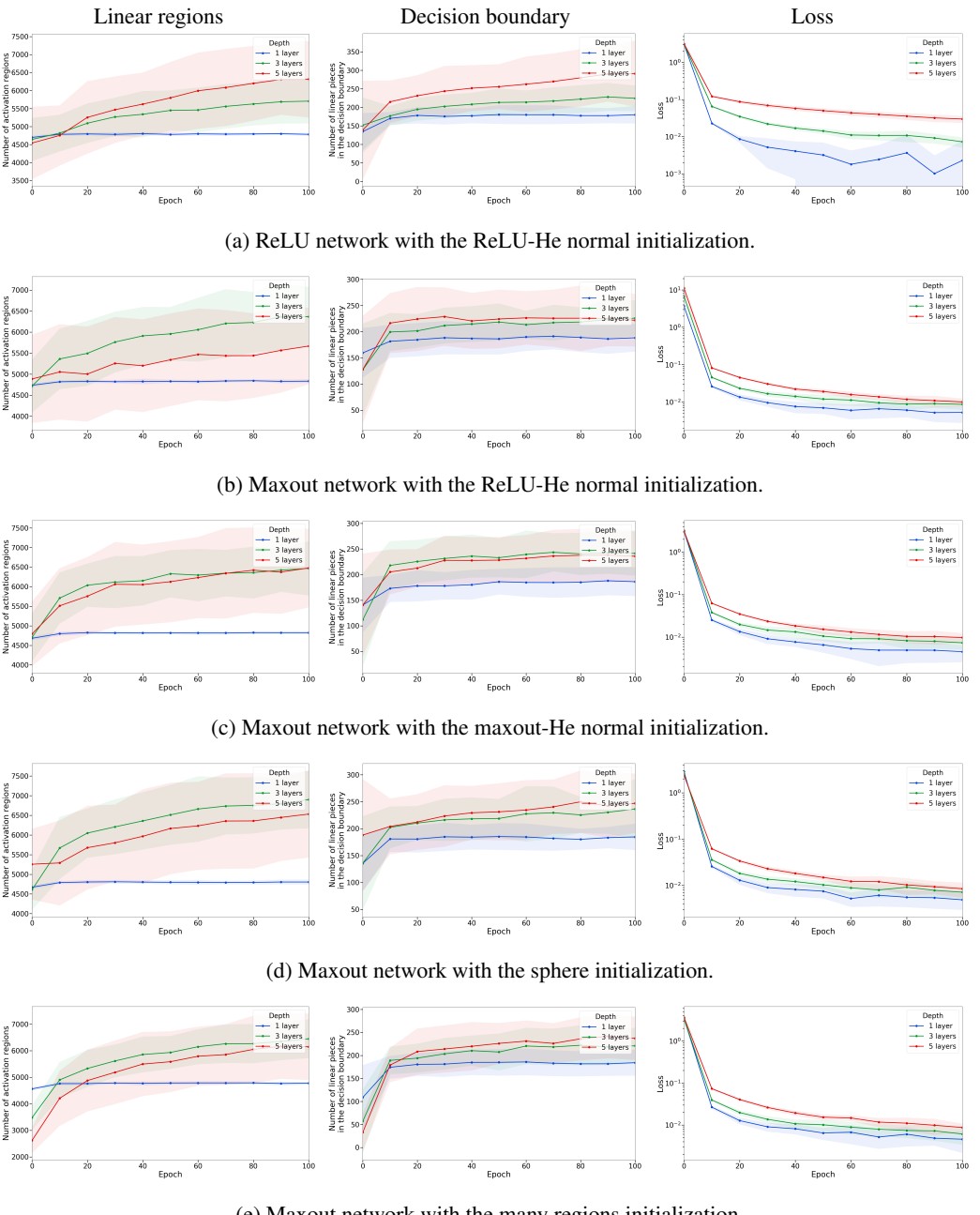

(a) ReLU network with the ReLU-He normal initialization.

(b) Maxout network with the ReLU-He normal initialization.

(c) Maxout network with the maxout-He normal initialization.

(d) Maxout network with the sphere initialization.

(e) Maxout network with the many regions initialization.

Figure 18: Change in the number of linear regions and the decision boundary pieces during 100 training epochs given different initializations. Networks had 100 neurons and for maxout networks $K = 2$. Both the number of linear regions and linear pieces of the decision boundary increases during training for all initializations but remain much smaller than the theoretical maximum. The settings were the same as in Figure 5.