# OpenReview forum: "On the Expected Complexity of Maxout Networks"
_NeurIPS.cc/2021/Conference — NeurIPS 2021 Poster_

### Official Review · Reviewer_9cCV · 2021-07-12

**Rating:** 6
**Confidence:** 3

**Summary:**

This paper studies the complexity of deep maxout networks from the point of view of (expected or maximal) number of linear regions. Theoretical lower and upper bound are derived, and the paper concludes with experiments to assess the quality of the theoretical bounds in trained networks.

**Main Review:**

The theoretical bounds are derived thoroughly. The paper puts particular emphasis on expected behavior, which should have more relevance in practice. Bounds for the expected complexity of ReLU networks were already found before, so the main novelty of this paper lies in the consideration of maxout networks instead of ReLU networks. This appears to me as a modest contribution. Perhaps the authors should consider to add a more explicit comparison of the bounds for ReLU and maxout networks and of the qualitative difference between ReLU and maxout cases, which could help appreciate the challenges and novelties of the maxout case.

The experiments are certainly useful but they do not demonstrate clear applications of this work.

Note: rating changed from 5 to 6 following discussion.

**Time Spent Reviewing:**

2

---

> ### Author Response · Authors · 2021-08-10
> **Response to Reviewer 9cCV**
>
> Thank you for the valuable feedback. Please see below for our response to your comments.
>
> __Q1: Bounds for the expected complexity of ReLU networks were already found before, so the main novelty of this paper lies in the consideration of maxout networks instead of ReLU networks. This appears to me as a modest contribution.__
>
> Kindly note that Theorem 7 and Theorem 8 are addressing a new and different type of questions compared with previous works, and the proofs also use a very different approach (including Minkowski sums of polytopes and tropical geometry). These results point at important differences between ReLU and maxout networks. For instance, maxout networks have multiple numbers of linear regions attained with positive probability (which is not true for shallow ReLU networks), and have a non-trivial lower bound on the number of linear regions (which is not true for deep ReLU networks).
>
> Note also that Theorem 11, Theorem 12, and Corollary 13 are new results not only for maxout networks but also for ReLU networks (although the proofs build on methods derived from the works of Hanin and Rolnick).
>
> Note that we also presented new parameter initialization procedures for maxout networks based on a specific analysis, and we also provided an implementation for computing the linear regions in maxout networks.
>
> The reviewer's comment probably refers to Theorem 9 and Theorem 10. We point out that although the statements are similar in spirit to results that have been previously obtained for ReLU networks, generalizing the previous approaches to make them viable in this setting requires significant efforts. Not only have maxout units multiple arguments compared with the single argument in ReLUs. There are clear quantitative differences including more complex combinatorics, but also qualitative differences which make the results not necessarily expected. In particular, for ReLU networks the existence of closed paths can drastically reduce the number of regions, but for maxout networks paths are always open. Recent works seeking to upper bound the number of linear regions of ReLU networks exploit dimension arguments based on the existence of closed paths (e.g., Serra et al; Hinz and Van de Geer; Xie et al; Hinz), which are not valid for maxout networks.
>
> We hope this clarifies that our contributions include new types of results and also non-trivial versions of interesting recent results for a different type of networks.
>
> __Q2: Perhaps the authors should consider to add a more explicit comparison of the bounds for ReLU and maxout networks and of the qualitative difference between ReLU and maxout cases, which could help appreciate the challenges and novelties of the maxout case.__
>
> Thank you for the suggestion.
>
> Some of the key challenges are described in line 58 ff, line 136, line 162 ff. ReLU layers are described by relatively well understood hyperplane arrangements, whereas maxout layers are described by so-called tropical hyperplane arrangements, which are not so well understood. Note that even the maximum number of regions of a shallow maxout network was not known until very recently (in spite of numerous works on related problems in geometric combinatorics). The asymptotic expansions in lines 148, 154, 156 give intuitive numerical values to compare against.
>
> We compare the upper bound on the expected number of activation regions to a similar bound for ReLU in lines 215 - 216. The comparison of the volumes would be similar, but we will add details in the revision following your suggestion.
>
> As we mentioned in line 215 the upper bounds for maxout networks in the special case of K=2 and ReLU have a slight difference in constants. On the other hand, the constant $C_{grad}$ can be different in the two architectures, as is evidenced in our analysis of parameter initialization in Appendix J and Appendix F. We will add more details on this in the revision, as pointed out in the answer to Q2 of Reviewer dn4N.
>
> Our results for the decision boundaries are new. We obtain a similar result for ReLU networks as mentioned in line 254 and Appendix H. The bounds for the case K=2 and for ReLUs are similar in this case.
>
> Following your suggestions, we will add a table comparing the bounds on the maximum and expected numbers of linear regions and other results for maxout networks of different ranks and ReLU networks.
>
> __Q3: The experiments are certainly useful but they do not demonstrate clear applications of this work.__
>
> The main focus of this work is theoretical. The main goal of the experiments was to illustrate the theoretical results numerically and evaluate their tightness in a practical setting.
>
> We point at two straightforward applications of our work:
> 1) the initialization procedures proposed in Section 6 and Appendix J.
> In the experiments we explored the consequences of these parameter initialization procedures on MNIST and obtained promising results. As we point out in line 306, exploring these procedures in more detail, particularly on other architectures (e.g. CNNs) and data sets (e.g. CIFAR) are interesting extensions. We are currently implementing these.
> 2) The lower bound on the distance to the decision boundary that can be useful for studying adversarial robustness.
>
> We believe that several other interesting applications can be derived based on our results. In particular, the presented results can facilitate studies of the speed of convergence, robustness, and curvature of the solutions after training, which have been connected to the distribution of linear regions at initialization for instance by Croce et al., 2019, Steinwart (2019), Williams et al. (2019), Jin and Montufar (2020).

---

> > ### Comment · Reviewer_9cCV · 2021-08-12
> > **Response to response**
> >
> > Thank you, that was helpful. I think now I appreciate more the differences between the two cases and the challenges of maxout networks.
> >
> > > Following your suggestions, we will add a table comparing the bounds on the maximum and expected numbers of linear regions and other results for maxout networks of different ranks and ReLU networks.
> >
> > That would be very helpful.

---

### Official Review · Reviewer_dn4N · 2021-07-16

**Rating:** 7
**Confidence:** 5

**Summary:**

This articles provides bounds for both average case and worst case complexity (in terms of number of linear regions and related measures) for max-out networks. I found the article to have a few results that were interesting (to do with the difference between ReLU and max-out), a few results that were correct but rather incremental (average case analysis of number of regions for max-out networks), and one result whose proof I cannot understand (though it may well be true).

After rebuttal: I am satisfied with the replies of thr authors and am raising my score by 1.

**Limitations And Societal Impact:**

Yes

**Main Review:**

Strengths:

1. I found this article to be clearly written.

2. I believe most of the results in this article are correct and correctly proved (see point 2 in weaknesses below for a concern).

3. I thought the authors did a good job of comparing and contrasting their results on average case complexity and the results on worst case complexity.

4. It was interesting to see the difference between ReLU and max-out complexity, even for one layer networks.


Weaknesses:

1. The proofs in this article for results on average number/volume of activation patterns and linear regions follow very closely prior work of Hanin and Rolnick, and I view them as incremental.

2. I am not sure whether the constant C_bias and C_grad in the average case results are actually bounded as in Appendix F. The issue is that I am not certain if the analysis from the prior work Hanin and Nica extends directly to max-out units. If it does, I think the authors should consider explaining how. Otherwise, this strikes me as a major weakness since I view average case results as primarily interesting when averaging with respect to the initialization schemes really used in practice.


Minor Remarks/Comments:

1. In Definition 1: why the word continuous?
2. In Theorem 7: The notation S \subseteq \binom{[n_1]}{j} should probably be defined?
3. In Theorem 9: “for for” in condition 3

**Time Spent Reviewing:**

3

---

> ### Author Response · Authors · 2021-08-10
> **Response to Reviewer dn4N**
>
> Thank you for the valuable feedback. Please see below for our response to your comments.
>
> __Q1: The proofs in this article for results on average number/volume of activation patterns and linear regions follow very closely prior work of Hanin and Rolnick, and I view them as incremental.__
>
> Kindly note that Theorem 7 and Theorem 8 are addressing a new and different type of questions compared with previous works, and the proofs also use a very different approach (including Minkowski sums of polytopes and tropical geometry). These results point at important differences between ReLU and maxout networks. For instance, maxout networks have multiple numbers of linear regions attained with positive probability (which is not true for shallow ReLU networks), and have a non-trivial lower bound on the number of linear regions (which is not true for deep ReLU networks).
>
> Note also that Theorem 11, Theorem 12, and Corollary 13 are new results not only for maxout networks but also for ReLU networks (although the proofs build on methods derived from the works of Hanin and Rolnick).
>
> Note that we also presented new parameter initialization procedures for maxout networks based on a specific analysis, and we also provided an implementation for computing the linear regions in maxout networks.
>
> The reviewer's comment probably refers to Theorem 9 and Theorem 10. We point out that although the statements are similar in spirit to results that have been previously obtained for ReLU networks, generalizing the previous approaches to make them viable in this setting requires significant efforts. Not only have maxout units multiple arguments compared with the single argument in ReLUs. There are clear quantitative differences including more complex combinatorics, but also qualitative differences which make the results not necessarily expected. In particular, for ReLU networks the existence of closed paths can drastically reduce the number of regions, but for maxout networks paths are always open. Recent works seeking to upper bound the number of linear regions of ReLU networks exploit dimension arguments based on the existence of closed paths (e.g., Serra et al; Hinz and Van de Geer; Xie et al; Hinz), which are not valid for maxout networks.
>
> We hope this clarifies that our contributions include new types of results and also non-trivial versions of interesting recent results for a different type of networks.
>
> __Q2: I am not sure whether the constant__ $C_{\text{bias}}$ __and__ $C_{\text{grad}}$ __in the average case results are actually bounded as in Appendix F. The issue is that I am not certain if the analysis from the prior work Hanin and Nica extends directly to max-out units. If it does, I think the authors should consider explaining how. Otherwise, this strikes me as a major weakness since I view average case results as primarily interesting when averaging with respect to the initialization schemes really used in practice.__
>
> This is a good point. The calculation is indeed different. We have carefully verified that a similar (and slightly stronger) statement holds. We will add full details in the revision. For now let us explain it here. First of all, the derivation for $C_{\text{bias}}$ in Appendix F is specific to maxout and does not depend on Hanin and Nica (2019). As for $C_{\text{grad}}$ the proof goes as follows.
> We assume that the weights at each layer are sampled iid from a Gaussian distribution with variance $c/{n_{l - 1}}$, where $c \in \mathbb{R}$ is some constant. Then the gradient is equal in distribution to a product of weight matrices $W^{(l)}$ and diagonal matrices $D^{(l)}$ (more precisely a contraction of 3-way tensors). Along the second dimension, the $\mathcal{D}^{(l)}$ have a single entry equal to one at the position corresponding to the pre-activation feature that attains a maximum. For any fixed input, this position is uniformly distributed because all pre-activation features receive the same input and apply affine linear functions with iid parameters. Using a result from Hanin et al. (2021), the norm of these products is equal to the product of the norms, whereby the norm of the product of each pair is a $\chi_{n_l}$ random variable. From there we can write a formula for any $t$ moment of the gradient norm. This can be upper-bounded again to get the result presented in the Appendix F. For $C_{\text{grad}}$ we may choose $c^{L/2} \prod_{l=0}^{L - 1}\left( 1 + t/n_l \right)^{1/2}
> \leq C_1 \exp ( C_2 \sum_{l=0}^{L-1} n_l^{-1/2} )$, where $C_1, C_2$ are some constants. This is a slight improvement over the bound that we had previously stated.
>
> __Q3: In Definition 1: why the word continuous?__
>
> The word ''continuous'' is redundant here and we will remove it.
>
> __Q4: In Theorem 7: The notation $S \subseteq \binom{[n_1]}{j}$ should probably be defined?__
>
> This should be $S \in \binom{[n_1]}{j}$, meaning $S$ is a subset of $[n_1]:=\{1,\ldots, n_1\}$ of cardinality $|S|=j$. We will add a definition.
>
> __Q5: In Theorem 9: ''for for'' in condition 3__
>
> Thank you for pointing out this typo. We will fix it in the revision.

---

> > ### Comment · Reviewer_dn4N · 2021-09-01
> > **Thanks for the clarification. A few concerns remain.**
> >
> > I thank the authors for the thorough response and make a few brief points:
> >
> > - I had a careful look at the proof of Theorems 7 and 8. I agree that they use new methods (at least to me), and I think the arguments are interesting. I am willing to raise my score by 1, assuming the authors can address my concerns below for C_grad.
> >
> > - I looked at the new proofs of Theorems 11 and 12 as well as Corollary 13. I still feel they are very close to the work of Hanin and Rolnick. However, I do agree the combinatorics is more tricky for max-out.
> >
> > - Thank you for the explanation re: C_grad. I still am not certain that I understand what you wrote. I have three questions.
> >
> > 1.  It seems to me that the position of the $1$ in the diagonal matrix $D^{(\ell)}$ is not independent of the weights matrices $W^{(\ell)}$. If I understand correctly, you are trying to figure out, among a collection $\{ W_1 x+b_1,…,W_kx + b_k \}$, which one has the largest magnitude. That’s basically going to be the position of the $1$. But it seems that the event that the $i$-th index has the largest magnitude is not independent of $W_i, b_i$. The product of chi-squared random variables result you mention probably requires this! Or am I missing something?
> >
> > 2. Less crucially, I think there are some typos in what you wrote about the moments of the chi-squared random variables. I believe the t-th moment of $n^{-1}\cdot \chi_n^2$ Should be like $(1+2/n)…(1+(2(t-1)/n)$ and the upper bound should have $\sum_{\ell=1}^L n_\ell^{-1}$ in the exponent.
> >
> > 3. Finally, how are you choosing the constant $c$? Are you setting it so that the average squared gradient of a neuron equals $1$? What is used in practice?

---

> > > ### Author Response · Authors · 2021-09-08
> > > **Response to Additional Questions to Reviewer dn4N**
> > >
> > > __Q1: It seems to me that the position of the $1$ in the diagonal matrix $D^{(l)}$ is not independent of the weights matrices $W^{(l)}$. If I understand correctly, you are trying to figure out, among a collection $W_1 x + b_1, \dots, W_k x + b_k$, which one has the largest magnitude. That’s basically going to be the position of the $1$. But it seems that the event that the $i$-th index has the largest magnitude is not independent of $W_i, b_i$. The product of chi-squared random variables result you mention probably requires this! Or am I missing something?__
> > >
> > > You are correct, the diagonal matrix is not independent of the weights. The construction of an upper bound for maxout networks leads to more complicated distributions compared with ReLU networks, as you have noticed.
> > >
> > > We seek to upper bound the moments of the gradient norm. To this end we consider the norm of the Jacobian $||J_x|| = \sup_{||u||=1}||J_x u||$. The Jacobian $J_x$ (of the output at a the $L$th layer with respect to the input to the network) can be written as a product $\prod_{l=1}^L \overline{W}^{(l)}$, where the factor matrices $\overline{W}^{(l)}$ depend on the activation region of the input $x$. Concretely, if $W^{(l)}_{i,k}$ denotes the weight vector of the $k$th pre-activation feature of the $i$th unit in the $l$th layer, then the matrix $\overline{W}^{(l)}$ has rows $\overline{W}^{(l)}_i = W^{(l)}$, where $k_i =\operatorname{argmax}_k W^{(l)} x^{(l-1)}$ (we are omitting the biases for simplicity).  (Note that the first $W^{(l)}$ should have the subscript $i ,k_i$ and $W^{(l)}$ inside argmax should have the subscript $i, k$ that we could not add because of the problem with built-in latex.)
> > >
> > > Now for a given vector $u$ of norm one, we can write $|| J_x u ||$ as $||W^{(L)} u^{(L-1)}|| || \overline{W}^{(L-1)} u^{(L-2)}|| \cdots || \overline{W}^{(2)} u^{(1)}|| || \overline{W}^{(1)} u^{(0)}||$ with $||u^{(l)}|| = 1$.
> > > Here the matrices $\overline{W}^{(l)}$ and the vectors $u^{(l)}$ depend on the input and the preceding matrices.
> > > We can consider an upper bound on the factors which gets rid of the dependencies.
> > > Namely $|| \overline{W}^{(l)} u^{(l-1)}||^2$ is upper-bounded by $\sum_{i=1}^{n_l} \max_{k \in [K]} ( ||W_{i,k}^{(l)}||^2 )$.
> > > The random variables $\sum_{i=1}^{n_l} \max_{k \in [K]} ( ||W_{i,k}^{(l)}||^2 )$, $l=1,\ldots, L$ are independent, because they are functions of distinct independent random variables only (the entries of the matrices $W^{(l)}$).
> > >
> > > Assume that the entries of each matrix $W^{(l)}$ are iid from a zero mean Gaussian with variance $\frac{c}{n_{l-1}}$. Then $C_{\text{grad}}$ can be chosen as an upper-bound on
> > > $c^{t/2} \mathbb{E}[\chi_{n_{L}}^t]  \prod_{l = 1}^{L-1} \mathbb{E}[( c\frac{1}{n_l} \sum_{i=1}^{n_l}  m^{(K)} )^{t/2} ]$, where  $m^{(K)}$ is a random variable following the distribution of the largest order statistic for a sample of size $K$ of $\chi_{n_{l-1}}^2$ variables, and $\chi_{n_{l-1}}^2$ denotes a chi-squared random variable with $n_{l-1}$ degrees of freedom. (Note that $m^{(K)}$ should have the subscript $n_{l-1}, i$ that we could not add because of the problem with built-in latex.)
> > >
> > > The only remaining question is upper-bounding the factor involving $m^{(K)}$. One way to do this is by considering the explicit expression for the moments of the largest order statistic of chi-squared variables.
> > > These moments have complicated form but are available in closed form, see, e.g. [1].
> > >
> > > Please keep in mind that this analysis pertains a discussion in the Appendix which seeks to provide additional context to our main results, but it is not the main focus of the submission.
> > > Estimating the moments of the gradient of maxout networks is a challenging topic that has been studied in much less detail than the corresponding question for ReLU networks.
> > > This question requires calculations with complicated distributions, as shown above, and is worthy of a separate investigation.
> > > Thank you very much for your careful review, which has helped us very much in making progress on this problem. We believe that this will be a valuable addition to the analysis in the paper.
> > >
> > > [1] Nadarajah, Saralees. "Explicit expressions for moments of $\chi^2$ order statistics." Bull. Inst. Math. Acad. Sin.(NS) 3 (2008): 433-444.
> > >
> > > __Q2: Less crucially, I think there are some typos in what you wrote about the moments of the chi-squared random variables. I believe the $t$-th moment of $n^{-1} \cdot \chi^2_n$ Should be like $(1 + 2/n), \dots, (1 + 2(t-1)/n)$ and the upper bound should have__ $\sum_{l=1}^{L} n_l^{-1}$  __in the exponent.__
> > >
> > > You are correct. The formula we provided previously does not have a typo but looks different from the formula you stated, because we used the chi distribution (instead of chi-squared) and we formulated an upper bound based on the representation with the Gamma ratio $2^{t/2} \frac{\Gamma((n + t) / 2)}{\Gamma(n / 2)}$.
> > >
> > > __Q3: Finally, how are you choosing the constant $c$? Are you setting it so that the average squared gradient of a neuron equals $1$? What is used in practice?__
> > >
> > > The constant $c$ is the constant that we use in the variance $c / n_{l-1}$ of the Gaussian distribution from which the weights are sampled.
> > > In contrast to ReLU networks, so far there are no works deriving an initialization for maxout networks which would ensure that the average squared gradient of a neuron equals $1$. However, we have made a step towards addressing this problem with the maxout-He initialization that we proposed in Appendix J.
> > > This initialization sets the constant $c$ depending on the rank of the maxout units to ensure that the squared gradient equals $1$. The precise calculation of $c$ depends on the assumed distribution of the pre-activation features. We computed it explicitly for the cases where the distribution is assumed to be uniform in an interval or Gaussian.
> > > For maxout-He $c$ is set according to the last column of Table 1.
> > >
> > > In practice, people usually use the same initialization as for the ReLU networks (although, as discussed above, strictly speaking this is not well justified). The standard He initialization (for ReLU networks) satisfies the assumptions of our derivations for the upper-bound on the gradient norm. In the case of using the standard He initialization, the constant is $c = 2$.
> > > Deriving the proper initialization specific to maxout networks is more difficult than for ReLU networks.
> > > Computing for more general settings will require more research, and we believe that our results can be good starting point for such studies.

---

> > > > ### Comment · Reviewer_dn4N · 2021-09-09
> > > > **Thanks for the explanation**
> > > >
> > > > I thank the authors for their thorough explanation. My understanding of the upper bound on C_grad as described is that it will grow exponentially with depth L for an init in which the mean gradient is norm squared is set to 1. This is unfortunate but I understand the authors’ point that it is potentially complicated and not the main point of this paper to find out how to do this more precisely.
> > > >
> > > > Overall, I am satisfied with the authors’ replies and am raising my score by 1 point as a result.

---

### Official Review · Reviewer_4fmj · 2021-07-16

**Rating:** 7
**Confidence:** 3

**Summary:**

This paper presents several results on the expected number of linear regions in maxout networks under mild assumptions on the parameter space. These results address a void of understanding in comparison to ReLU networks.

**Limitations And Societal Impact:**

Limitations and social impact are clearly discussed.

**Main Review:**

For the most part, this paper can be regarded as an extension to maxout networks of the results in the couple of 2019 papers by Hanin and Rolnick on the expected number of linear regions of ReLU networks. However, it is a bit more than that because the multiple affine functions of maxout units make it more difficult to analyze their number of linear regions. Hence, the authors had to first characterize linear regions according to the number of affine functions being maximized per unit, and then show that under mild assumptions on their parameters those regions with multiple affine functions being simultaneously maximized are usually empty.

Although maxout networks are not as often used as ReLU networks, I believe that this study is both valid and needed because we never know what may happen to deep learning a few years down the road. Given that maxout is also a generalization of ReLU, some of the results discussed in the paper also apply to ReLU. However, the authors interestingly observe that Lemma 5 does not apply to ReLU because of the sizeable part of the input that is mapped to 0 by each unit.

I must also state that this is a mathematically heavy paper, and I cannot say that I have carefully evaluated all of their results yet. Hence, I appreciate having feedback of the authors regarding my questions below to help me complete my evaluation of their paper.

Questions and comments to the authors:
- In line 19, what do you mean by "generic"?
- In line 122, why does r go up to n_0?
- Do activation regions correspond to linear regions (line 130) and vice versa (line 133)? The paragraph says one thing, and the result says another.
- Figure 1 Right is not clear. Can you please describe it in more detail?
- Would it be correct to say that Lemma 6 corresponds to every possible partial activation pattern being valid? If so, what is the purpose of stating it if there is a sharper bound in Montufar et. al. (2021)?
- How can a linear region have finite volume (line 158), if in some cases it is unbounded?
- Theorem 8 seems to be more of a Lemma or even a Proposition that could also apply to ReLU given hyperplanes in general position.
- In section 5, can you provide a clear definition by what you mean with the pieces of the decision boundary?

Other stylistic comments:
- In line 159, add "about" between question and which.
- Misuse of n_0 as size of maxout input, since n_0 refers to size of input layer, starting in Theorem 7. Likewise for the use of n_1 for size of the layer output.
- First bullet point of Theorem 7 could be a corollary.
- Second bullet point of Theorem 7 could be a separate result.
- Remove repeated for in line 203.

Literature review:
- In line 33, another related work on function-preserving transformations is [1].
- In line 34, another related work on robustness based on linear regions is [2].
- In line 268, another related work on counting linear regions is [3] and you should also refer to the sampling method in Xiong et al. (2020), which is applied in [4].
- In line 304, is Xavier and He a missing reference?

[1] https://arxiv.org/abs/2001.00218

[2] https://arxiv.org/abs/1907.03207

[3] https://arxiv.org/abs/1810.03370

[4] https://arxiv.org/abs/2102.11535

**Time Spent Reviewing:**

5

---

> ### Author Response · Authors · 2021-08-10
> **Response to Reviewer 4fmj**
>
> Thank you for the valuable feedback and appreciating the challenge of working with maxout networks. Please see below for our response to your questions and comments.
>
> __Q1: In line 19, what do you mean by "generic"?__
>
> We use ''generic'' in the standard sense, to refer to a positive Lebesgue measure event. For clarity and simplicity of the presentation we will remove the word ''generic'' in line 19, as this can be subsumed in the ''expected behavior'', and add a formal definition later in the paper.
>
> __Q2: In line 122, why does $r$ go up to $n_0$?__
>
> We can also define this for higher values of $r$ up to $N(K-1)$. However, as shown in Lemma 4, for almost every choice of the parameter, $r$-partial activation regions have co-dimension $r$ (and in turn they have dimension $n_0-r$) or are empty. If $r$ is larger than $n_0$, then any $r$-partial activation region is empty for almost every choice of the parameters. Therefore, when discussing expected values for distributions which have a density, we only need to consider $r$ up to $n_0$. We will add a remark on this.
>
> __Q3: Do activation regions correspond to linear regions (line 130) and vice versa (line 133)? The paragraph says one thing, and the result says another.__
>
> Yes, Lemma 5 shows that for almost every choice of $\theta$, linear regions and 0-partial activation regions correspond to each other. With ''A corresponds to B'' we mean ''A and B correspond to each other''. We will rephrase this for clarity.
>
> __Q4: Figure 1 Right is not clear. Can you please describe it in more detail?__
>
> As indicated in the caption, the full details are provided in Appendix K. The figure shows a heatmap of the function that takes parameter values $\xi$ to the number of linear regions that the function $f_\xi$ has over a fixed portion of the input space. In this figure, we consider parameter values $\xi$ from a 2D square in parameter space. For each parameter value $\xi$, the number of linear regions of the represented function $f_\xi$ is computed numerically over the subset $[0,1]^2$ of the input space $\mathbb{R}^2$. In order to make the figure clearer, we will add examples of the function $f_\xi$ (shown in Fig 11 in the appendix) for a few different values of the parameter $\xi$.
>
> __Q5: Would it be correct to say that Lemma 6 corresponds to every possible partial activation pattern being valid? If so, what is the purpose of stating it if there is a sharper bound in Montufar et. al. (2021)?__
>
> Yes, the total number of activation patterns corresponds to every activation region being non-empty. The value of Lemma 6 lies in the simplicity of the bound, which is easy to parse and use in the proofs. Kindly note that we also provide the exact number of activation patterns in Proposition 14 in Appendix A. The bounds in Montufar et al 2021 are for the number of non-empty 0-partial activation regions ($r=0$) for either shallow networks or for certain types of deep networks. In contrast, our bound in Lemma 6 (or Proposition 14) is valid for any architecture and for any $r$.
>
> __Q6: How can a linear region have finite volume (line 158), if in some cases it is unbounded?__
>
> Here we used ''finite volume'' in the sense of ''non-zero volume''. We will rephrase for clarity.
>
> __Q7: Theorem 8 seems to be more of a Lemma or even a Proposition that could also apply to ReLU given hyperplanes in general position.__
>
> The proof of this statement is surprisingly more difficult than one might think at first sight. In our proof we are using 4 propositions and results from Adiprasito 2017 and Montufar et al 2021 (see Appendix B.4).
> If you suggest, we would nonetheless rename this as a Proposition.
>
> We also point out that the statement of Theorem 8 does not apply to ReLU networks unless they have a single layer of ReLUs. Indeed, for a network with 2 layers of ReLUs there exists a positive measure subset of parameters for which the represented functions have only 1 linear region. We will add a remark about this and an example in the appendix.
>
> __Q8: In section 5, can you provide a clear definition by what you mean with the pieces of the decision boundary?__
>
> By ''pieces of the decision boundary'' we mean the $n_0-1$-dimensional polyhedra that build up the decision boundary. More precisely: The decision boundary is a union of certain $r$-partial activation regions for the network with a maxout unit as the output layer. We have results for the number of $r$-partial activation regions that comprise the decision boundary, for any value of $r$ (see Lemma 39 in Appendix H.4). For simplicity, in the main part we presented only the result for the $n_0-1$-dimensional regions, which we called ''pieces''. We will add a proper definition in the revision.
>
> __Q9: In line 159, add "about" between question and which.__
>
> Thank you for pointing out this grammar error. We will fix it in the revision.
>
> __Q10: Misuse of $n_0$ as size of maxout input, since $n_0$ refers to size of input layer, starting in Theorem 7. Likewise for the use of $n_1$ for size of the layer output.__
>
> In Theorem 7, $n_0$ still refers to the size of the input layer and $n_L$ to the size of the last maxout layer. Note that in the first item, the network consists of a single maxout unit with an input layer of size $n_0$, and in the second item, the network has a single maxout layer, so that $L=1$ and $n_L=n_1$. We will emphasize this for clarity.
>
> __Q11: First bullet point of Theorem 7 could be a corollary. Second bullet point of Theorem 7 could be a separate result.__
>
> Thank you for the suggestion. Notice that item 1 is not a corollary since it includes the statement ''and else it is a null set'' which is not covered in item 2 nor 3. We grouped the three items into a single statement since they all address the same question (but for different architectures). We will try to follow your suggestion if space permits.
>
> __Q12: Remove repeated for in line 203.__
>
> Thank you for pointing out this typo. We will fix it in the revision.
>
> __Q13: In line 33, another related work on function-preserving transformations is [1]. In line 34, another related work on robustness based on linear regions is [2].__
>
> Thank you for pointing out these interesting papers related to function preserving transformations and robustness. We will add references in the introduction.
>
> __Q14: In line 268, another related work on counting linear regions is [3] and you should also refer to the sampling method in Xiong et al. (2020), which is applied in [4].__
>
> Thanks for these pointers. In [3] the authors state that they use the approach from (Serra et al., 2018) for counting the regions exactly. Since we are listing different methods for counting the regions, we did not cite [3] but rather the earlier work (Serra et al., 2018). In Xiong et al. (2020), which we already cite in the introduction, the authors count linear regions by randomly sampling data points from the input space and determining which linear regions they belong to. This approach for approximate counting has been known before. For instance, it was mentioned in Hanin and Rolnick (2019a) in the discussion of their exact method. Following your suggestion we will nonetheless mention these works in the discussion of methods for counting linear regions.
>
>
> __Q15: In line 304, is Xavier and He a missing reference?__
>
> Thank you for pointing out this typo. We will fix it in the revision.
>
> [1] https://arxiv.org/abs/2001.00218
>
> [2] https://arxiv.org/abs/1907.03207
>
> [3] https://arxiv.org/abs/1810.03370
>
> [4] https://arxiv.org/abs/2102.11535

---

> > ### Comment · Reviewer_4fmj · 2021-09-03
> > **Following up**
> >
> > I would like to thank the authors for their careful follow up and commitment to improve the manuscript.
> >
> > It is always a challenge to review a heavily theoretical paper, but from the combination of reviews and perspectives of the other reviewers I feel sufficiently confident now that your work meets the standards.
> >
> > I will keep my positive score.

---

### Official Review · Reviewer_Udb7 · 2021-07-17

**Rating:** 6
**Confidence:** 4

**Summary:**

EDIT: I have read the authors' response and decided to keep my (positive) score.

The paper studies the problem of expressivity of feedforward neural networks with maxout (multi-argument) activation functions. Maxout units compute parametric affine functions followed by a fixed multi-argument activation function of the form (s1, . . . , sK)→max{s1, . . . , sK}.

A special case of course is ReLU nets, and many prior results have been obtained, showing how the activation patterns and linear regions of the network depend on the architecure of the network, mainly the depth and the width. For example, Telgarsky's "Benefits of depth in neural networks." paper, shows how depth in neural nets can be exponentially more efficient that width in representing certain functions like the triangular wave and compositions with itself.

The question that is raised here however is how do the linear regions of maxout nets look like for "practical" neural nets. Practical here, just means networks initialized with a multitude of different distributions that are common in practice. The importance of this question is motivated by understanding average case performance instead of worst-case performance. This reflects an approach originally put forth by two works of Hanin and Rolnick.

Briefly, the main results of the paper generalize prior results for ReLUs to the case of maxout networks. For example, the main result of the paper is Theorem 9 and 10 that state that for most parameter settings in the network, the expected number of activation regions is polynomial in the number of units. This comes in contrast with Telgarsky's and follow-up works that give exponential bounds for deep ReLU nets.





**Main Review:**

This is an interesting paper on a very timely topic for deep learning theory. It gives several upper and lower bounds for the activation regions of maxout networks, capturing prior works on ReLU nets.

Many of the technical details rely on the works of Hanin and Rolnick and ideas in Montufar et al., however dealing with maxout activations is certainly more challenging. The techniques for analyzing ReLUs need significant changes for the arguments to pass through and altough it may appear straightforward, it certainly isn't.

The main result itself may not be too surprising given the prior results for ReLUs however the authors analyze several related questions and give a nice collection of results that deepens our understanding of activation regions, expected volume of activation regions, initialization schemes and their behavior and because of that I vote for weak accept.

Questions to the authors:

The initialization process used here is similar to Hanin and Rolnick and it seems to imply that linear regions depend only the size of the network (and only with polynomial dependence) while Telgarsky's result states that in worst-case there is exponential dependence on the depth. On the other hand, several follow-up works have shown that Telgarsky's triangle wave construction is not the only to yield exponential dependence on depth; in fact, the reason behind such exponential dependencies actually comes from ideas in dynamical systems and in particular chaos theory (see more on "Depth-Width Trade-offs for ReLU Networks via Sharkovsky's Theorem", "Exponential expressivity in deep neural networks through transient chaos", and "Better Depth-Width Trade-offs for Neural Networks through the lens of Dynamical Systems"). The reviewer is curious to see how the parameter settings used in the present paper (and Hanin's works for that matter) can break such worst-case results since they do not particularly depend on exact parameter settings like Telgarsky's?

A bit more braodly, the results in the paper seem to suggest that depth doesn't matter. Can you elaborate on why depth stops being important in such settings and only the size of the network matters? Is this sth that the authors think is in accordance with what practitioners seem to be using in practice? If not, why should we even use deep nets, if initiliazation tells us that linear regions won't significantly change (as long as size is same, we can just increase width not depth)?

**Time Spent Reviewing:**

4

---

> ### Author Response · Authors · 2021-08-10
> **Response to Reviewer Udb7**
>
> Thank you for the valuable feedback and appreciating the challenge of working with maxout networks. Please see below for our response to your questions.
>
> __Q1: How the parameter settings used in the present paper (and Hanin's works for that matter) can break such worst-case results since they do not particularly depend on exact parameter settings like Telgarsky's?__
>
> This is a good observation that we explain as follows. There are indeed several choices of parameters that result in an exponential number of linear regions for deep networks (in fact, the set of parameters for which this happens has positive Lebesgue measure, as we discuss in line 158 and line 718 of the appendix, which is also argued informally in Montufar et al 2014 and proven formally in Montufar et al 2021). We also observed in Theorem 7 that for specific parameter distributions, the expected number of linear regions is exponential.
>
> Nonetheless, the set of parameter values with an exponential number of linear regions does have a small measure for common distributions, which satisfy the $C_{bias}$ and $C_{grad}$ bounds that we consider. The intuitive reason is as follows. For an exponential number of linear regions to occur, each layer needs to subdivide a significant fraction of the regions of the function computed up to the preceding layer. This means that the $l$th layer has a non-linear locus that cuts through the range of the oscillations of the function computed up to the $(l-1)$th layer. This is only possible when either the bias of the $l$th layer is finely tuned to a small range of values, or when the amplitude of the oscillations is very high. But the probability of these conditions happening is controlled by the density of the bias distribution ($C_{bias}$) and the expectation value of the gradient norms ($C_{grad}$).
>
> __Q2: Can you elaborate on why depth stops being important in such settings and only the size of the network matters? Is this sth that the authors think is in accordance with what practitioners seem to be using in practice? If not, why should we even use deep nets, if initiliazation tells us that linear regions won't significantly change (as long as size is same, we can just increase width not depth)?__
>
> We believe that depth still is important, even if not, or not just, for implementing functions with many linear regions. For instance, even if the number of linear regions might be similar in shallow and deep networks, the shape of the linear regions can be very different. Also, two networks that are capable of representing the same functions can still have very different training behavior due to different parametrizations. We think that the comparison of deep and shallow networks involves all of these effects. The biases that are implemented implicitly by the parameter optimization procedures depending on the parametrization are an active topic of investigation. The explicit form of these biases is currently not very well understood for deep networks. Our results indicate that the form and utility of these biases might involve properties other than the number of linear regions.

---

### Author Response · Authors · 2021-09-01
**Note from the Authors**

Dear reviewers and AC,

Thank you for your work and your positive evaluation of our submission. We already provided detailed responses to your initial reviews. As the discussion period is coming to the end, we wanted to ask again if there are any further details or clarifications that we could provide to facilitate your discussion.


Best regards,

Authors

---

> ### Comment · Reviewer_Udb7 · 2021-09-02
> **keeping my (positive) score**
>
> After reading again the paper, the authors' responses and the opinions of the other reviewers, I have decided to keep my score.

---

### Decision · Program_Chairs · 2021-09-27

**Decision:**

Accept (Poster)

**Comment:**

This paper considers the number of linear regions (and pieces of the decision boundary) of maxout networks at initialization, showing that these numbers grow polynomially in expectation, instead of exponentially as in the worst case. This is an interesting theoretical investigation building on prior work answering the same question for ReLU networks. The reviewers note that the approaches used in this paper are somewhat incremental to this prior work, and that the upper bound on the constant C_grad presented in the paper's formal results has an exponential dependence on depth for natural initializations. However, the reviewers agree that these points do not detract from the interest and validity of the paper's results, and I recommend acceptance.